# Reward Distance Comparisons Under Transition Sparsity

**Clement Nyanhongo**                    *clement.k.nyanhongo.th@dartmouth.edu*
*Thayer School of Engineering*
*Dartmouth College*

**Bruno Miranda Henrique**              *bruno.miranda.henrique.th@dartmouth.edu*
*Thayer School of Engineering*
*Dartmouth College*

**Eugene Santos Jr.**                     *eugene.santos.jr@dartmouth.edu*
*Thayer School of Engineering*
*Dartmouth College*

**Reviewed on OpenReview:** *https://openreview.net/forum?id=haP586YomL*

## Abstract

Reward comparisons are vital for evaluating differences in agent behaviors induced by a set of reward functions. Most conventional techniques utilize the input reward functions to learn optimized policies, which are then used to compare agent behaviors. However, learning these policies can be computationally expensive and can also raise safety concerns. Direct reward comparison techniques obviate policy learning but suffer from transition sparsity, where only a small subset of transitions are sampled due to data collection challenges and feasibility constraints. Existing state-of-the-art direct reward comparison methods are ill-suited for these sparse conditions since they require high transition coverage, where the majority of transitions from a given coverage distribution are sampled. When this requirement is not satisfied, a distribution mismatch between sampled and expected transitions can occur, leading to significant errors. This paper introduces the Sparsity Resilient Reward Distance (SRRD) pseudometric, designed to eliminate the need for high transition coverage by accommodating diverse sample distributions, which are common under transition sparsity. We provide theoretical justification for SRRD's robustness and conduct experiments to demonstrate its practical efficacy across multiple domains.

## 1 Introduction

In sequential decision problems, reward functions often serve as the most "succinct, robust, and transferable" representations of a task (Abbeel & Ng, 2004), encapsulating agent goals, social norms, and intelligence (Silver et al., 2021; Zahavy et al., 2021; Singh et al., 2009). For problems where a reward function is specified and the goal is to find an optimal policy that maximizes cumulative rewards, Reinforcement Learning (RL) is predominantly employed (Sutton & Barto, 2018). Conversely, when a reward function is complex or difficult to specify and past expert demonstrations (or policies) are available, the reward function can be learned via Inverse Reinforcement Learning (IRL) (Ng & Russell, 2000).

In both RL and IRL paradigms, reward functions govern agent decision-making, and reward comparisons can help assess the similarity of these functions in terms of their induced behaviors. The task of reward comparisons aims to assess the similarity among a collection of reward functions. This can be done through pairwise comparisons where the similarity distance $D(R_A, R_B)$ between two reward functions, $R_A$ and $R_B$ (vectors, not scalars), is computed. The similarity distance should reflect variations not only in magnitude but also in the preferences and behaviors induced by the reward functions. This is characterized by the property of policy invariance, which ensures that reward functions yielding the same optimal policies are considered

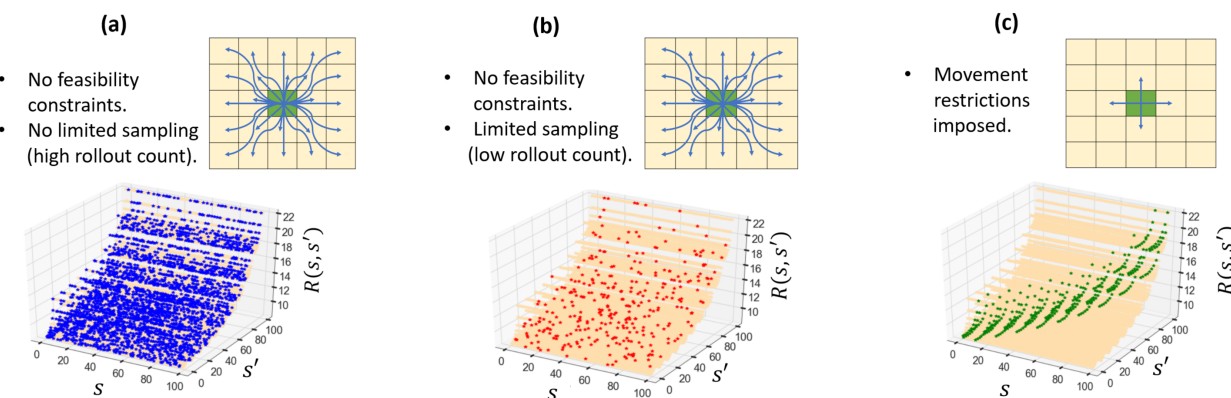

Figure 1: (Transition Sparsity in a $10 \times 10$ Gridworld Domain) In this illustration, each transition starts from a starting state $s$ and ends in a destination state $s'$. For clarity in visualization, we consider action-independent rewards, $R(s, s')$. In (**a**), high transition coverage results from a high rollout count (number of policy rollouts) in the absence of feasibility constraints, leading to the majority of transitions being sampled (blue points). In (**b**), low coverage results from a low rollout count in the absence of feasibility constraints, leading to fewer sampled transitions (red points). In (**c**), low coverage results from feasibility constraints, such as movement restrictions that only allow actions to adjacent cells, which can significantly reduce the space of sampled transitions (green points) irrespective of rollout count.

similar even if their numerical reward values differ (Ng et al., 1999). This makes direct reward comparisons via distance measures such as Euclidean or Kullback-Leibler (KL) divergence unfavorable since these distances do not maintain policy invariance. To satisfy policy invariance, traditional reward comparison techniques have adopted indirect approaches that compare behaviors derived from the optimized policies associated with the reward functions under comparison (Arora & Doshi, 2021). However, these indirect approaches pose the following challenges: (1) they can be slow and resource-intensive due to iterative policy learning via RL, and (2) policy learning may not be favorable in critical online environments such as healthcare or autonomous vehicles, where safety considerations are paramount (Amodei et al., 2016; Thomas et al., 2021). Therefore, developing direct reward comparison methods that bypass the computationally expensive process of policy learning, while maintaining policy invariance is highly important.

To achieve policy invariance in direct reward comparisons, Gleave et al. (2021) introduced the Equivalent Policy Invariant Comparison (EPIC) pseudometric. Given the task to compare two reward functions, EPIC first performs reward canonicalization to express rewards in a standardized form by removing shaping, and then computes the Pearson distance to measure the difference between the canonical reward functions. Although theoretically rigorous, EPIC falls short in practice since it is designed to compare reward functions under high transition coverage, when the majority of transitions within the explored state and action space are sampled. In many practical scenarios, achieving high transition coverage can be impractical due to transition sparsity, a condition where a minority of transitions are sampled. The remaining unsampled transitions may have unknown or undefined reward values (especially if unrealizable), which can distort the computation of reward expectations during the canonicalization process.

Transition sparsity can be attributed to: (1) **limited sampling** - when data collection challenges result in fewer sampled transitions; and (2) **feasibility constraints** - when environmental or agent-specific limitations restrict certain transitions. Consider, for instance, a standard $10 \times 10$ Gridworld domain which has a total of 100 states, each represented by an $(x, y)$ coordinate. The total number of possible transitions is at least $10,000$ (100 states $\times$ number of actions $\times$ 100 states) if at least one action exists between every pair of states (see Figure 1). However, feasibility constraints such as movement restrictions might limit the transitions that can be sampled. For example, an agent that can only move to its neighboring states by taking single-step cardinal directions (actions: up, right, down, left) in each state will explore fewer than 400 transitions (100 states $\times$4 actions), as shown in Figure 1c. To illustrate the impact of limited sampling, consider a scenario where transitions are sampled via policy rollouts. Assuming that factors such as feasibility

constraints, the transition model, and the policy rollout method are kept constant, the extent of sampled transitions is proportional to the number of policy rollouts (see Figure 1a and 1b). To alleviate the impact of transition sparsity in reward comparisons, we present the Sparsity Resilient Reward Distance (SRRD) pseudometric, which does not assume the existence of reward samples with high transition coverage. During canonicalization, SRRD introduces additional reward expectation terms to accommodate a wider and more diverse range of transition distributions.

In practical settings, reward comparisons can be useful for a broad range of applications such as: (1) *Evaluating Agent Behaviors* – By comparing how different reward functions align or differ using specified similarity measures, agent rewards can be grouped, to reason and interpret different agent behaviors. This can be useful in IRL domains, where there is need to extract meaning from inferred rewards computed to represent agent preferences (Ng & Russell, 2000; Santos & Nyanhongo, 2019). For instance, in sport domains such as hockey, reward comparisons could be useful in inferring player rankings and their decision-making strategies (Luo et al., 2020). (2) *Initial Reward Screening* – In RL domains, direct reward comparisons could serve as a preliminary step to quickly identify rewards that will achieve a spectrum of desired behaviors before actual training. This could be beneficial in scenarios where multiple possible reward configurations exist, but some might be more efficient. (3) *Addressing Reward Sparsity*[1] – Reward comparisons could also tackle issues such as reward sparsity, by identifying more informative and easier-to-learn reward functions that might be similar in terms of optimal policies but are more desirable than sparse reward functions.

**Contributions**   In this paper, we introduce the Sparsity Resilient Reward Distance (SRRD) pseudometric, designed to improve direct reward comparisons in environments characterized by high transition sparsity. SRRD demonstrates greater robustness compared to existing pseudometrics (such as EPIC), which require high transition coverage. SRRD's strength lies in its ability to integrate reward samples with diverse transition distributions, which are common in scenarios with low coverage. We provide the theoretical justification for SRRD's robustness and demonstrate its superiority through experiments in several domains of varying complexity: Gridworld, Bouncing Balls, Drone Combat, Robomimic, Montezuma's Revenge, StarCraft II, and MIMIC-IV. For the simpler domains, Gridworld and Bouncing Balls, we evaluate SRRD against manually defined factors such as nonlinear reward functions and feasibility constraints, to fully understand its strengths and limitations under controlled conditions. In the more complex domains such as StarCraft II, we assess SRRD in environments characterized by large state and action spaces, to gauge how it is likely to perform in realistic settings. Our final experiment explores a novel and practical application of these pseudometrics as distance measures within a *k*-nearest neighbors algorithm, tailored to classify agent behaviors based on reward functions computed via IRL. Empirical results highlight SRRD's superior performance, as evidenced by its ability to find higher similarity between rewards generated from the same agents and higher variation between rewards from different agents. These results underscore the crucial need to account for transition sparsity in direct reward comparisons.

## 2   Related Works

The EPIC pseudometric is the first direct reward comparison technique that circumvents policy learning while maintaining policy invariance (Gleave et al., 2021). In practical settings, EPIC's major limitation is that it is designed to compare rewards under high transition coverage. In scenarios characterized by transition sparsity, EPIC underperforms due to its high sensitivity to unsampled transitions, which can distort the computation of reliable reward expectation terms, needed during canonicalization. This limitation has been observed by Wulfe et al. (2022), who introduced the Dynamics-Aware Reward Distance (DARD) pseudometric. DARD improves on EPIC by relying on transitions that are closer to being physically realizable; however, it still remains highly sensitive to unsampled transitions.

Skalse et al. (2024) also introduced a family of reward comparison pseudometrics, known as Standardized Reward Comparisons (STARC). These pseudometrics are shown to induce lower and upper bounds on worst-case regret, implying that the metrics are tight, and differences in STARC distances between two reward

---

[1]Transition sparsity arises when a minority of transitions are sampled. This is different from reward sparsity, which occurs when rewards are infrequent or sparse, making RL tasks difficult.

functions correspond to differences in agent behaviors. Among the different STARC metrics explored, the Value-Adjusted Levelling (VAL) and the VALPotential functions are empirically shown to have a marginally tighter correlation with worst-case regret compared to both EPIC and DARD. While an improvement, a significant limitation of these metrics is their reliance on value functions, which can be computed via policy evaluation—a process that incurs a substantially higher computational overhead than sample-based approximations for both EPIC and DARD. In small environments, these metrics can be computed using dynamic programming for policy evaluation, an iterative process with polynomial complexity relative to the state and action spaces (Skalse et al., 2024). In larger environments, computing the exact value functions becomes impractical hence the value functions need to be approximated via neural networks updated with Bellman updates (Skalse et al., 2024). Since the primary motivation for direct reward comparisons is to eliminate the computationally expensive process of policy learning, incorporating value functions is somewhat contradictory since policy evaluation is iterative, and it can have comparable complexity with policy learning techniques such as value iteration. Our work focuses on computationally scalable direct reward comparison pseudometrics (such as EPIC and DARD), which do not involve iterative policy learning or evaluation.

The task of reward comparisons lies within the broader theme of reward evaluations, which aim to explain or interpret the relationship between rewards and agent behavior. Some notable works tackling this theme, include, Lambert et al. (2024), who developed benchmarks to evaluate reward models in Large Language Models (LLMs), which are often fine-tuned using RL via human feedback (RLHF) to align the rewards with human values. These benchmarks assess criteria such as communication, safety and reasoning capabilities across a variety of reward models. In another line of work, Mahmud et al. (2023) presented a framework leveraging human explanations to evaluate and realign rewards for agents trained via IRL on limited data. Lastly, Russell & Santos (2019) proposed a method that examines the consistency between global and local explanations, to determine the extent to which a reward model can capture complex agent behavior. Similar to reward comparisons, reward evaluations can be influenced by shaping functions, thus necessitating techniques such as canonicalization as preprocessing steps to eliminate shaping (Jenner & Gleave, 2022).

Reward shaping is a technique that transforms a base reward function into alternate forms (Ng et al., 1999). This technique is mainly employed in RL for reward design where heuristics and domain knowledge are integrated to accelerate learning (Mataric, 1994; Hu et al., 2020; Cheng et al., 2021; Gupta et al., 2022; Suay et al., 2016). Several applications of reward shaping have been explored, and some notable examples include: training autonomous robots for navigation (Tenorio-Gonzalez et al., 2010); training agents to ride bicycles (Randløv & Alstrøm, 1998); improving agent behavior in multiagent contexts such as the Prisoner's Dilemma (Babes et al., 2008); and scaling RL algorithms in complex games (Lample & Chaplot, 2017; Christiano et al., 2017). Among several reward shaping techniques, potential-based shaping is the most popular due to its preservation of policy invariance, ensuring that the set of optimal policies remains unchanged between different versions of reward functions (Ng et al., 1999; Wiewiora et al., 2003; Gao & Toni, 2015).

## 3 Preliminaries

This section introduces the foundational concepts necessary for understanding direct reward comparisons, and the critical challenge of transition sparsity which our proposed approach, SRRD (detailed in Section 4), is designed to address. We begin by outlining the Markov Decision Process formalism, followed by a discussion of policy invariance. Finally, we review the existing key direct reward comparison pseudometrics (EPIC and DARD) and examine their limitations, which motivate the development of SRRD.

### 3.1 Markov Decision Processes

A Markov Decision Process (MDP) is defined as a tuple $(\mathcal{S}, \mathcal{A}, \gamma, T, R)$, where $\mathcal{S}$ and $\mathcal{A}$ are the state and action spaces, respectively. The transition model $T : \mathcal{S} \times \mathcal{A} \times \mathcal{S} \to [0, 1]$, dictates the probability distribution of moving from one state, $s \in \mathcal{S}$, to another state, $s' \in \mathcal{S}$, under an action $a \in \mathcal{A}$, and each given transition is specified by the tuple $(s, a, s')$. The discount factor $\gamma \in [0, 1]$ reflects the preference for immediate over future rewards. The reward function $R : \mathcal{S} \times \mathcal{A} \times \mathcal{S} \to \mathbb{R}$ assigns a reward $R(s, a, s')$ to each transition. A trajectory $\tau = \{(s_0, a_0), (s_1, a_1), \cdots, (s_n)\}$, $n \in \mathbb{Z}^+$, is a sequence of states and actions, with a total return:

$g(\tau) = \sum_{t=0}^{\infty} \gamma^t R(s_t, a_t, s_{t+1})$. The goal in an MDP is to find a policy $\pi : \mathcal{S} \times \mathcal{A} \to [0, 1]$ (often via RL) that maximizes the expected return $\mathbb{E}[g(\tau)]$.

Given the subsets $S_i \subseteq \mathcal{S}$, $A \subseteq \mathcal{A}$, and $S_j \subseteq \mathcal{S}$, the tuple $(S_i, A, S_j)$ represents the set of transitions within the cross-product $S_i \times A \times S_j$. The associated rewards are:

$$R(S_i, A, S_j) = \{R(s, a, s')|(s, a, s') \in S_i \times A \times S_j\},$$

and the expected reward over these transitions is denoted by $\mathbb{E}[R(S_i, A, S_j)]$. In the standard MDP formulation, the reward function is fully specified for all possible transitions including those that may be unrealizable. However, in many practical settings, such as offline RL (Levine et al., 2020; Agarwal et al., 2020; Chen et al., 2024), we are often limited to datasets of reward samples that are only defined over a subset of observed realizable transitions. In this paper, a **reward sample** is defined as a restriction of $R$ to a subset $B \subseteq \mathcal{S} \times \mathcal{A} \times \mathcal{S}$, where $B$ consists of sampled transitions under a specified policy. We assume that rewards are defined for transitions in $B$, and are undefined for transitions not in $B$. Given a batch of sampled transitions $B$, the coverage distribution $\mathcal{D}(s, a, s')$ defines the probability distribution over transitions used to generate $B$. The sampled state space and action space are denoted by $S^{\mathcal{D}} \subseteq \mathcal{S}$ and $A^{\mathcal{D}} \subseteq \mathcal{A}$. The sets of all possible distributions over $\mathcal{A}$ and $\mathcal{S}$ are denoted by $\Delta\mathcal{A}$ and $\Delta\mathcal{S}$ respectively, and the individual distributions over states and actions are denoted by $\mathcal{D}_{\mathcal{S}} \in \Delta\mathcal{S}$ and $\mathcal{D}_{\mathcal{A}} \in \Delta\mathcal{A}$.

### 3.2 Policy Invariance and Direct Reward Comparisons

In direct reward comparisons, policy invariance is crucial as it ensures that reward functions that differ due to potential shaping are treated as equivalent, since they yield the same optimal policies (Ng et al., 1999). Formally, any shaped reward can be represented by the relationship: $R'(s, a, s') = R(s, a, s') + F(s, a, s')$, where $F$ is a shaping function. Potential shaping guarantees policy invariance, and takes the form:

$$R'(s, a, s') = R(s, a, s') + \gamma\phi(s') - \phi(s), \tag{1}$$

where $R$ is the original reward function, and $\phi$ is a state-potential function. Reward functions $R$ and $R'$ are deemed equivalent as they yield the same optimal policies. To effectively compare reward functions that may differ numerically but induce the same optimal policies, the use of **pseudometrics** is highly important. Let $X$ be a set, with $x, y, z \in X$, and let $d : X \times X \to [0, \infty)$ define a pseudometric. This pseudometric adheres to the following axioms: (premetric) $d(x, x) = 0$ for all $x \in X$; (symmetry) $d(x, y) = d(y, x)$ for all $x, y \in X$; and (triangular inequality) $d(x, y) \leq d(x, z) + d(z, y)$ for all $x, y, z \in X$. Unlike a true metric, a pseudometric does not require that: $d(x, y) = 0 \implies x = y$, making it ideal for identifying equivalent reward functions that might have different numerical values.

The EPIC pseudometric was introduced by Gleave et al. (2021), as a direct reward comparison method that maintains policy invariance. To compute EPIC, reward functions are first transformed into a canonical form without potential shaping; and then, the Pearson distance is computed to differentiate the canonical rewards. The EPIC canonicalization function is defined as follows:

$$C_{EPIC}(R)(s, a, s') = R(s, a, s') + \mathbb{E}[\gamma R(s', A, S') - R(s, A, S') - \gamma R(S, A, S')], \tag{2}$$

where, $S \sim \mathcal{D}_{\mathcal{S}}$, $S' \sim \mathcal{D}_{\mathcal{S}}$, $A \sim \mathcal{D}_{\mathcal{A}}$, with $\mathcal{D}_{\mathcal{S}}$ and $\mathcal{D}_{\mathcal{A}}$ being distributions over states and actions, respectively. Given a potentially shaped reward, $R'(s, a, s') = R(s, a, s') + \gamma\phi(s') - \phi(s)$, canonicalization yields: $C_{EPIC}(R')(s, a, s') = C_{EPIC}(R)(s, a, s') + \phi_{res}$, where $\phi_{res} = \gamma\mathbb{E}[\phi(S)] - \gamma\mathbb{E}[\phi(S')]$ is the remaining residual shaping. EPIC assumes that $S$ and $S'$ are identically distributed such that $\mathbb{E}[\phi(S)] = \mathbb{E}[\phi(S')]$, resulting in $\phi_{res} = 0$. This makes EPIC, invariant to potential shaping, since, $C_{EPIC}(R) = C_{EPIC}(R')$. Finally, the EPIC pseudometric between two reward functions $R_A$ and $R_B$ is computed as:

$$D_{\text{EPIC}}(R_A, R_B) = D_\rho(C_{EPIC}(R_A), C_{EPIC}(R_B)), \tag{3}$$

where, for any random variables $X$ and $Y$, the Pearson distance $D_\rho$ is defined as:

$$D_\rho(X, Y) = \sqrt{1 - \rho(X, Y)}/\sqrt{2}. \tag{4}$$

The Pearson correlation coefficient is given by: $\rho(X, Y) = \mathbb{E}[(X - \mu_X)(Y - \mu_Y)]/(\sigma_X \sigma_Y)$, where $\mu$ denotes the mean, $\sigma$ the standard deviation, and $\mathbb{E}[(X - \mu_X)(Y - \mu_Y)]$ is the covariance between $X$ and $Y$. The Pearson distance is defined over the range: $0 \leq D_\rho(X, Y) \leq 1$, where $D_\rho(X, Y) = 0$ indicates that $X$ and $Y$ are highly similar since $\rho(X, Y) = 1$ (perfect positive correlation), and $D_\rho(X, Y) = 1$ indicates that $X$ and $Y$ are maximally different since $\rho(X, Y) = -1$. The Pearson distance is scale and shift invariant since $D_\rho(aX + c, bY + d) = D_\rho(X, Y)$, where, $a, b, c, d \in \mathbb{R}$ are constants (Gleave et al., 2021). Therefore, the EPIC pseudometric is scale, shift and shaping invariant, which are policy-preserving transformations. Computing $C_{EPIC}$ requires access to all transitions in a reward function, making it feasible only for small environments. For reward functions with large or infinite state and action spaces, Gleave et al. (2021) introduced the sample-based EPIC approximation, denoted as $\hat{C}_{EPIC}$, and is computed as:

$$\hat{C}_{EPIC}(R)(s, a, s') = R(s, a, s') + \frac{\gamma}{N_M} \sum_{(x,u) \in B_M} R(s', u, x) - \frac{1}{N_M} \sum_{(x,u) \in B_M} R(s, u, x)$$
$$- \frac{\gamma}{N_M^2} \sum_{(x,\cdot) \in B_M} \sum_{(x',u) \in B_M} R(x, u, x'), \tag{5}$$

where transitions are sampled from a batch $B_V$ of $N_V$ samples from a coverage distribution $\mathcal{D}$, and state-action pairs, are sampled from a batch $B_M$ of $N_M$ samples from the joint state and action distributions, $\mathcal{D}_S \times \mathcal{D}_A$. Each term in $\hat{C}_{EPIC}$ approximates the corresponding expectation term in $C_{EPIC}$ (Equation 2), for example, $\frac{\gamma}{N_M} \sum_{(x,u) \in B_M} R(s', u, x)$ estimates $\mathbb{E}[\gamma R(s', A, S')]$.

Wulfe et al. (2022) observed that EPIC often depends on transitions that are physically unrealizable, as it requires all (both realizable and unrealizable) transitions from the space $S \times A \times S'$. The unrealizable transitions can introduce errors in reward comparisons since rewards for these transitions are often arbitrary and unreliable. To mitigate this challenge, the authors introduced DARD, which incorporates transition models to prioritize physically realizable transitions. The DARD canonicalization function is given by:

$$C_{DARD}(R)(s, a, s') = R(s, a, s') + \mathbb{E}[\gamma R(s', A, S'') - R(s, A, S') - \gamma R(S', A, S'')], \tag{6}$$

where $A \sim \mathcal{D}_A$, $S' \sim T(s, A)$, $S'' \sim T(s', A)$, and $T$ is the transition model. DARD is invariant to potential shaping and generally improves upon EPIC by distinguishing between the subsequent states to $s$ (denoted by $S'$) and $s'$ (denoted by $S''$). Transitions $(s, A, S')$ and $(s', A, S'')$ are generally in-distribution with respect to the transition dynamics $T$ since $S'$ is distributed conditionally based on $(s, A)$, and $S''$ is distributed conditionally based on $(s', A)$. Therefore, these transitions naturally align with the dynamics of the sampled transitions, resulting in a lower likelihood of being unrealizable. However, transitions $(S', A, S'')$ are more likely to be out-of-distribution with respect to the transition dynamics $T$, since $S''$ is not distributed conditionally according to $(S', A)$, but rather to $(s', A)$. Consequently, DARD can be sensitive to out-of-distribution transitions in $(S', A, S'')$, which have a higher likelihood of being unrealizable. Nonetheless, Wulfe et al. (2022) argue that these transitions are closer to being physically realizable since $(S', A, S'')$ transitions are in close proximity to $s$ and $s'$, compared to transitions that are utilized in EPIC. Computing the exact DARD computation can also be impractical in large environments, hence, Wulfe et al. (2022) introduced a sample-based DARD approximation, denoted by $\hat{C}_{DARD}$ (see Appendix A.1).

### 3.3 Unsampled Transitions

Consider a reward function $R : \mathcal{S} \times \mathcal{A} \times \mathcal{S} \rightarrow \mathbb{R}$, where $\mathcal{S}$ is the state space and $\mathcal{A}$ is the action space. A reward sample is generated according to a coverage distribution $\mathcal{D}$, and it spans a state space $S^\mathcal{D} \subseteq \mathcal{S}$ and an action space $A^\mathcal{D} \subseteq \mathcal{A}$. We define the following sets of transitions:

**Full Coverage Transitions ($\mathcal{T}^\mathcal{D}$)** - The set of all theoretically possible transitions within the reward sample's state-action space. This set is represented as $\mathcal{T}^\mathcal{D} = S^\mathcal{D} \times A^\mathcal{D} \times S^\mathcal{D} \subseteq \mathcal{S} \times \mathcal{A} \times \mathcal{S}$.

**Sampled Transitions ($\mathcal{T}^S$)** - The set of transitions that are actually present in the reward sample. Due to feasibility constraints and limited sampling, this set is a subset of the full coverage transitions: $\mathcal{T}^S \subseteq \mathcal{T}^\mathcal{D}$.

**Unsampled Transitions ($\mathcal{T}^U$)** - The set of full coverage transitions that are not explored in the reward sample. These transitions can be both realizable and unrealizable, and $\mathcal{T}^U = \mathcal{T}^\mathcal{D} \setminus \mathcal{T}^S$.

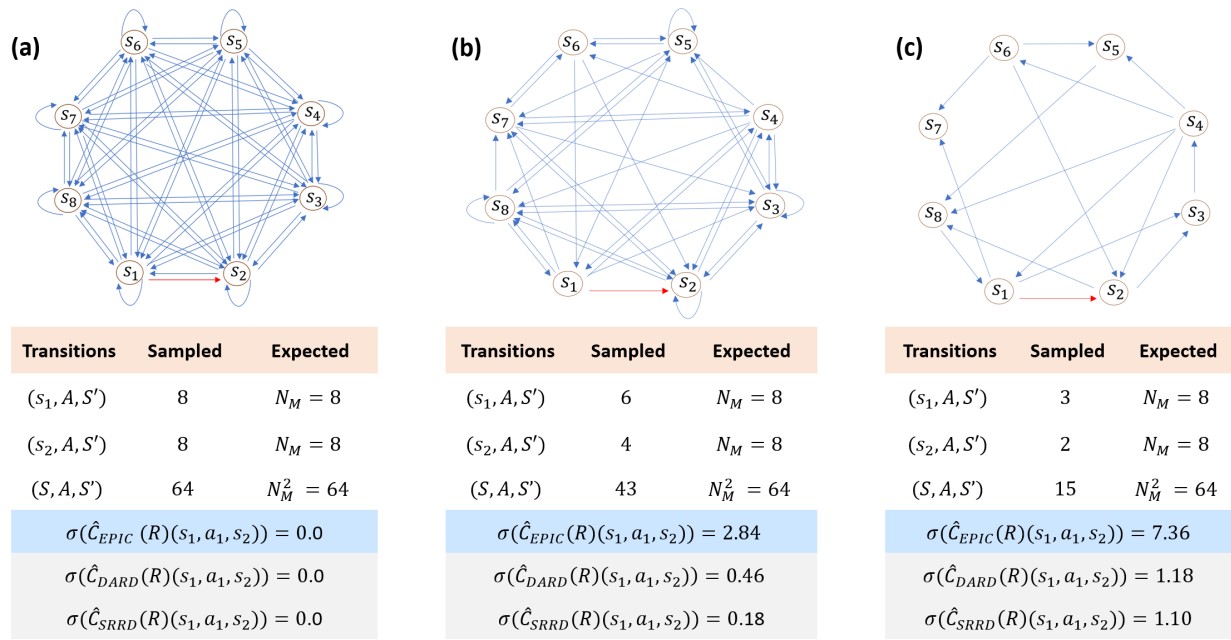

Figure 2: (Impact of unsampled transitions on canonicalizing $R(s_1, a_1, s_2)$) Sampled transitions are those explored in the reward sample, while expected transitions are those anticipated by $\hat{C}_{EPIC}$ assuming full coverage. As coverage decreases from (**a**) to (**c**), due to a reduction in the number of sampled transitions, the standard deviation of $\hat{C}_{EPIC}(R)(s_1, a, s_2)$ increases, indicating $\hat{C}_{EPIC}$'s increased instability to unsampled transitions. For comparison, $\hat{C}_{SRRD}$ and $\hat{C}_{DARD}$ have lower standard deviations, signifying higher stability.

A major limitation of the EPIC and DARD pseudometrics is that they are designed to compare reward functions under high coverage, where $|\mathcal{T}^S| \approx |\mathcal{T}^{\mathcal{D}}|$. As $|\mathcal{T}^U| \to |T^D|$, the performance of these pseudometrics significantly degrades due to an increase in the number of unsampled transitions. To illustrate this limitation, consider Equation 5 used to approximate $C_{EPIC}$. To perform the computation, we need to estimate: $\mathbb{E}[R(s', A, S')]$ by dividing the sum of rewards from $s'$ to $S'$ by $N_M$ transitions; $\mathbb{E}[R(s, A, S')]$ by dividing the sum of rewards from $s$ to $S'$ by $N_M$ transitions; and $\mathbb{E}[R(S, A, S')]$ by dividing the sum of all rewards from $S$ to $S'$ by $N_M{}^2$ transitions, where $N_M \leq |S^{\mathcal{D}} \times A^{\mathcal{D}}|$ is the size of the state-action pairs in the batch $B_M$. Every state $s \in S$ is ideally expected to have $N_M$ transitions to all other states $S'$, which can be impractical under transition sparsity when some transitions might be unsampled. Since reward summations are divided by large denominators due to $N_M$ (see Equation 5), when coverage is low, the number of sampled transitions needed to estimate the reward expectation terms will be fewer than expected, introducing significant error.

Figure 2 illustrates an example showing the effect of unsampled transitions on canonicalization across three reward samples spanning a state space $S = S' = \{s_1, ..., s_8\}$ and an action space, $A = \{a_1\}$, such that $N_M = 8$, under different levels of transition sparsity. Rewards are defined as $R(s_i, a_1, s_j) = 1 + \gamma\phi(s_j) - \phi(s_i)$, where $i, j \in \{1, ..., 8\}$, and state potentials are randomly generated such that: $|\phi(s)| \leq 20$, with $\gamma = 0.5$. The task is to compute $\hat{C}_{EPIC}(R)(s_1, a_1, s_2)$ over 1000 simulations. For all reward samples, the mean $\mu(\hat{C}_{EPIC}(R)(s_1, a_1, s_2)) \approx 0$, but the standard deviation $\sigma(\hat{C}_{EPIC}(R)(s_1, a_1, s_2))$ varies based on coverage. In Figure 2a, the reward sample has high coverage (100%), hence, the number of observed and expected transitions are equal. In this scenario, EPIC is highly effective and all shaped rewards are mapped to the same value ($\approx 0$), resulting in a standard deviation $\sigma(\hat{C}_{EPIC}(R)(s_1, a_1, s_2)) = 0$, highlighting consistent reward canonicalization. In Figure 2b, the reward sample has moderate coverage and the fraction of unsampled transitions is approximately 33%. As a result, $\sigma(\hat{C}_{EPIC}(R)(s_1, a_1, s_2)) = 2.84$, which is relatively high, signifying EPIC's sensitivity to unsampled transitions. In Figure 2c, the reward sample exhibits low coverage and the fraction of unsampled transitions is approximately 77%, indicating a significant discrepancy between the number of observed and expected transitions. Consequently, $\sigma(\hat{C}_{EPIC}(R)(s_1, a_1, s_2)) = 7.36$, highlighting EPIC's increased instability due to unsampled transitions. For comparison, we include the DARD and SRRD

estimates, which exhibit lower standard deviations, signifying greater stability. DARD reduces the effect of unsampled transitions by relying mostly on transitions that are closer to $s$ and $s'$ ($S'$ and $S''$), which typically comprise a smaller subset of states compared to those required by EPIC. However, this localized focus can make DARD highly sensitive to variations in the composition of transitions between the reward samples under comparison, since it might lack the context of transitions further from $s$ and $s'$, potentially limiting its robustness. In this paper, we use DARD as an experimental baseline.

# 4  Approach: Sparsity Resilient Reward Distance (SRRD)

The motivation behind SRRD is to establish a direct reward comparison technique that imposes minimal assumptions about the structure and distribution of transitions in reward samples, ensuring robustness under transition sparsity. To derive SRRD, we integrate key characteristics of $C_{DARD}$ and $C_{EPIC}$ as follows:

- $C_{DARD}$ eliminates the requirement that the set of states that can be reached from $s$ and $s'$ must be similar, thereby increasing the flexibility of transition sample distributions considered. We will refer to states that can be reached from $s'$ as $S_1$, and states from $s$ as $S_2$.

- In $C_{DARD}$, the transitions $(S', A, S'')$ are generally in close proximity to $s$ and $s'$ because $S' \sim T(s, A)$ and $S'' \sim T(s', A)$. These transitions may not capture the states that are further away from $s$ and $s'$, potentially lacking the overall context of the reward sample. To address this issue, similar to how $C_{EPIC}$ uses transitions for the entire sample, $(S, A, S')$, we utilize transitions for the entire sample, which we denote as: $(S_3, A, S_4)$, where $S_3$ encompasses all initial states from the sampled transitions, and $S_4$ is the set of all states that are subsequent to $S_3$.

These modifications reduce the impact of unsampled transitions, as reward expectations are computed based on the observed structure of the sampled transitions, without assuming full coverage. With these considerations, we derive the modified canonical equation as:

$$C_1(R)(s, a, s') = R(s, a, s') + \mathbb{E}[\gamma R(s', A, S_1) - R(s, A, S_2) - \gamma R(S_3, A, S_4)], \tag{7}$$

where: $S_1$ and $S_2$ are subsequent states to $s'$ and $s$, respectively; $S_3$ encompasses all initial states from all sampled transitions; and $S_4$ are subsequent states to $S_3$. Applying $C_1$ to a potentially shaped reward $R'(s, a, s') = R(s, a, s') + \gamma\phi(s') - \phi(s)$, we get: $C_1(R')(s, a, s') = C_1(R)(s, a, s') + \phi_{res1}$, where,

$$\phi_{res1} = \mathbb{E}[\gamma^2\phi(S_1) - \gamma^2\phi(S_4) + \gamma\phi(S_3) - \gamma\phi(S_2)]. \tag{8}$$

$C_1$ is not theoretically robust since it yields the residual shaping $\phi_{res1}$ (the remaining shaping after canonicalization). To cancel $\mathbb{E}[\phi(S_i)], \; \forall_i \in \{1, ..., 4\}$ in $\phi_{res1}$, we can add rewards $R(S_i, A, k_i)$ to induce potentials $\gamma\phi(k_i) - \phi(S_i)$; where $k_i$ can be any arbitrary set of states. This results in the equation:

$$\begin{aligned} C_2(R)(s, a, s') = {} & R(s, a, s') + \mathbb{E}[\gamma R(s', A, S_1) - R(s, A, S_2) - \gamma R(S_3, A, S_4) \\ & + \gamma^2 R(S_1, A, k_1) - \gamma R(S_2, A, k_2) + \gamma R(S_3, A, k_3) - \gamma^2 R(S_4, A, k_4)]. \end{aligned} \tag{9}$$

Applying $C_2$ to a potentially shaped reward, we get: $C_2(R')(s, a, s') = C_2(R)(s, a, s') + \phi_{res2}$, where,

$$\phi_{res2} = \mathbb{E}[\gamma^3\phi(k_1) - \gamma^3\phi(k_4) + \gamma^2\phi(k_3) - \gamma^2\phi(k_2)]. \tag{10}$$

*See Appendix A.3 for derivations of $\phi_{res1}$ and $\phi_{res2}$.*

The canonical form $C_2$ is preferable to $C_1$, since it enables the selection of $k_i$ to eradicate $\phi_{res2}$. A convenient solution is to ensure that: $k_1 = k_4$ and $k_2 = k_3$ such that $\mathbb{E}[\phi(k_1)] = \mathbb{E}[\phi(k_4)]$ and $\mathbb{E}[\phi(k_2)] = \mathbb{E}[\phi(k_3)]$, resulting in $\phi_{res2} = 0$. We choose the solution: $k_1 = k_4 = S_5$, and $k_2 = k_3 = S_6$; where $S_5$ are subsequent states to $S_1$, and $S_6$ are subsequent states to $S_2$. This leads to the following SRRD definition:

**Definition 1** (Sparsity Resilient Canonically Shaped Reward). *Let $R : \mathcal{S} \times \mathcal{A} \times \mathcal{S} \to \mathbb{R}$ be a reward function. Given distributions $\mathcal{D}_{\mathcal{S}} \in \Delta\mathcal{S}$ and $\mathcal{D}_{\mathcal{A}} \in \Delta\mathcal{A}$ over states and actions, let $S_3$ be the set of states sampled according to $\mathcal{D}_{\mathcal{S}}$, and let $A$ be the set of actions sampled according to $\mathcal{D}_{\mathcal{A}}$. Furthermore, let $T(S_4|S_3, A)$ be a transition model governing the conditional distribution over next states, where, $S_4$ are subsequent states to $S_3$. For each $s \in S_3$ and $s' \in S_4$, let $S_1$ be the set of states sampled according to $T(S_1|s', A)$, and $\tilde{S}_2$ be the set of states sampled according to $T(\tilde{S}_2|s, A)$. Let $S_2$ represent non-terminal states in $\tilde{S}_2$. Similarly, let $S_5$ and $S_6$ be set of states sampled according to $T(S_5|S_1, A)$ and $T(S_6|S_2, A)$, respectively. The Sparsity Resilient Canonically Shaped Reward is defined as:*

$$
\begin{aligned}
C_{SRRD}(R)(s,a,s') = R(s, a, s') &+ \mathbb{E}[\gamma R(s', A, S_1) - R(s, A, S_2) - \gamma R(S_3, A, S_4) \\
&+ \gamma^2 R(S_1, A, S_5) - \gamma R(S_2, A, S_6) + \gamma R(S_3, A, S_6) - \gamma^2 R(S_4, A, S_5)].
\end{aligned}
\tag{11}
$$

Note that in $C_{SRRD}$, we first sample $\tilde{S}_2$ as subsequent states from $s$, and then derive $S_2$ as non-terminal states in $\tilde{S}_2$. This modification ensures that $(S_2, A, S_6) \subseteq (S_3, A, S_6)$, which is crucial for SRRD's robustness in Theorem 1. In practice though, the difference between $\hat{S}_2$ and $S_2$ is generally minimal, especially in long-horizon problems where terminal states tend to be fewer compared to non-terminal states. The SRRD canonicalization function is invariant to potential shaping as described by Proposition 1.

**Proposition 1.** *(The Sparsity Resilient Canonically Shaped Reward is Invariant to Shaping) Let $R : \mathcal{S} \times \mathcal{A} \times \mathcal{S} \to \mathbb{R}$ be a reward function and $\phi : \mathcal{S} \to \mathbb{R}$ be a state potential function. Applying $C_{SRRD}$ to a potentially shaped reward $R'(s, a, s') = R(s, a, s') + \gamma\phi(s') - \phi(s)$ satisfies: $C_{SRRD}(R) = C_{SRRD}(R')$.*

*Proof.* See Appendix A.4.

Given reward functions $R_A$ and $R_B$, the SRRD pseudometric is computed as:

$$
D_{\mathrm{SRRD}}(R_A, R_B) = D_\rho(C_{SRRD}(R_A), C_{SRRD}(R_B)),
\tag{12}
$$

where $D_\rho$ is the Pearson distance. For robustness, we establish an upper bound on regret showing that as $D_{\mathrm{SRRD}} \to 0$, the performance difference between the policies induced by the reward functions under comparison, $R_A$ and $R_B$, approaches 0, as described by Theorem 2. Depending on the transition model dictating the composition of $\{S_1, ..., S_6\}$, $C_{SRRD}$ can be invariant to shaping across various transition distributions without requiring full coverage, provided that, for each set of transitions $(S_i, A, S_j)$ in the reward expectation terms, all transitions in the cross-product $S_i \times A \times S_j$ have defined reward values. This is guaranteed when the full reward function $R$ is available.

In practical scenarios, the full reward function may be unavailable, and $C_{SRRD}$ can be approximated from reward samples, resulting in $\hat{C}_{SRRD}$ (see Definition 8). In these settings, the transitions needed for each reward expectation term in $\hat{C}_{SRRD}$, might be unsampled due to transition sparsity. Despite this challenge, approximations for $\hat{C}_{SRRD}$ are robust due to the strategic choices of $k_i$ (see Equation 9 and 11), $S_5$ and $S_6$, which ensure that the approximations are ideal for the following two reasons: First, for any reward sample, we can compute reliable expectation estimates for the first six terms, since for each set of transitions $(S_i, A, S_j)$, $S_j$ is distributed conditionally based on $(S_i, A)$; hence, these transitions naturally align with the transition dynamics that dictate the nature of the reward sample. However, for transitions in the last two terms, $(S_3, A, S_6)$ and $(S_4, A, S_5)$, $S_6$ is not distributed conditionally based on $(S_3, A)$ but on $(S_2, A)$, and $S_5$ is not distributed conditionally based on $(S_4, A)$ but on $(S_1, A)$; hence these transitions may not align well with the transition dynamics that dictate the structure of the reward sample, making these transitions highly susceptible to being unsampled. Second, while there might be significant fractions of unsampled transitions in $(S_4, A, S_5)$, and $(S_3, A, S_6)$, a minimal set of sampled transitions are likely to exist, because:

- **Transitions** $(S_1, A, S_5) \subseteq (S_4, A, S_5)$:
  Since $S_1$ are subsequent states to $s'$, and $S_4$ are subsequent states for all sampled transitions. It follows that, $S_1 \subseteq S_4$ (see example in Appendix A.6), hence, $(S_1, A, S_5) \subseteq (S_4, A, S_5)$.

- **Transitions** $(S_2, A, S_6) \subseteq (S_3, A, S_6)$.
  $S_2$ is the set of non-terminal subsequent states from $s$. Since $S_3$ encompasses all initial non-terminal states from all sampled transitions, it follows that: $S_2 \subseteq S_3$, hence, $(S_2, A, S_6) \subseteq (S_3, A, S_6)$.

Therefore, we can get decent fractions of sampled transitions in $(S_3, A, S_6)$ and $(S_4, A, S_5)$, which typically reduces the extent and impact of unsampled transitions in SRRD, making it robust under transition sparsity, as described in Section 4.1, Theorem 1.

In summary, SRRD is designed to improve reward function comparisons under transition sparsity. To achieve this, SRRD introduces additional reward expectation terms into canonicalization, to ensure that rewards are standardized (remove shaping) based on the observed distribution of sampled transitions, without assuming full transition coverage. Regarding computational complexity, when employing the double-batch sampling method that involves a batch $B_V$ of $N_V$ transitions, and another batch $B_M$ of $N_M$ state-action pairs for canonicalization (refer to Appendix A.1); EPIC has a complexity of $O(\max(N_V N_M, N_M^2))$, and both DARD and SRRD have complexities of $O(N_V N_M^2)$ (see Appendix A.11). This paper adopts the double-batch sampling approach to maintain consistency with prior works, however, alternative sample-based approximations are also viable. Specifically, Appendix B.2 explores a sample-based approximation method that employs unbiased estimates, and Appendix B.3, explores the use of regression to infer reward values for unsampled transitions. Across all these approximation variants, SRRD consistently outperforms both DARD and EPIC under conditions of transition sparsity, confirming its robustness.

## 4.1 Relative Shaping Errors

This section provides a theoretical evaluation on the robustness of the sample-based approximations: $\hat{C}_{SRRD}$, $\hat{C}_{DARD}$, and $\hat{C}_{EPIC}$, to unsampled transitions. We first discuss the relevant definitions and assumptions for the analysis, then present Theorem 1 comparing the three methods. For a reward function $R$, the structure of an arbitrary reward canonicalization method (such as $C_{SRRD}$, $C_{DARD}$, and $C_{EPIC}$) takes the form:

$$C_S(R)(s, a, s') = R(s, a, s') + \mathbb{E} \sum_{i=1}^{n-1} [\alpha_i R(S_i, A, S_i')], \tag{13}$$

where, $\alpha_i$ is a constant, $|\alpha_i| \leq 1$, and $i \in \{1, ..., n-1\}$ are indices denoting the state subsets, $S_i, S_i' \subseteq \mathcal{S}$. The sample-based approximation for $C_S$ is denoted by $\hat{C}_S$. Given a non-zero reward sample $R$, we can bound the maximum range of each canonical reward by defining the upper bound canonical reward as follows:

**Definition 2.** *(Upper Bound Canonical Reward) Let $R$ be a non-zero reward sample defined over a set of transitions $B$, and let $\hat{C}_S$ be an arbitrary sample-based canonicalization method. Furthermore, let $Z = \max_{(s,a,s') \in B}(|R(s, a, s')|)$ be the maximum absolute reward. The upper bound canonical reward is given by:*

$$U(\hat{C}_S(R)(s, a, s')) = nZ \tag{14}$$

The justification for Definition 2 is that $C_S(R)(s, a, s')$ has $n$-terms with absolute values bounded by $Z$ (both $|R(s, a, s')| \leq Z$ and $|\alpha_i \mathbb{E}[R(S_i, A, S_i')]| \leq Z$), hence, $U(\hat{C}_S(R)(s, a, s')) = nZ$. The non-zero assumption on $R$ guarantees that there exists at least one sampled transition with a non-zero reward, ensuring that $U(\hat{C}_S(R)(s, a, s')) \geq 0$. To quantify the performance bounds of the canonicalization methods, we define the relative shaping error as follows:

**Definition 3.** *(Relative Shaping Error (RSE)) Let $R'$ be a shaped, non-zero reward sample defined over a set of transitions $B$, and let $\hat{C}_S$ be a sample-based reward canonicalization method, such that: $\hat{C}_S(R') = \hat{C}_S(R) + \phi_R$, where $\phi_R$ is the residual shaping term. Suppose that $\phi_R$ can be expressed as: $\phi_R = \phi_{\tilde{R}} + K_\phi$, where $K_\phi$ is a constant that does not vary with $(s, a, s')$, and $\phi_{\tilde{R}}$ is the effective residual shaping term. Furthermore, let $U(\hat{C}_S(R)(s, a, s')) = nZ$ represent the upper bound of the unshaped canonical reward, where, $Z = \max_{(s,a,s') \in B}(|R(s, a, s')|)$. The relative shaping error is defined as:*

$$RSE(\hat{C}_S(R)(s, a, s')) = \frac{|\phi_{\tilde{R}}(s, a, s')|}{U(\hat{C}_S(R)(s, a, s'))} = \frac{|\phi_{\tilde{R}}(s, a, s')|}{nZ}. \tag{15}$$

The relative shaping error (RSE) is designed to theoretically quantify the impact of residual shaping in reward distance comparisons. The denominator, $U(\hat{C}_S(R)(s, a, s')) = nZ$, represents an upper bound on the

magnitude of the base (unshaped) canonical reward, and it serves to normalize the impact of shaping. A low RSE suggests that $U(\hat{C}_S(R)(s,a,s'))$ is substantially larger than $|\phi_{\tilde{R}}(s,a,s')|$, indicating that the impact of shaping is likely minimal. Conversely, a high RSE implies that $U(\hat{C}_S(R)(s,a,s'))$ is relatively small compared to $|\phi_{\tilde{R}}(s,a,s')|$, highlighting a more likely significant influence of the effective residual shaping. Note that in the RSE definition, the effective residual shaping term is $\phi_{\tilde{R}} = \phi_R - K_\phi$, where $K_\phi$ is a constant that does not impact the Pearson distance (since its shift invariant), and hence, the reward distances. For a comprehensive discussion on the derivation of the RSE definition, refer to Appendix A.2.1.

With regard to reward samples, we define forward and non-forward transitions as follows:

**Definition 4** (Forward transitions). *Given a reward sample that spans a state space $S$, and an action space $A$. Consider the state subsets $S_i, S_j \subseteq S$. Transitions $(S_i, A, S_j)$ are forward transitions if $S_j$ is distributed conditionally based on $(S_i, A)$, according to the underlying transition dynamics of the reward sample.*

**Definition 5** (Non-forward transitions). *Given a reward sample that spans a state space $S$, and an action space $A$. Consider the state subsets $S_i, S_j, S_k \subseteq S$. Transitions $(S_i, A, S_j)$ are non-forward transitions when the states in $S_j$ are not distributed conditionally based on $(S_i, A)$, but are instead based on $(S_k, A)$, according to the underlying transition dynamics of the reward sample.*

In reward canonicalization methods, given transitions $(S_i, A, S_j)$ needed in computing reward expectations, both forward and non-forward transitions can have unsampled transitions since canonicalization methods typically require the cross-product of all transitions from $S_i$ to $S_j$. However, forward transitions are generally more robust to being unsampled since they are usually in-distribution with the underlying transition dynamics governing the reward samples, hence, they naturally align with the progression of rewards in the sample. In contrast, non-forward transitions are highly prone to being unsampled (and also unrealizable) as they may include a significant fraction of transitions that are out-of-distribution with the reward sample's transition dynamics. Based on the rationale that forward transitions are more robust to being unsampled compared to non-forward transitions, to make our analysis more tractable, we will assume that the fraction of unsampled transitions in forward transitions is negligible, leading to Theorem 1:

**Theorem 1.** *Consider the reward comparison task on two equivalent non-zero reward samples that differ due to potential-based shaping, and share the same set of sampled transitions, $B$. During canonicalization, consider the forward and non-forward transition sets needed to compute the reward expectation terms, and assume that the fraction of unsampled forward transitions in both reward samples is negligible. Each reward sample can be expressed as $R'_i(s,a,s') = R(s,a,s') + \gamma\phi_i(s') - \phi_i(s)$ for $i \in \{1,2\}$, where, $R$ is the unshaped reward sample, $\gamma$ is a discount factor, and $\phi_i(s)$ is the potential shaping function for $R'_i$. Under transition sparsity, the upper bound of the Relative Shaping Error (RSE) for $\hat{C}_{SRRD}$ is lower than that of $\hat{C}_{DARD}$ and $\hat{C}_{EPIC}$ respectively, in the order:*

$$RSE(\hat{C}_{SRRD}) \leq \frac{M}{3Z}; \quad RSE(\hat{C}_{DARD}) \leq \frac{2M}{3Z}; \ RSE(\hat{C}_{EPIC}) \leq \frac{M}{Z},$$

*where, $M = \max_{s \in S}(|\phi_i(s)|)$ is the maximum magnitude of potential shaping across all states for the reward samples, and $Z = \max_{(s,a,s') \in B}(|R(s,a,s')|)$ is the maximum absolute value of the unshaped reward sample.*

*Proof.* See Appendix A.2 □

In more precise terms, Theorem 1 evaluates the robustness $\hat{C}_{SRRD}$, $\hat{C}_{DARD}$, and $\hat{C}_{EPIC}$ against residual shaping introduced by unsampled transitions from non-forward transitions, which are more likely to be out-of-distribution with the transition dynamics that generated the reward samples. The theorem assesses the upper bounds of the RSE across the three canonicalization methods to determine their sensitivity to residual shaping, assuming that the reward samples under comparison induce the same optimal policies, and share the same set of transitions. These assumptions ensure that we isolate the differences between the canonicalized reward samples to variations in residual shaping. A lower upper bound implies that a canonicalization method is likely to be less sensitive to the effects of residual shaping and, hence, it is more likely to reveal the actual similarity between the unshaped reward samples under comparison. As shown by the upper bounds of the relative shaping errors, the approximation for $\hat{C}_{SRRD}$ is theoretically more robust, compared to those of $\hat{C}_{EPIC}$ and $\hat{C}_{DARD}$, since it has the smallest upper bound.

## 5 Experiments

To empirically evaluate SRRD, we examine the following hypotheses:

**H1:** SRRD is a reliable reward comparison pseudometric under high transition sparsity.

**H2:** SRRD can enhance the task of classifying agent behaviors based on their reward functions.

In these hypotheses, we compare the performance of SRRD to both EPIC and DARD using sample-based approximations (see Appendix A.1). In **H1**, we analyze SRRD's robustness under transition sparsity resulting from limited sampling and feasibility constraints. In **H2**, we investigate a practical use case to classify agent behaviors using their reward functions. Experiment 1 tests **H1** and Experiment 2 tests **H2**.

**Domain Specifications**   To conduct Experiment 1, we need the capability to vary the number of sampled transitions, since the goal is to test SRRD's performance under different levels of transition sparsity. Therefore, Experiment 1 is performed in the Gridworld and Bouncing Balls domains, as they provide the flexibility for parameter variation to control the size of the state and action spaces[2]. These two domains have also been studied in the EPIC and DARD papers, respectively. The Gridworld domain simulates agent movement from a given initial state to a specified terminal state under a static policy, within 200 timesteps. States are defined by $(x, y)$ coordinates where $0 \leq x < N$ and $0 \leq y < M$ implying $|\mathcal{S}| = NM$. The action space consists of four cardinal directions (single steps), and the environment is stochastic, with a probability $\epsilon$ of transitioning to any random state irrespective of the selected action. When $\epsilon = 0$, a feasibility constraint is imposed, preventing the agent from making random transitions. The Bouncing Balls domain, adapted from Wulfe et al. (2022), simulates a ball's motion from a starting state to a target state while avoiding randomly mobile obstacles. These obstacles add complexity to the environment since the ball might need to change its strategy to avoid obstacles (at a distance, $d = 3$). Each state is defined by the tuple $(x, y, d)$, where $(x, y)$ indicates the ball's current location, and $d$ indicates the ball's Euclidean distance to the nearest obstacle, such that: $0 \leq x < N$ and $0 \leq y < M$. The action space includes eight directions (cardinals and ordinals), and we also define the stochasticity-level parameter $\epsilon$ for choosing random transitions.

The objective for Experiment 2 is to test SRRD's performance in diverse and near-realistic domain settings, where we have no control over factors such as the nature of rewards and the level of transition sparsity. Therefore, in addition to the Gridworld and the Bouncing Balls domains used in Experiment 1 (but with fixed parameters), we extend our evaluation to the following testbeds: Drone Combat - a battlefield environment between two swarms, adapted from a predator-prey gym environment (Anurag, 2019), Montezuma's Revenge - an Atari benchmark dataset with human demonstrations for the Montezuma's Revenge game (Kurin et al., 2017), StarCraft II - a simulation of combat scenarios where a controlled multiagent team aims to defeat a default AI enemy team (Vinyals et al., 2019), Robomimic - an open source dataset of robotics manipulation tasks incorporating both human and simulated demonstrations (Mandlekar et al., 2021), and MIMIC-IV - a real-world de-identified electronic health dataset for patients admitted at an emergency or intensive care unit at Beth Israel Deaconess Medical Center in Boston, MA (Johnson et al., 2023). These domains resemble complex scenarios with large state and action spaces, enabling us to test SRRD's (as well as the other pseudometrics) generalization to near-realistic scenarios. Further details about these domains, including information about the state and action features are described in Appendix C.5.

**Reward Functions**   Extrinsic reward functions are manually defined using a combination of state and action features. For the Drone Combat, Montezuma's Revenge, and StarCraft II domains, we use the default game engine scores as the reward function, and for Robomimic, rewards are based on task completion (see Appendix C.2). For the Gridworld and Bouncing Balls domains, in each reward function, the reward values are derived from the decomposition of state and action features, where, $(s_{f1}, ..., s_{fn})$ is from the starting state $s$; $(a_{f1}, ..., a_{fm})$ is from the action $a$; and $(s'_{f1}, ..., s'_{fn})$ is from the subsequent state $s'$. For the Gridworld domain, these features are the $(x, y)$ coordinates, and for the Bouncing Balls domain, these include $(x, y, d)$, where $d$ is the distance of the obstacle nearest to the ball. For each unique transition, using randomly

---

[2]Experiment 1 excludes the Drone Combat, MIMIC-IV, Robomimic, Montezuma's Revenge, and StarCraft II, since these domains have very large state and action spaces that hinder effective coverage computation.

generated constants: $\{u_1, ..., u_n\}$ for incoming state features; $\{w_1, ..., w_m\}$ for action features; $\{v_1, ...v_n\}$ for subsequent state features, we create polynomial and random rewards as follows:

$$\text{Polynomial:} \quad R(s, a, s') = u_1 s_{f1}^{\alpha} + \ldots + u_n s_{fn}^{\alpha} + w_1 a_{f1}^{\alpha} + \ldots + w_m a_{fm}^{\alpha} + v_1 s_{f1}'^{\alpha} + \ldots + v_n s_{fn}'^{\alpha},$$

where $\alpha$ is randomly generated from 1-10, denoting the degree of the polynomial.

$$\text{Random:} \quad R(s, a, s') = \beta,$$

where $\beta$ is a randomly generated reward for each unique transition.

For the polynomial rewards, $\alpha$ is the same across the entire sample, but other constants vary between different transitions. The same reward relationships are used to model potential shaping functions. In addition, we also explore linear and sinusoidal reward models (see Appendix C.2). For complex environments such as StarCraft II and MIMIC-IV, specifying reward functions can be challenging, hence we also incorporate IRL to infer rewards from demonstrated behavior. For IRL, we consider the following methods: Maximum Entropy IRL (Maxent) (Ziebart et al., 2008); Adversarial IRL (AIRL) (Fu et al., 2018); and Preferential-Trajectory IRL (PTIRL) (Santos et al., 2021). The algorithms are summarized in Appendix C.7.

### 5.1 Experiment 1: Transition Sparsity

**Objective:** The goal of this experiment is to test SRRD's ability to identify similar reward samples under transition sparsity as a result of limited sampling and feasibility constraints.

**Relevance:** The EPIC and DARD pseudometrics struggle in conditions of high transition sparsity since they are designed to compare reward functions under high coverage. SRRD is developed to be resilient under transition sparsity and this experiment tests SRRD's performance relative to both EPIC and DARD.

**Approach:** This experiment is conducted on a $20 \times 20$ Gridworld domain and a $20 \times 20$ Bouncing Balls domain. For all simulations, manual rewards are used since they enable the flexibility to vary the nature of the relationship between reward values and features, enabling us to test the performance of the pseudometrics on diverse reward values, which include polynomial and random reward relationships. We also vary the shaping potentials such that $|R(s, a, s')| \le |\gamma\phi(s') - \phi(s)| \le 5|R(s, a, s')|$. For each domain, a ground truth reward function ($GT$) and an equivalent potentially shaped reward function ($SH$) are generated, both with full coverage (100%). Using rollouts from a uniform policy, rewards $R$ and $R'$ are sampled from $GT$ and $SH$ respectively, and these might differ in transition composition. After sample generation, $R$ and $R'$ are canonicalized and reward distances are computed using common transitions between the reward samples, under varying levels of coverage due to limited sampling and feasibility constraints. The SRRD, DARD, and EPIC reward distances are computed, as well as DIRECT, which is the Pearson distance between the reward samples, without canonicalization. Since $R$ and $R'$ are drawn from equivalent reward functions, an accurate pseudometric should yield distances close to the minimum Pearson distance, $D_\rho = 0$; and the least accurate should yield a distance close to the maximum, $D_\rho = 1$. DIRECT serves as a worst-case performance baseline, since it computes reward distances without canonicalization (needed to remove shaping). We perform 200 simulation trials for each comparison task and record the mean reward distances.

**Simulations and Results: Limited Sampling:** Using rollouts from a uniform policy, we sample $R$ and $R'$ from $GT$ and $SH$, respectively, under a stochasticity-level parameter, $\epsilon = 0.1$. The number of transitions sampled is controlled by varying the number of policy rollouts from 1 to 2000. The corresponding coverage is computed as the number of sampled transitions over the number of all theoretically possible transitions ($= |\mathcal{S} \times \mathcal{A} \times \mathcal{S}|$). Figure 3a summarizes the variation of reward distances to transition coverage across different levels of transition sampling in the Gridworld and Bouncing Balls domains. As shown, SRRD outperforms other baselines as it converges towards $D_\rho = 0$ faster, even when coverage is low. DARD generally outperforms EPIC, however, it is highly prone to shaping compared to SRRD since it is more sensitive to unsampled transitions. All pseudometrics generally outperform DIRECT, illustrating the value of removing shaping via canonicalization. No significant differences are observed in the general trends of results between the two domains, and additional simulations are presented in Appendix C.4. In conclusion, the proposed SRRD consistently outperforms both EPIC and DARD under limited sampling.

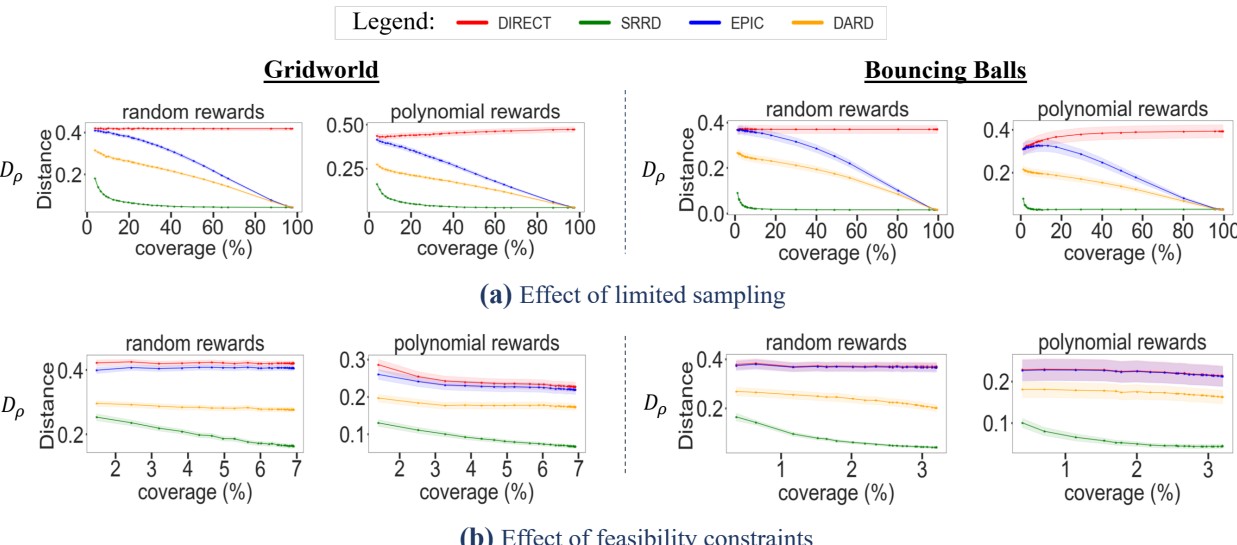

**(a)** Effect of limited sampling

**(b)** Effect of feasibility constraints

Figure 3: (Transition Sparsity). The figure illustrates the performance of reward comparison pseudometrics in identifying the similarity between potentially shaped reward functions under two conditions: (a) limited sampling and (b) feasibility constraints. A more accurate pseudometric yields a Pearson distance $D_\rho$ close to 0, indicating a high degree of similarity between shaped reward functions, while a less accurate pseudometric results in $D_\rho$ close to 1. In both experiments, transition coverage is calculated as the ratio of sampled transitions to the set of all theoretically possible transitions $|S \times A \times S|$, including both feasible and unfeasible transitions. Each coverage data point represents an average over 200 simulations at a constant policy rollout count, with coverage data points generated by varying the number of policy rollouts from 1 to 2000 (see Appendix C.3). In panel (a), **EPIC** and **DARD** lag behind **SRRD** at low transition coverage due to limited sampling, but their performance gradually improves as coverage increases with higher rollout counts. In panel (b), movement restrictions significantly reduce transition coverage, regardless of rollout sampling frequency, which negatively impacts **EPIC's** performance (almost similar to **DIRECT**).

**Simulations and Results: Feasibility Constraints:** Using rollouts (ranging from 1 to 2000) from a uniform policy, we sample $R$ and $R'$ from $GT$ and $SH$, respectively. To impose feasibility constraints, we set the stochasticity-level parameter, $\epsilon = 0$, restricting random transitions between states such that only movement to adjacent states is permitted. These movement restrictions ensure that coverage is generally low ($< 10\%$), even though the number of rollouts is similar to those in the first experiment. Figure 3b summarizes the results for the variation of reward distances to transition coverage under the movement restrictions. As shown, SRRD significantly outperforms all the baselines indicating its high robustness.

## 5.2 Experiment 2: Classifying Agent Behaviors

**Objective:** The goal of this experiment is to assess SRRD's effectiveness as a distance measure in classifying agent behaviors based on reward functions. If SRRD is robust, it should identify similarities among reward functions from the same agents while differentiating reward functions from distinct agents.

**Relevance:** This experiment demonstrates how reward comparison pseudometrics can be used to interpret reward functions by relating them to agent behavior. In many real-world situations, samples of agent behaviors are available, and there is a need to interpret the characteristics of the agents that produced these behaviors. For example, several works have attempted to predict player rankings and strategies using past game histories (Luo et al., 2020; Yanai et al., 2022). This experiment takes a similar direction by attempting to classify the identities of agents from their unlabeled past trajectories using reward functions. The reliance on reward functions rather than the original trajectories is based on the premise that reward functions are "succinct" and "robust", hence a preferable means to interpret agent behavior (Abbeel & Ng, 2004).

**Approach:** In this experiment, we train a $k$-nearest neighbors ($k$-NN) classifier to classify unlabeled agent trajectories by indirectly using computed rewards, to identify the agents that produced these trajectories. We examine the $k$-NN algorithm since it is one of the most popular distance-based classification techniques. The experiment is conducted across all domains, and since we want to maximize classification accuracy, we consider different IRL rewards, including: Maxent, AIRL, PTIRL as well as manual (extrinsic) rewards. For manual rewards: we utilize the default game score for the Drone Combat, StarCraft II, and Montezuma's Revenge domains; environmental sparse rewards for task completion in the Robomimic domain; and feature-based (for example, polynomial) rewards for the Gridworld, Bouncing Balls and MIMIC-IV domains, where we induce random potential shaping. For each domain, we examine SRRD, DIRECT, EPIC, and DARD as distance measures for a $k$-NN reward classification task. The steps for the approach are as follows:

1. Create agents $X = \{x_1, ..., x_m\}$ with distinct behaviors.

2. For each agent $x_i \in X$, generate a collection of sets $\{\zeta_1^{x_i}, \ldots, \zeta_p^{x_i}\}$, where each $\zeta_j^{x_i} = \{\tau_{j,1}^{x_i}, \ldots, \tau_{j,q}^{x_i}\}$ is a $q$-sized set of trajectories. Compute reward functions $\{R_1^{x_i}, \ldots, R_p^{x_i}\}$ using IRL or manual specification based on each corresponding $\zeta_j^{x_i}$.

3. Randomly shuffle all the computed reward functions $R$ (from all agents), and split into the training $R_{train}$ and testing $R_{test}$ sets.

4. Train a $k$-NN classifier using each pseudometric (as distance measure) on $R_{train}$ and test it on $R_{test}$.

Table 1: The accuracy (%) of different reward comparison distances in $k$-NN reward classification.

| DOMAIN | REWARDS | DIRECT | EPIC | DARD | SRRD |
|--------|---------|--------|------|------|------|
| Gridworld | Manual | 69.8 | 69.3 | 70.0 | **75.8** |
| | Maxent | 57.4 | 57.5 | 68.9 | **70.0** |
| | AIRL | 82.3 | 84.9 | 85.0 | **86.2** |
| | PTIRL | 82.2 | 84.2 | 83.4 | **86.0** |
| Bouncing Balls | Manual | 46.5 | 47.3 | 52.0 | **55.2** |
| | Maxent | 39.7 | 46.0 | **50.8** | 49.9 |
| | AIRL | 41.2 | 46.1 | 49.8 | **56.3** |
| | PTIRL | 70.3 | 71.1 | 69.5 | **72.4** |
| Drone Combat | Manual | 67.1 | 67.2 | 66.2 | **73.9** |
| | Maxent | 70.3 | **77.7** | 73.2 | 76.7 |
| | AIRL | 90.1 | 90.7 | 92.3 | **93.8** |
| | PTIRL | 52.5 | 63.7 | 65.1 | **78.3** |
| StarCraft II | Manual | 65.5 | 67.4 | 69.5 | **76.5** |
| | Maxent | 72.3 | 74.1 | 73.9 | **74.8** |
| | AIRL | 75.1 | 75.3 | **78.1** | 77.0 |
| | PTIRL | 77.2 | 78.1 | 77.6 | **79.8** |
| Montezuma's Revenge | Manual | 66.4 | 70.1 | 68.3 | **73.5** |
| | Maxent | 67.8 | 69.1 | 68.7 | **71.2** |
| | AIRL | 68.2 | 71.4 | 69.8 | **72.3** |
| | PTIRL | 68.2 | 69.6 | **70.6** | 70.2 |
| Robomimic | Manual | 78.2 | 80.3 | 79.5 | **82.4** |
| | Maxent | 82.3 | 86.8 | 79.5 | **89.8** |
| | AIRL | 85.9 | 87.1 | 86.3 | **91.8** |
| | PTIRL | 80.3 | 83.6 | 83.1 | **84.2** |
| MIMIC-IV | Manual | 53.5 | 56.5 | 57.3 | **59.2** |
| | Maxent | 57.8 | 59.1 | 53.9 | **60.2** |
| | AIRL | 56.5 | 60.7 | 57.6 | **63.3** |
| | PTIRL | 58.9 | **61.4** | 60.3 | 60.9 |

Across all domains, in step 1 and 2, fixed parameters are defined such that: $m$ - is the number of distinct agent policies; $p$ - is the number of trajectory sets per agent; and $q$ - is the number of trajectories per set in each IRL run (refer to Appendix C.6). In step 1, different agent behaviors are controlled by varying the agents' policies. In step 4, to train the classifier, grid-search is used to identify candidate values for $k$ and $\gamma$, and twofold cross-validation (using $R_{train}$) is used to optimize hyper-parameters based on accuracy. Since we assume potential shaping, $\gamma$ is a hyperparameter as its value is unknown beforehand. To classify a reward function $R_i \in R_{test}$, we traverse reward functions $R_j \in R_{train}$, and compute the distance, $D_\rho(R_i, R_j)$ using the reward pseudometrics. We then identify the top $k$-closest rewards to $R_i$, and choose the label of the most frequent class. We select a training to test set ratio of $70 : 30$, and repeat this experiment 200 times.

**Simulations and Results:** Table 1 summarizes experimental results. As shown, SRRD generally achieves higher accuracy compared to DIRECT, EPIC and DARD across all domains, indicating SRRD effectiveness at discerning similarities between rewards produced by the same agents, and differences between those generated by different agents. This trend is more pronounced with manual rewards where SRRD significantly outperforms other baselines. This can be attributed to potential shaping, which is intentionally induced in manual rewards that SRRD is specialized to tackle. Therefore, SRRD proves to be a more effective distance measure at classifying rewards subjected to potential shaping. For IRL-based rewards such as Maxent, AIRL, and PTIRL, while we assume potential shaping, non-potential shaping could be present. This explains the reduction in SRRD's performance gap over EPIC and DARD, as well as the few instances where EPIC and DARD outperform SRRD, though SRRD is still generally dominant. We also observe that all the pseudometrics tend to perform better on AIRL rewards compared to other IRL-based rewards. This result is likely due to the formulation of the AIRL algorithm, which is designed to effectively mitigate the effects of unwanted shaping in reward approximation (Fu et al., 2018), thus providing more consistent rewards. Overall, SRRD, EPIC, and DARD outperform DIRECT, emphasizing the importance of canonicalization at reducing the impact of shaping.

To verify the validity of results, Welch's t-tests for unequal variances are conducted across all domain and reward type combinations, to test the null hypotheses: (1) $\mu_{\text{SRRD}} \leq \mu_{\text{DIRECT}}$, (2) $\mu_{\text{SRRD}} \leq \mu_{\text{EPIC}}$, and (3) $\mu_{\text{SRRD}} \leq \mu_{\text{DARD}}$; against the alternative: (1) $\mu_{\text{SRRD}} > \mu_{\text{DIRECT}}$, (2) $\mu_{\text{SRRD}} > \mu_{\text{EPIC}}$, and (3) $\mu_{\text{SRRD}} > \mu_{\text{DARD}}$, where $\mu$ represents the sample mean. We reject the null when the p-value $< 0.05$ (level of significance), and conclude that: (1) $\mu_{\text{SRRD}} > \mu_{\text{DIRECT}}$ for all instances; (2) $\mu_{\text{SRRD}} > \mu_{\text{EPIC}}$ for 22 out of 28 instances, and (3) $\mu_{\text{SRRD}} > \mu_{\text{DARD}}$ for 24 out of 28 instances. These tests are performed assuming normality as per central limit theorem, since the number of trials is 200. For additional details about the tests, refer to Appendix C.8. In summary, we conclude that SRRD is a more effective distance measure for classifying reward samples compared to its baselines.

# 6 Conclusion and Future Work

This paper introduces SRRD, a reward comparison pseudometric designed to address transition sparsity, a significant challenge encountered when comparing reward functions without high transition coverage. Conducted experiments, and theoretical analysis, demonstrate SRRD's superiority over state-of-the-art pseudometrics, such as EPIC and DARD, under limited sampling and feasibility constraints. Additionally, SRRD proves effective as a distance measure for $k$-NN classification using reward functions to represent agent behavior. This implies that SRRD can find higher similarities between reward functions generated by the same agent and higher differences between reward functions that are generated from different agents.

Most existing studies, including ours, have primarily focused on potential shaping, as it is the only additive shaping technique that guarantees policy invariance (Ng et al., 1999; Jenner et al., 2022). Future research should consider the effects of non-potential shaping on SRRD (see Appendix B.1) or random perturbations, as these might distort reward functions that would otherwise be similar. This could help to standardize and preprocess a wider range of rewards that might not necessarily be potentially shaped. Future studies should also explore applications of reward distance comparisons in scaling reward evaluations in IRL algorithms. For example, iterative IRL approaches such as MaxentIRL, often perform policy learning to assess the quality of the updated reward in each training trial. Integrating direct reward comparison pseudometrics to determine if rewards are converging, could help to skip the policy learning steps, thereby speeding up IRL. Finally, the

development of reward comparison metrics has primarily aimed to satisfy policy invariance. A promising area to examine in the future is multicriteria policy invariance, where invariance might be conditioned to different criteria. For example, in the context of reward functions in Large Language Models (LLMs), it might be important to compute reward distance pseudometrics that consider different criteria such as bias, safety, or reasoning, to advance interpretability, which could be beneficial for applications such as reward fine-tuning and evaluation (Lambert et al., 2024).

## Acknowledgments

This research was funded in part by the Air Force Office of Scientific Research Grant No. FA9550-20-1-0032, the Office of Naval Research Grant No. N00014-19-1-2211, and Fulbright-CAPES, Grant No. 88881.625406/2021-01. We thank the anonymous reviewers for their valuable suggestions, which helped to improve the paper.

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

## A Derivations, Theorems and Proofs

### A.1 Sample-Based Approximations

While the EPIC, DARD, and SRRD pseudometrics can be computed exactly in small environments, doing so in environments with large or infinite state-action spaces is generally infeasible because it requires evaluating rewards for all possible transitions. To address this computational barrier, we adopt sample-based approximations for scalability. Specifically, we employ the **double-batch sampling** technique introduced by Gleave et al. (2021) to estimate $C_{EPIC}$ and was later applied by Wulfe et al. (2022) to approximate $C_{DARD}$. This technique uses two separate batches of samples: $B_V$, a set of $N_V$ transitions; and $B_M$, a set of $N_M$ state-action pairs that are sampled from the joint state-action space. To ensure consistency across the three pseudometrics, we extend the double-batch sampling approach to estimate $C_{SRRD}$. The resulting sample-based approximations used in this paper are defined as follows:

**Definition 6.** *(Sample-based EPIC) Given a batch $B_V$ of $N_V$ samples from the coverage distribution $\mathcal{D}$, and a batch $B_M$ of $N_M$ samples from the joint state and action distributions, $\mathcal{D}_S \times \mathcal{D}_A$. For each $(s,a,s') \in B_V$, the canonically shaped reward is approximated by taking the mean over $B_M$:*

$$\hat{C}_{EPIC}(R)(s,a,s') = R(s,a,s') + \frac{\gamma}{N_M} \sum_{(x,u) \in B_M} R(s',u,x) - \frac{1}{N_M} \sum_{(x,u) \in B_M} R(s,u,x)$$
$$- \frac{\gamma}{N_M^2} \sum_{(x,\cdot) \in B_M} \sum_{(x',u) \in B_M} R(x,u,x') \tag{16}$$

**Definition 7.** *(Sample-based DARD) Given a batch $B_V$ of $N_V$ samples from the coverage distribution $\mathcal{D}$, and a batch $B_M$ of $N_M$ samples from the joint state and action distributions, $\mathcal{D}_S \times \mathcal{D}_A$. For each $(s,a,s')$ transition in $B_V$, we derive sets $X', X'' \subseteq B_M$, where, $|X'| = N'$ and $|X''| = N''$. Here, for the set $X' = \{(x',u)\}$, $x'$ denotes the subsequent states for transitions starting from $s$; and in $X'' = \{(x'',u)\}$, $x''$ denotes subsequent states for transitions starting from $s'$, such that:*

$$\hat{C}_{DARD}(R)(s,a,s') = R(s,a,s') + \frac{\gamma}{N''} \sum_{(x'',u) \in X''} R(s',u,x'') - \frac{1}{N'} \sum_{(x',u) \in X'} R(s,u,x')$$
$$- \frac{\gamma}{N'N''} \sum_{(x',\cdot) \in X'} \sum_{(x'',u) \in X''} R(x',u,x'') \tag{17}$$

**Definition 8.** *(Sample-based SRRD) Given a set of transitions $B_V$ from a reward sample, let $B_M$ be a batch of $N_M$ state-action pairs sampled from the explored states and actions. From $B_M$, derive sets $X_i \subseteq B_M$, for $i \in \{1, ..., 6\}$. Each $X_i$ is a set, $\{(x,u)\}$, where $x$ is a state and $u$ is an action. The magnitude, $|X_i| = N_i$. We define $X_3 = \{(x_3,u)\}$, where $x_3$ denotes all initial states for all transitions; $X_4 = \{(x_4,u)\}$, where $x_4$ denotes all subsequent states for all transitions. For each $(s,a,s')$ transition in $B_V$, we define $X_1 = \{(x_1,u)\}$, where $x_1$ denotes subsequent states for transitions starting from $s'$; $X_2 = \{(x_2,u)\}$, where $x_2$ denotes non-terminal subsequent states for transitions starting from $s$; $X_5 = \{(x_5,u)\}$, where $x_5$ denotes subsequent states to $X_1$; and $X_6 = \{(x_6,u)\}$, where $x_6$ denotes subsequent states to $X_2$. For each $(s,a,s')$ in $B_V$:*

$$C_{SRRD}(R)(s,a,s') \approx R(s,a,s') + \frac{\gamma}{N_1} \sum_{(x_1,u) \in X_1} R(s',u,x_1) - \frac{1}{N_2} \sum_{(x_2,u) \in X_2} R(s,u,x_2)$$
$$- \frac{\gamma}{N_3 N_4} \sum_{(x_3,u) \in X_3} \sum_{(x_4,\cdot) \in X_4} R(x_3,u,x_4) + \frac{\gamma^2}{N_1 N_5} \sum_{(x_1,u) \in X_1} \sum_{(x_5,\cdot) \in X_5} R(x_1,u,x_5)$$
$$- \frac{\gamma}{N_2 N_6} \sum_{(x_2,u) \in X_2} \sum_{(x_6,\cdot) \in X_6} R(x_2,u,x_6) + \frac{\gamma}{N_3 N_6} \sum_{(x_3,u) \in X_3} \sum_{(x_6,\cdot) \in X_6} R(x_3,u,x_6) \tag{18}$$
$$- \frac{\gamma^2}{N_4 N_5} \sum_{(x_4,u) \in X_4} \sum_{(x_5,\cdot) \in X_5} R(x_4,u,x_5).$$

In each approximation, each term estimates the corresponding term in the original formula. For example, in $\hat{C}_{SRRD}, -\frac{\gamma}{N_3 N_4}\sum_{(x_3,u)\in X_3}\sum_{(x_4,\cdot)\in X_4} R(x_3, u, x_4)$ estimates $-\gamma\mathbb{E}[R(S_3, A, S_4)]$ and $\frac{\gamma}{N_1}\sum_{(x_1,u)\in X_1} R(s', u, x_1)$ estimates $\gamma\mathbb{E}[R(s', A, S_1)]$. These approximations utilize two batches: $B_V$ containing $N_V$ transitions and $B_M$ containing $N_M$ state-action pairs, to canonicalize rewards in $B_V$. Each transition in $B_V$ is canonicalized using state-action pairs in $B_M$. Depending on how $B_M$ is sampled (usually uniformly), it can help reduce bias from the sampling policy used to generate $B_V$, by including state-action pairs that may not be present in $B_V$; thereby providing a better representation of the entire state-action space. As a result, $B_M$ can help to provide broader coverage, and reduce bias and variance in results, using relatively fewer state-action pairs compared to enumerating all possible combinations. In the original $\hat{C}_{EPIC}$, and $\hat{C}_{DARD}$ approximations, it's unclear how duplicates would be handled. In our experiments, we implement all batches as sets, thereby removing duplicates to further reduce sampling bias.

## A.2 Relative Shaping Error Comparisons

This section provides the proof for Theorem 1 which compares the upper bounds of the relative shaping errors for $\hat{C}_{SRRD}$, $\hat{C}_{DARD}$, and $\hat{C}_{EPIC}$. We organize the presentation as follows:

1. Discuss the motivation for the relative shaping errors in quantifying the variation in the residual shaping normalized by the upper bound base (unshaped) canonical rewards (Appendix A.2.1).

2. Proof of Theorem 1 (Appendix A.2.2).

### A.2.1 Motivation for the Relative Shaping Error (RSE) Measure

Consider the task of comparing two equivalent reward samples, $R_1'$ and $R_2'$, which induce similar policies, share the same set of transitions $B$, but may differ due to potential shaping. To compare the reward samples, we first eliminate shaping by applying a sample-based canonicalization method, $\hat{C}_S$. After canonicalization, we compute the Pearson distance to quantify the difference between the canonical rewards as:

$$D_\rho(\hat{C}_S(R_1'), \hat{C}_S(R_2')) = \sqrt{1 - \rho(\hat{C}_S(R_1'), \hat{C}_S(R_2'))}/\sqrt{2}, \tag{19}$$

where $\rho$ is the Pearson correlation. Since $R_1'$ and $R_2'$ are assumed to have the same set of transitions, and only differ due to potential shaping, after canonicalization, we have:

$$\hat{C}_S(R_1') = \hat{C}_S(R_1) + \phi_{R_1} = \hat{C}_S(R) + \phi_{R_1},$$
$$\hat{C}_S(R_2') = \hat{C}_S(R_2) + \phi_{R_2} = \hat{C}_S(R) + \phi_{R_2},$$

where $\phi_{R_1}$ and $\phi_{R_2}$ are residual shaping terms, and $\hat{C}_S(R)$ is the **base canonical reward**, which is similar between the reward samples since the sets of explored transitions are similar[3], hence, $\hat{C}_S(R_1) = \hat{C}_S(R_2) = \hat{C}_S(R)$. When computing $D_\rho$ (Equation 19), the Pearson correlation can thus be expressed as:

$$\rho(\hat{C}_S(R_1'), \hat{C}_S(R_2')) = \rho(\hat{C}_S(R) + \phi_{R_1}, \hat{C}_S(R) + \phi_{R_2}). \tag{20}$$

Analyzing Equation 20:

- When $|\phi_{R_i}| << |\hat{C}_S(R)|$ for all $i \in \{1, 2\}$ then $\rho(\hat{C}_S(R_1'), \hat{C}_S(R_2')) \approx \rho(\hat{C}_S(R), \hat{C}_S(R)) = 1$, since the residual shaping terms have negligible impact. Conversely, when either $|\phi_{R_1}| >> |\hat{C}_S(R)|$ or $|\phi_{R_2}| >> |\hat{C}_S(R)|$, then the Pearson correlation is predominantly influenced by the residual shaping terms, and $\rho$ can become low or even negative, especially when $\phi_{R_1}$ and $\phi_{R_2}$ differ significantly.

- Since $\hat{C}_S(R)$ is the same across $\hat{C}_S(R_1')$ and $\hat{C}_S(R_2')$, variations in the Pearson correlation can be primarily attributed to variations in the residual shaping terms.

---

[3]We assume that the reward samples under comparison are equivalent (in terms of optimal policies) and share the same transitions, to simplify our analysis by isolating the differences between the reward samples to potential shaping.

- To assess the impact of residual shaping, it is essential to also consider the magnitude $\hat{C}_S(R)$ as a normalizing factor. If $|\hat{C}_S(R)|$ is significantly larger than the residual shaping terms, the influence of the shaping terms diminishes, resulting in a higher Pearson correlation closer to 1. Conversely, if $|\hat{C}_S(R)|$ is significantly smaller than the residual shaping terms, these shaping terms will have a higher influence on the correlation, potentially reducing it.

To generalize these insights, let's consider the canonicalization task on a reward sample $R'$ such that:

$$\hat{C}_S(R') = \hat{C}_S(R) + \phi_R. \tag{21}$$

To quantify the influence of the residual shaping term $\phi_R$ relative to the base canonical reward $\hat{C}_S(R)$, let's establish the Relative Shaping Error (RSE) as follows:

$$RSE(\hat{C}_S(R)(s,a,s')) = \frac{|\phi_R(s,a,s')|}{U(\hat{C}_S(R)(s,a,s'))} = \frac{|\phi_R(s,a,s')|}{nZ}. \tag{22}$$

A low RSE indicates that $U(\hat{C}_S(R)(s,a,s'))$ is substantially large relative to $|\phi_R(s,a,s')|$, suggesting that the residual shaping has minimal impact on $D_\rho$. Conversely, a high RSE implies that $U(\hat{C}_S(R)(s,a,s'))$ is small relative to $|\phi_R(s,a,s')|$, highlighting a more significant influence of residual shaping. By normalizing with the upper bound of the RSE, we obtain a conservative measure to quantify the impact of shaping in extreme scenarios. This formulation helps assess the robustness of reward canonicalization methods in mitigating shaping effects during reward comparisons. From Equation 21, suppose we can express:

$$\hat{C}_S(R)(s,a,s') = \hat{C}_S(\tilde{R})(s,a,s') + K_R,$$

$$\phi_R(s,a,s') = \phi_{\tilde{R}}(s,a,s') + K_\phi,$$

where $K_R$ and $K_\phi$ are constants that do not vary with $(s,a,s')$, and $\phi_{\tilde{R}}$ is the **effective residual shaping**. When comparing reward samples $R_1'$ and $R_2'$, in computing $D_\rho$, the constants $K_R$ and $K_\phi$ do not affect the Pearson correlation since it is shift invariant. Therefore:

$$\begin{aligned}
\rho(\hat{C}_S(R_1'), \hat{C}_S(R_2')) &= \rho(\hat{C}_S(R_1) + \phi_{R_1}, \hat{C}_S(R_2) + \phi_{R_2}) \\
&= \rho(\hat{C}_S(\tilde{R}_1) + K_{R_1} + \phi_{\tilde{R}_1} + K_{\phi_1}, \hat{C}_S(\tilde{R}_2) + K_{R_2} + \phi_{\tilde{R}_2} + K_{\phi_2}) \\
&= \rho(\hat{C}_S(\tilde{R}) + \phi_{\tilde{R}_1}, \hat{C}_S(\tilde{R}) + \phi_{\tilde{R}_2})
\end{aligned} \tag{23}$$

Relating this to the RSE in Equation 22, note that $|\phi_R|$ serves as a measure of reward variation due to shaping, and $U(\hat{C}_S(R))$ acts as a normalizing term. For the term $\phi_R$, we can omit $K_\phi$ to get:

$$\phi_{\tilde{R}} = \phi_R - K_\phi, \tag{24}$$

since, $K_\phi$ does not affect the variation in the Pearson correlation. However, for the denominator $U(\hat{C}_S(R))$, we cannot omit $K_R$ (hence no change to the denominator) because the denominator serves to normalize the residual shaping. When $K_R$ is large, even though it does not affect the Pearson correlation, it lowers the impact of $|\phi_{\tilde{R}}|$ and vice versa. This leads to the final RSE equation below:

$$RSE(\hat{C}_S(R)(s,a,s')) = \frac{|\phi_{\tilde{R}}(s,a,s')|}{U(\hat{C}_S(R)(s,a,s'))} = \frac{|\phi_{\tilde{R}}(s,a,s')|}{nZ}. \tag{25}$$

### A.2.2 Proof of Theorem 1

Theorem 1 aims to compare the upper bounds of the RSEs for $\hat{C}_{EPIC}$, $\hat{C}_{DARD}$, and $\hat{C}_{SRRD}$.

*Proof.* Assuming a finite reward sample that spans a state space $S^{\mathcal{D}}$ and an action space $A^{\mathcal{D}}$, where $\mathcal{D}$ is the coverage distribution. The following subsets are defined: for $\hat{C}_{EPIC}$, $S \subseteq S^{\mathcal{D}}$ and $S' \subseteq S^{\mathcal{D}}$; for $\hat{C}_{DARD}$, $S'' \subseteq S^{\mathcal{D}}$ and $S' \subseteq S^{\mathcal{D}}$; and for $\hat{C}_{SRRD}$, each $S_i \subseteq S^{\mathcal{D}}$, where $i \in \{1,\ldots,6\}$. Let's define:

$$M = \max_{s \in S^{\mathcal{D}}}(|\phi(s)|),$$

as the maximum absolute shaping for states in $S^{\mathcal{D}}$, where $M \in \mathbb{R}$. Then, for all shaping expectations:

$$|\mathbb{E}[\phi(S)]| \leq M, \quad |\mathbb{E}[\phi(S')]| \leq M, \quad |\mathbb{E}[\phi(S'')]| \leq M, \quad \text{and} \quad |\mathbb{E}[\phi(S_i)]| \leq M \quad \text{for all} \quad i \in \{1, \ldots, 6\}.$$

In the following analysis, we impose the assumption that unsampled forward transitions are negligible, based on the rationale that forward transitions are generally consistent, or in-distribution with the transition dynamics of reward samples, compared to non-forward transitions which have a higher chance of being out-of-distribution with respect to the reward samples' transition dynamics. For the analysis, we find the upper bound of the RSEs under different scenarios of transition sparsity from the best case (no unsampled transitions) to the worst case (high levels of unsampled non-forward transitions).

**Analysis of EPIC:**

Considering Equation 2 for $C_{EPIC}$, the transitions $(S, A, S')$ are forward transitions since by definition, $S'$ is created based on $(S, A)$. However, the transitions: $(s, A, S')$ and $(s', A, S')$ are non-forward transitions since $S'$ is not created based on $(s, A)$ or $(s', A)$, but $(S, A)$. Since we only have unsampled non-forward transitions, let the fraction of randomly sampled transitions in $(s', A, S')$ and $(s, A, S')$ be $u$ and $v$ respectively, where $0 \leq u, v \leq 1$. Note that $u$ and $v$ vary based on $(s, a, s')$; however, for visual clarity, we will denote them as $u$ and $v$. Incorporating $u$ and $v$ into $\hat{C}_{EPIC}$:

$$\hat{C}_{EPIC}(R)(s, a, s') = R(s, a, s') + \mathbb{E}[u\gamma R(s', A, S') - vR(s, A, S') - \gamma R(S, A, S')], \tag{26}$$

Applying $\hat{C}_{EPIC}$ (Equation 26) to a shaped reward $R'(s, a, s')$, we get the residual shaping:

$$\phi_{epic} = (\gamma - \gamma u)\phi(s') + (v - 1)\phi(s) + \mathbb{E}[(\gamma^2 u - \gamma^2)\phi(S')] + \mathbb{E}[\gamma\phi(S)] - \mathbb{E}[\gamma v\phi(S')] \tag{27}$$

**Best Case:** In this scenario there are no unsampled transitions such that: $u, v = 1$. Applying $u, v = 1$ into Equation 27, we get: $\phi_{epic} = \mathbb{E}[\gamma\phi(S)] - \mathbb{E}[\gamma\phi(S')]$.

In $\hat{C}_{EPIC}$, all transitions are canonicalized using the same state subsets $S$ and $S'$ such that the shaping terms $\phi(S)$ and $\phi(S')$ do not vary with changes in $(s, a, s')$. Therefore, in $\phi_{epic}$, the constant $K_\phi = \mathbb{E}[\gamma\phi(S)] - \mathbb{E}[\gamma\phi(S')]$ does not vary with changes in $(s, a, s')$, and based on Equation 24, $\phi_{\tilde{epic}} = 0$, such that:

$$RSE(\hat{C}_{EPIC}(R)) = 0.$$

**Average Case:** In this scenario, $0 < u, v < 1$. From $\phi_{epic}$ (Equation 27), we can extract the constant term $K_\phi = -\mathbb{E}[\gamma^2\phi(S')] + \mathbb{E}[\gamma\phi(S)]$, which does not vary with $(s, a, s')$. Therefore, the effective residual shaping (see Equation 24) is given by:

$$\phi_{\tilde{epic}} = (\gamma - \gamma u)\phi(s') + (v - 1)\phi(s) + \mathbb{E}[(\gamma^2 u - \gamma v)\phi(S')], \tag{28}$$

such that:

$$\begin{aligned}
|\phi_{\tilde{epic}}| &\leq (\gamma - \gamma u)|\phi(s')| + (1 - v)|(-\phi(s))| + |\gamma^2 u - \gamma v||\mathbb{E}[\phi(S')]| \\
&\leq M(\gamma - \gamma u + 1 - v + |\gamma^2 u - \gamma v|), \qquad \left(\text{since } M = \max_{s \in S^{\mathcal{D}}} |\phi(s)|\right). \\
&\leq M(2 - u - v + |u - v|) \qquad \text{(when } \gamma = 1) \\
&\leq 2M
\end{aligned}$$

From Equation 26, $\hat{C}_{EPIC}$ has four reward terms. Following Definition 2, the upper bound of $\hat{C}_{EPIC}(R)$ is $U(\hat{C}_{EPIC}(R)) = 4Z$, hence:

$$RSE(\hat{C}_{EPIC}(R)) = \frac{|\phi_{\tilde{epic}}|}{U(\hat{C}_{EPIC}(R))} \leq \frac{2M}{4Z} = \frac{M}{2Z}.$$

**Worst Case:** In this scenario, $u, v = 0$. Applying $u$ and $v$ into $\hat{C}_{EPIC}$ (Equation 26), we get:

$$\hat{C}_{EPIC}(R)(s, a, s') = R(s, a, s') - \mathbb{E}[\gamma R(S, A, S')], \tag{29}$$

and $\phi_{epic} = \gamma\phi(s') - \phi(s) - \mathbb{E}[\gamma^2\phi(S')] + \mathbb{E}[\gamma\phi(S)]$. Since $S$ and $S'$ are the same across all transitions, $K_\phi = -\mathbb{E}[\gamma^2\phi(S')] + \mathbb{E}[\gamma\phi(S)]$ is a constant, hence, $\phi_{\tilde{epic}} = \gamma\phi(s') - \phi(s)$ such that:

$$|\phi_{\tilde{epic}}| \leq |\gamma\phi(s')| + |(-\phi(s))| \leq \gamma M + M \leq 2M$$

From Equation 29, $\hat{C}_{EPIC}$ has two reward terms. Following Definition 2, $U(\hat{C}_{EPIC}(R)) = 2Z$, hence:

$$RSE(\hat{C}_{EPIC}(R)) = \frac{|\phi_{\tilde{epic}}|}{U(\hat{C}_{EPIC}(R))} \leq \frac{M}{Z}.$$

$\therefore$ Based on all the three cases, we have:

$$RSE(\hat{C}_{EPIC}(R)) \leq \frac{M}{Z}. \tag{30}$$

**Analysis of DARD:**

Considering Equation 6 for $C_{DARD}$, the state subset $S''$ is subsequent to $s'$, and $S'$ is subsequent to $s$. For approximations, the transitions $(S', A, S'')$ are non-forward transitions since $S''$ is created based on $(s', A)$ rather than $(S', A)$. Let the fraction of randomly sampled transitions in $(S', A, S'')$ be $w$, where $0 \leq w \leq 1$. Note that $w$ also depends on $(s, a, s')$; however, for visual clarity we denote it as simply $w$. Incorporating $w$ into $\hat{C}_{DARD}$:

$$\hat{C}_{DARD}(R)(s, a, s') = R(s, a, s') + \mathbb{E}[\gamma R(s', A, S'') - R(s, A, S') - w\gamma R(S', A, S'')], \tag{31}$$

Applying $\hat{C}_{DARD}$ to a shaped reward $R'$, we get the residual shaping:

$$\phi_{dard} = \mathbb{E}[(\gamma^2 - w\gamma^2)\phi(S'') + (w\gamma - \gamma)\phi(S')] \tag{32}$$

**Best Case:** In this scenario, $w = 1$, such that $|\phi_{dard}| = 0$, hence, $RSE(\hat{C}_{DARD}) = 0$.

**Average Case:** For this scenario, $0 < w < 1$:

$$|\phi_{dard}| \leq (\gamma^2 - w\gamma^2)|\mathbb{E}[\phi(S'')]| + (\gamma - w\gamma)|(-\mathbb{E}[\phi(S')])| \leq (\gamma^2 - w\gamma^2)M + (\gamma - w\gamma)M \leq 2M$$

In $\hat{C}_{DARD}$, the state subsets $S'$ and $S''$ vary due to $(s, a, s')$; hence, $K_\phi = 0$. Following Definition 3, and applying it to $\hat{C}_{DARD}$ (Equation 31), $U(\hat{C}_{DARD}(R)) = 4Z$, such that:

$$RSE(\hat{C}_{DARD}(R)) = \frac{|\phi_{dard}|}{U(\hat{C}_{DARD}(R))} \leq \frac{2M}{4Z} = \frac{M}{2Z}.$$

**Worst Case:** In this scenario, $w = 0$, hence we eliminate terms with $w$ in Equation 31 to get:

$$\hat{C}_{DARD}(R)(s, a, s') = R(s, a, s') + \mathbb{E}[\gamma R(s', A, S'') - R(s, A, S')], \tag{33}$$

which corresponds to: $\phi_{dard} = \mathbb{E}[\gamma^2\phi(S'') + \gamma\phi(S')]$, such that:

$$|\phi_{dard}| \leq |\mathbb{E}[\gamma^2\phi(S'')]| + |\mathbb{E}[\gamma\phi(S')]| \leq |\gamma^2 M| + |\gamma M| \leq 2M$$

Following Definition 2, and applying it to $\hat{C}_{DARD}$ (Equation 33): $U(\hat{C}_{DARD}(R)) = 3Z$, such that:

$$RSE(\hat{C}_{DARD}(R)) = \frac{|\phi_{dard}|}{U(\hat{C}_{DARD}(R))} \leq \frac{2M}{3Z}.$$

$\therefore$ Based on all the three cases, we have:

$$RSE(\hat{C}_{DARD}(R)) \leq \frac{2M}{3Z}. \tag{34}$$

**Analysis of SRRD**

Considering $C_{SRRD}$ in Definition 1, the first six reward terms are forward transitions, since for each set of transitions $(S_i, A, S_j)$, $S_j$ is distributed conditionally based on $(S_i, A)$. The last two terms violate this condition; hence, they are non-forward transitions. As mentioned in Section 4, SRRD is designed such that:

- **Transitions** $(S_1, A, S_5) \subseteq (S_4, A, S_5)$:

  Since $S_1$ are subsequent states to $s'$, and $S_4$ are subsequent states for all sampled transitions. It follows that $S_1 \subseteq S_4$, hence, $(S_1, A, S_5) \subseteq (S_4, A, S_5)$.

- **Transitions** $(S_2, A, S_6) \subseteq (S_3, A, S_6)$:

  $S_2$ is the set of non-terminal subsequent states to $s$. Since $S_3$ encompasses all initial states from all sampled transitions, it follows that $S_2 \subseteq S_3$, hence, $(S_2, A, S_6) \subseteq (S_3, A, S_6)$.

For $\hat{C}_{SRRD}$, let the fraction of the randomly sampled transitions for $(S_4, A, S_5)$ and $(S_3, A, S_6)$ be $p$ and $q$, respectively. Considering that the number of unsampled forward transitions are negligible, we establish minimal thresholds $m_1, m_2 > 0$ such that: $p = m_1$ when $(S_4, A, S_5)$ only contains sampled transitions from $(S_1, A, S_5)$, and $q = m_2$ when $(S_3, A, S_6)$ only contains sampled transitions from $(S_2, A, S_6)$. Therefore: $m_1 \leq p \leq 1$ and $m_2 \leq q \leq 1$. Note that $p$ and $q$ vary based on $(s, a, s')$, but for visual clarity, we express them as $p$ and $q$. Incorporating $p$ and $q$, we get:

$$\hat{C}_{SRRD}(R)(s,a,s') = R(s,a,s') + \mathbb{E}[\gamma R(s', A, S_1) - R(s, A, S_2) - \gamma R(S_3, A, S_4) + \gamma^2 R(S_1, A, S_5)$$
$$- \gamma R(S_2, A, S_6) + q\gamma R(S_3, A, S_6) - p\gamma^2 R(S_4, A, S_5)] \tag{35}$$

Applying $\hat{C}_{SRRD}$ (Equation 35) to a shaped reward $R'(s, a, s')$, we get the residual shaping:

$$\phi_{srrd} = \mathbb{E}[(\gamma - q\gamma)\phi(S_3) + (p\gamma^2 - \gamma^2)\phi(S_4) + (\gamma^3 - p\gamma^3)\phi(S_5) + (q\gamma^2 - \gamma^2)\phi(S_6)] \tag{36}$$

**Best Case:** In this scenario there are no unsampled transitions such that: $p, q = 1$. Applying $p$ and $q$ into Equation 36, we get: $\phi_{srrd} = 0$, hence, $RSE(\hat{C}_{SRRD}(R)) = 0$.

**Average Case:** In this scenario, $m_1 < p < 1$ and $m_2 < q < 1$. Note that in $\hat{C}_{SRRD}$, $S_3$ and $S_4$ are the same for all transitions, since $S_3$ encompasses all initial states, and $S_4$ includes all subsequent states to $S_3$. Therefore, $\phi(S_3)$ and $\phi(S_4)$ do not vary with $(s, a, s')$. From $\phi_{srrd}$ (Equation 36), we can extract the constant $K_\phi = \mathbb{E}[\gamma\phi(S_3) - \gamma^2\phi(S_4)]$. Based on Equation 24, the effective residual shaping is given by:

$$\phi_{s\tilde{r}rd} = \mathbb{E}[-q\gamma\phi(S_3) + p\gamma^2\phi(S_4) + (\gamma^3 - p\gamma^3)\phi(S_5) + (q\gamma^2 - \gamma^2)\phi(S_6)]$$

such that:

$$\phi_{s\tilde{r}rd} \leq q\gamma|(-\mathbb{E}[\phi(S_3)])| + p\gamma^2|\mathbb{E}[\phi(S_4)]| + (\gamma^3 - p\gamma^3)|\mathbb{E}[\phi(S_5)]| + (\gamma^2 - q\gamma^2)|(-\mathbb{E}[\phi(S_6)])|$$
$$\leq M(q\gamma + p\gamma^2 + \gamma^3 - p\gamma^3 + \gamma^2 - q\gamma^2)$$
$$\leq M(q + p + 1 - p + 1 - q) \qquad \text{(when } \gamma = 1\text{)}$$
$$\leq 2M$$

Since $\hat{C}_{SRRD}$ has 8 terms, following Definition 3 and applying it to Equation 35, we get $U(\hat{C}_{SRRD}(R)) = 8Z$, Therefore:

$$RSE(\hat{C}_{SRRD}(R)) = \frac{|\phi_{s\tilde{r}rd}|}{U(\hat{C}_{SRRD}(R))} \leq \frac{M}{4Z}.$$

**Worst Case:** In this scenario, $p = m_1$ (when $(S_4, A, S_5)$ only has sampled transitions from $(S_1, A, S_5)$) and $q = m_2$ (when $(S_3, A, S_6)$ only has sampled transitions from $(S_2, A, S_6)$). Consider a situation where $|(S_4, A, S_5)| >> |(S_1, A, S_5)|$, and $|(S_3, A, S_6)| >> |(S_2, A, S_6)|$, such that: $m_1 \to 0$ and $m_2 \to 0$. Therefore:

$$\hat{C}_{SRRD}(R)(s,a,s') \approx R(s,a,s') + \mathbb{E}[\gamma R(s', A, S_1) - R(s, A, S_2) - \gamma R(S_3, A, S_4) + \gamma^2 R(S_1, A, S_5)$$
$$- \gamma R(S_2, A, S_6)] \tag{37}$$

with the corresponding residual shaping:

$$\phi_{srrd} \approx \mathbb{E}[\gamma\phi(S_3) - \gamma^2\phi(S_4) + \gamma^3\phi(S_5) - \gamma^2\phi(S_6)]. \tag{38}$$

From $\phi_{srrd}$ (Equation 38), we can extract the constant $K_\phi = \mathbb{E}[\gamma\phi(S_3) - \gamma^2\phi(S_4)]$ since it does not vary with $(s, a, s')$. Therefore, the effective residual shaping is given by:

$$\phi_{s\tilde{r}rd} \approx \mathbb{E}[\gamma^3\phi(S_5) - \gamma^2\phi(S_6)],$$

hence:

$$|\phi_{s\tilde{r}rd}| \le \gamma^3|\mathbb{E}[\phi(S_5)]| + \gamma^2|(-\mathbb{E}[\phi(S_6)])|| \le |\gamma^3 M + \gamma^2 M| \le 2M$$

Following Definition 2, and applying it to $\hat{C}_{SRRD}$ (Equation 37): $U(\hat{C}_{SRRD}(R)) = 6Z$, such that:

$$RSE(\hat{C}_{SRRD}(R)) = \frac{|\phi_{s\tilde{r}rd}|}{U(\hat{C}_{SRRD}(R)} \le \frac{M}{3Z}.$$

$\therefore$ Based on all the three cases, we have:

$$RSE(\hat{C}_{SRRD}(R)) \le \frac{M}{3Z}. \tag{39}$$

**Conclusion**

Based on the upper bounds of the RSE values aggregated from the best, average and worst case scenarios of transition sparsity (Equation 39, 34, 30), we can conclude that:

$$\text{RSE}(\hat{C}_{SRRD}) \le \frac{M}{3Z}; \quad \text{RSE}(\hat{C}_{DARD}) \le \frac{2M}{3Z}; \ \text{RSE}(\hat{C}_{EPIC}) \le \frac{M}{Z},$$

$$\square$$

The RSE serves as a theoretical measure to evaluate the robustness of the pseudometrics by quantifying the influence of residual shaping errors relative to the rewards. It's important to acknowledge that the RSE is a conservative measure, since the residual shaping is normalized by $U(C_S(R))$. Therefore, in practice, it may not guarantee the order of performance predicted in Theorem 1, as the impact of residual shaping could be more pronounced when normalized by smaller reward values. Despite this limitation, the RSE still offers a meaningful theoretical measure on the robustness of the pseudometrics.

**A.3 Residual Shaping**

**Derivation of $\phi_{res1}$:** Applying $C_1$ (Equation 7) to a shaped reward $R'(s, a, s') = R(s, a, s') + \gamma\phi(s') - \phi(s)$:

$$
\begin{aligned}
C_1(R')(s, a, s') &= R'(s, a, s') + \mathbb{E}[\gamma R'(s', A, S_1) - R'(s, A, S_2) - \gamma R'(S_3, A, S_4)] \\
&= R(s, a, s') + \gamma\phi(s') - \phi(s) + \mathbb{E}[\gamma(R(s', A, S_1) + \gamma\phi(S_1) - \phi(s')) \\
&\quad - (R(s, A, S_2) + \gamma\phi(S_2) - \phi(s)) - \gamma(R(S_3, A, S_4) + \gamma\phi(S_4) - \phi(S_3))] \\
&= C_1(R)(s, a, s') + \mathbb{E}[\gamma^2\phi(S_1) - \gamma^2\phi(S_4) + \gamma\phi(S_3) - \gamma\phi(S_2)]
\end{aligned}
$$

Hence, $C_1(R)(s, a, s')$ yields the residual shaping:

$$\phi_{res1} = \mathbb{E}[\gamma^2\phi(S_1) - \gamma^2\phi(S_4) + \gamma\phi(S_3) - \gamma\phi(S_2)].$$

**Derivation of $\phi_{res2}$:**   Applying $C_2$ (Equation 9) to shaped reward $R'(s, a, s') = R(s, a, s') + \gamma\phi(s') - \phi(s)$:

$$
\begin{aligned}
C_2(R')(s, a, s') = {} & R(s, a, s') + \gamma\phi(s') - \phi(s) + \mathbb{E}[\gamma(R(s', A, S_1) + \gamma\phi(S_1) - \phi(s')) \\
& - (R(s, A, S_2) + \gamma\phi(S_2) - \phi(s)) - \gamma(R(S_3, A, S_4) + \gamma\phi(S_4) - \phi(S_3)) \\
& + \gamma^2(R(S_1, A, k_1) + \gamma\phi(k_1) - \phi(S_1)) - \gamma(R(S_2, A, k_2) + \gamma\phi(k_2) - \phi(S_2)) \\
& + \gamma(R(S_3, A, k_3) + \gamma\phi(k_3) - \phi(S_3)) - \gamma^2(R(S_4, A, k_4) + \gamma\phi(k_4) - \phi(S_4))]. \\
= {} & C_2(R)(s, a, s') + \mathbb{E}[\gamma^3\phi(k_1) - \gamma^3\phi(k_4) + \gamma^2\phi(k_3) - \gamma^2\phi(k_2)]
\end{aligned}
$$

Hence, $C_2(R)(s, a, s')$ yields the residual shaping:

$$
\phi_{res2} = \mathbb{E}[\gamma^3\phi(k_1) - \gamma^3\phi(k_4) + \gamma^2\phi(k_3) - \gamma^2\phi(k_2)].
$$

## A.4   The Sparsity Resilient Canonically Shaped Reward is Invariant to Shaping

**Proposition 1.**   *(The Sparsity Resilient Canonically Shaped Reward is Invariant to Shaping) Let $R$ : $\mathcal{S} \times \mathcal{A} \times \mathcal{S}$ be a reward function and $\phi : \mathcal{S} \to \mathbb{R}$ be a state potential function. Applying $C_{SRRD}$ to a potentially shaped reward $R'(s, a, s') = R(s, a, s') + \gamma\phi(s') - \phi(s)$ satisfies: $C_{SRRD}(R) = C_{SRRD}(R')$.*

*Proof.* Let's apply $C_{SRRD}$, Definition 1, to a shaped reward $R'(s, a, s') = R(s, a, s') + \gamma\phi(s') - \phi(s)$:

$$
\begin{aligned}
C_{SRRD}(R')(s, a, s') = {} & R(s, a, s') + \gamma\phi(s') - \phi(s) + \mathbb{E}[\gamma[R(s', A, S_1) + \gamma\phi(S_1) - \phi(s')] \\
& - [R(s, A, S_2) + \gamma\phi(S_2) - \phi(s)] - \gamma[R(S_3, A, S_4) + \gamma\phi(S_4) - \phi(S_3)] \\
& + \gamma^2[R(S_1, A, S_5) + \gamma\phi(S_5) - \phi(S_1)] - \gamma[R(S_2, A, S_6) + \gamma\phi(S_6) - \phi(S_2)] \\
& + \gamma[R(S_3, A, S_6) + \gamma\phi(S_6) - \phi(S_3)] - \gamma^2[R(S_4, A, S_5) + \gamma\phi(S_5) - \phi(S_4)]],
\end{aligned}
$$

Regrouping the reward terms and the potentials, this reduces to:

$$
\begin{aligned}
C_{SRRD}(R')(s, a, s') = {} & C_{SRRD}(R)(s, a, s') + (\gamma\phi(s') - \gamma\mathbb{E}[\phi(s')]) + (-\phi(s) + \mathbb{E}[\phi(s)]) \\
& + \mathbb{E}[\gamma^2(\phi(S_1) - \phi(S_1))] + \mathbb{E}[\gamma(-\phi(S_2) + \phi(S_2))] + \mathbb{E}[\gamma(\phi(S_3) - \phi(S_3))] \\
& + \mathbb{E}[\gamma^2(-\phi(S_4) + \phi(S_4))] + \mathbb{E}[\gamma^3(\phi(S_5) - \phi(S_5))] + \mathbb{E}[\gamma^2(-\phi(S_6) + \phi(S_6))]
\end{aligned}
$$

Since $\mathbb{E}[\gamma\phi(s')] = \gamma\phi(s')$ and $\mathbb{E}[\phi(s)] = \phi(s)$, this leads to:

$$
C_{SRRD}(R')(s, a, s') = C_{SRRD}(R)(s, a, s').
$$

$\square$

## A.5   Invariance under the Sample-Based SRRD Approximation

For the sample-based SRRD canonical reward (Definition 8), under transition sparsity, some rewards may be undefined since the associated transitions may be unsampled. However, *when all the necessary rewards are defined, policy invariance can be achieved.* To show this, we will first derive Lemma 1, then present the proof that assumes all the necessary rewards are available.

**Lemma 1.** *Let $\phi : S \to \mathbb{R}$ be a potential function. Given states $x_i \in S$ and $x_j \in S$, then:*

$$
\frac{1}{n_1 n_2} \sum_{i=1}^{n_1} \sum_{j=1}^{n_2} (\gamma\phi(x_i) - \phi(x_j)) = \frac{\gamma}{n_1} \sum_{i=1}^{n_1} \phi(x_i) - \frac{1}{n_2} \sum_{j=1}^{n_2} \phi(x_j).
$$

*Proof.*

$$\frac{1}{n_1 n_2} \sum_{i=1}^{n_1} \sum_{j=1}^{n_2} (\gamma\phi(x_i) - \phi(x_j)) = \frac{1}{n_1 n_2} \left( \gamma \sum_{i=1}^{n_1} \left( \sum_{j=1}^{n_2} \phi(x_i) \right) - \sum_{j=1}^{n_2} \left( \sum_{i=1}^{n_1} \phi(x_j) \right) \right)$$

Notice that $\phi(x_i)$ is independent of $j$ and $\phi(x_j)$ is independent of $i$, thus,

$$= \frac{\gamma}{n_1 n_2} \sum_{i=1}^{n_1} n_2 \phi(x_i) - \frac{1}{n_1 n_2} \sum_{j=1}^{n_2} n_1 \phi(x_j)$$

$$= \frac{\gamma}{n_1} \sum_{i=1}^{n_1} \phi(x_i) - \frac{1}{n_2} \sum_{j=1}^{n_2} \phi(x_j).$$

$\square$

**Proposition 2.** *Given transition sets $(S_i, A, S_j)$ needed to compute each reward expectation term in $\hat{C}_{SRRD}$, if the transitions fully cover the cross-product $S_i \times A \times S_j$, then for a shaped reward $R'(s, a, s') = R(s, a, s') + \gamma\phi(s') - \phi(s)$, the sample-based SRRD approximation is invariant to shaping.*

*Proof.*

$$\hat{C}_{SRRD}(R')(s, a, s') \approx R(s, a, s') + \gamma\phi(s') - \phi(s)$$

$$+ \frac{\gamma}{N_1} \sum_{(x_1, u) \in X_1} [R(s', u, x_1) + \gamma\phi(x_1) - \phi(s')] - \frac{1}{N_2} \sum_{(x_2, u) \in X_2} [R(s, u, x_2) + \gamma\phi(x_2) - \phi(s)]$$

$$- \frac{\gamma}{N_3 N_4} \sum_{(x_3, u) \in X_3} \sum_{(x_4, \cdot) \in X_4} [R(x_3, u, x_4) + \gamma\phi(x_4) - \phi(x_3)]$$

$$+ \frac{\gamma^2}{N_1 N_5} \sum_{(x_1, u) \in X_1} \sum_{(x_5, \cdot) \in X_5} [R(x_1, u, x_5) + \gamma\phi(x_5) - \phi(x_1)]$$

$$- \frac{\gamma}{N_2 N_6} \sum_{(x_2, u) \in X_2} \sum_{(x_6, \cdot) \in X_6} [R(x_2, u, x_6) + \gamma\phi(x_6) - \phi(x_2)]$$

$$+ \frac{\gamma}{N_3 N_6} \sum_{(x_3, u) \in X_3} \sum_{(x_6, \cdot) \in X_6} [R(x_3, u, x_6) + \gamma\phi(x_6) - \phi(x_3)]$$

$$- \frac{\gamma^2}{N_4 N_5} \sum_{(x_4, u) \in X_4} \sum_{(x_5, \cdot) \in X_5} [R(x_4, u, x_5) + \gamma\phi(x_5) - \phi(x_4)].$$

Rearranging terms, the above equation can be written as:

$$\hat{C}_{SRRD}(R')(s, a, s') = \hat{C}_{SRRD}(R)(s, a, s') + \phi_{residuals},$$

where:

$$\phi_{residuals} = \gamma\phi(s') - \phi(s) + \frac{\gamma}{N_1} \sum_{(x_1, u) \in X_1} [\gamma\phi(x_1) - \phi(s')] - \frac{1}{N_2} \sum_{(x_2, u) \in X_2} [\gamma\phi(x_2) - \phi(s)]$$

$$- \frac{\gamma}{N_3 N_4} \sum_{(x_3, u) \in X_3} \sum_{(x_4, \cdot) \in X_4} [\gamma\phi(x_4) - \phi(x_3)] + \frac{\gamma^2}{N_1 N_5} \sum_{(x_1, u) \in X_1} \sum_{(x_5, \cdot) \in X_5} [\gamma\phi(x_5) - \phi(x_1)]$$

$$- \frac{\gamma}{N_2 N_6} \sum_{(x_2, u) \in X_2} \sum_{(x_6, \cdot) \in X_6} [\gamma\phi(x_6) - \phi(x_2)] + \frac{\gamma}{N_3 N_6} \sum_{(x_3, u) \in X_3} \sum_{(x_6, \cdot) \in X_6} [\gamma\phi(x_6) - \phi(x_3)]$$

$$- \frac{\gamma^2}{N_4 N_5} \sum_{(x_4, u) \in X_4} \sum_{(x_5, \cdot) \in X_5} [\gamma\phi(x_5) - \phi(x_4)]. \tag{40}$$

Applying Lemma 1 to Equation 40 and simplifying terms, we get:

$$\phi_{residuals} = \gamma\phi(s') - \phi(s) + \frac{\gamma^2}{N_1}\sum_{(x_1,u)\in X_1}[\phi(x_1)] - \gamma\phi(s') - \frac{\gamma}{N_2}\sum_{(x_2,u)\in X_2}[\phi(x_2)] + \phi(s)$$

$$- \frac{\gamma^2}{N_4}\sum_{(x_4,u)\in X_4}[\phi(x_4)] + \frac{\gamma}{N_3}\sum_{(x_3,u)\in X_3}[\phi(x_3)] + \frac{\gamma^3}{N_5}\sum_{(x_5,u)\in X_5}[\phi(x_5)] - \frac{\gamma^2}{N_1}\sum_{(x_1,u)\in X_1}[\phi(x_1)]$$

$$- \frac{\gamma^2}{N_6}\sum_{(x_6,u)\in X_6}[\phi(x_6)] + \frac{\gamma}{N_2}\sum_{(x_2,u)\in X_2}[\phi(x_2)] + \frac{\gamma^2}{N_6}\sum_{(x_6,u)\in X_6}[\phi(x_6)] - \frac{\gamma}{N_3}\sum_{(x_3,u)\in X_3}[\phi(x_3)]$$

$$- \frac{\gamma^3}{N_5}\sum_{(x_5,u)\in X_5}[\phi(x_5)] + \frac{\gamma^2}{N_4}\sum_{(x_4,u)\in X_4}[\phi(x_4)] = 0$$

Therefore: $\hat{C}_{SRRD}(R') = \hat{C}_{SRRD}(R)$. $\qquad\square$

### A.6 Example: SRRD State Definitions

Figure 4 illustrates a transition graph from a reward sample with 10 states $S^{\mathcal{D}} = \{x_0, ..., x_9\}$, and a single action $A^{\mathcal{D}} = \{a_1\}$. This example illustrates how the sets $\{S_1, ..., S_6\}$ are defined in SRRD, along with their relationships: $(S_1 \subseteq S_4)$ and $(S_2 \subseteq S_3)$, which contribute to SRRD's robustness to unsampled transitions.

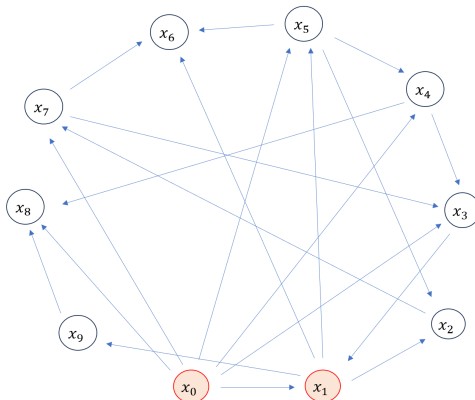

Figure 4: A transition graph with 10 states $\{x_0, ...x_9\}$, and a single action $\{a_1\}$. State subsets are defined based on the transition: $(x_0, a_1, x_1)$.

The Sparsity Resilient Canonically Shaped Reward is given by:

$$C_{SRRD}(R)(s,a,s') = R(s,a,s') + \mathbb{E}[\gamma R(s', A, S_1) - R(s, A, S_2) - \gamma R(S_3, A, S_4) + \gamma^2 R(S_1, A, S_5)$$
$$- \gamma R(S_2, A, S_6) + \gamma R(S_3, A, S_6) - \gamma^2 R(S_4, A, S_5)],$$

where: $S_1$ are subsequent states to $s'$, and $S_2$ are subsequent non-terminal states to $s$. $S_3$ encompasses all initial states from all transitions; $S_4$, $S_5$, and $S_6$ are subsequent states to $S_3$, $S_1$ and $S_2$, respectively.

**Following the SRRD definition, the states in Figure 4, are defined as follows:**

$s$: state $x_0$.

$s'$: state $x_1$.

$S_1$ (subsequent to $s'$): $\{x_2, x_5, x_6, x_9\}$.

$S_2$ (subsequent non-terminal states to $s$): $\{x_1, x_3, x_4, x_5, x_7\}$.

$S_3$ (initial states from all transitions): $\{x_0, x_1, x_2, x_3, x_4, x_5, x_7, x_9\}$.     terminal states $x_6$ and $x_8$ not included

$S_4$ (subsequent states to $S_3$): $\{x_1, x_2, x_3, x_4, x_5, x_6, x_7, x_8, x_9\}$.     starting state $x_0$ not included

$S_5$ (subsequent states to $S_1$): $\{x_2, x_4, x_6, x_7, x_8\}$

$S_6$ (subsequent states to $S_2$): $\{x_1, x_2, x_3, x_4, x_5, x_6, x_8, x_9\}$

**Transition Relationships (see Section 4 for reference)**

**1.** $S_1 \subseteq S_4$, therefore, $(S_1, A, S_5) \subseteq (S_4, A, S_5)$.

**2.** $S_2 \subseteq S_3$, therefore, $(S_2, A, S_6) \subseteq (S_3, A, S_6)$.

## A.7 Pseudometric Equivalence Under Full Coverage

**Proposition 3.** *Given the state subset $S \subseteq \mathcal{S}$, the SRRD, DARD, and EPIC canonical rewards are equivalent under full coverage when: $S = S_1 = ... = S_6$ for SRRD; $S = S' = S''$ for DARD, and $S = S'$ for EPIC.*

*Proof.*

$$C_{EPIC}(R)(s, a, s') = R(s, a, s') + \mathbb{E}[\gamma R(s', A, S') - R(s, A, S') - \gamma R(S, A, S')]$$
$$C_{DARD}(R)(s, a, s') = R(s, a, s') + \mathbb{E}[\gamma R(s', A, S'') - R(s, A, S') - \gamma R(S', A, S'')]$$
$$C_{SRRD}(R)(s, a, s') = R(s, a, s') + \mathbb{E}[\gamma R(s', A, S_1) - R(s, A, S_2) - \gamma R(S_3, A, S_4)$$
$$+ \gamma^2 R(S_1, A, S_5) - \gamma R(S_2, A, S_6) + \gamma R(S_3, A, S_6) - \gamma^2 R(S_4, A, S_5)]$$

Under full coverage, every state $s \in S$ transitions to every $s' \in S$ under every action $a \in A$. Consequently, all relative subsets are equal: $S = S' = S'' = S_1 = S_2 = S_3 = S_4 = S_5 = S_6$, such that:

$$C_{EPIC} = C_{DARD} = C_{SRRD} = R(s, a, s') + \mathbb{E}[\gamma R(s', A, S) - R(s, A, S) - \gamma R(S, A, S)]. \tag{41}$$

$\square$

## A.8 Repeated Canonicalization Under Full Coverage

**Proposition 4.** *Given the state subset $S \subseteq \mathcal{S}$, under full coverage when: $S = S_1 = ... = S_6$ for SRRD; $S = S' = S''$ for DARD, and $S = S'$ for EPIC. Then: $C_{SRRD}$, $C_{EPIC}$, and $C_{DARD}$ cannot be further canonicalized.*

*Proof.* From Proposition 3, we showed that under full coverage, $C_S = C_{EPIC} = C_{DARD} = C_{SRRD}$. Applying $C_S$ to canonicalize a previously canonicalized reward we get:

$$C_S[C_S(R)(s,a,s')] = C_S[R(s,a,s') + \mathbb{E}[\gamma R(s',A,S) - R(s,A,S) - \gamma R(S,A,S)]]$$

$$= C_S(R)(s,a,s') + \gamma\mathbb{E}[C_S(R(s',A,S))] - \mathbb{E}[C_S(R(s,A,S))] - \gamma\mathbb{E}[C_S(R(S,A,S))]$$

$$\begin{aligned}
= \; & C_S(R)(s,a,s') \\
& + \gamma\mathbb{E}[R(s',A,S) + \mathbb{E}[\gamma R(S,A,S) - R(s',A,S) - \gamma R(S,A,S)]] \\
& - \mathbb{E}[R(s,A,S) + \mathbb{E}[\gamma R(S,A,S) - R(s,A,S) - \gamma R(S,A,S)]] \\
& - \gamma\mathbb{E}[R(S,A,S) + \mathbb{E}[\gamma R(S,A,S) - R(S,A,S) - \gamma R(S,A,S)]]
\end{aligned}$$

$$\begin{aligned}
= \; & C_S(R)(s,a,s') \\
& + \gamma\mathbb{E}[R(s',A,S) - R(s',A,S)] + \mathbb{E}[\gamma R(S,A,S) - \gamma R(S,A,S)] \\
& - \mathbb{E}[R(s,A,S) - R(s,A,S)] + \mathbb{E}[\gamma R(S,A,S) - \gamma R(S,A,S)] \\
& - \gamma\mathbb{E}[R(S,A,S) - R(S,A,S)] + \mathbb{E}[\gamma R(S,A,S) - \gamma R(S,A,S)]
\end{aligned}$$

$$= \quad C_S(R)(s,a,s')$$

$\square$

## A.9 Regret Bound

In this section, we establish a regret bound in terms of the SRRD distance. The procedure for the analysis is adapted from the related work on EPIC by Gleave et al. (2021). Given reward functions $R_A$ and $R_B$ and their optimal policies $\pi_A^*$ and $\pi_B^*$, we show that the regret of using policy $\pi_B^*$ instead of a policy $\pi_A^*$ is bounded by a function of $D_{\text{SRRD}}(R_A, R_B)$. We also show that as $D_{\text{SRRD}}(R_A, R_B) \to 0$, the regret approaches 0 suggesting that $\pi_A^* \approx \pi_B^*$. The concept of regret bounds is important as it shows that differences in $D_{\text{SRRD}}$ reflect differences between the optimal policies induced by the input rewards.

For our analysis, we will use the following Lemmas:

**Lemma 2.** *Let $f \in \mathbb{R}^n$ be a vector of real numbers and let $f_i \in \mathbb{R}^k$ be a subvector formed by selecting a subset of the entries of $f$. Then:*

$$||f_i||_2 \leq ||f||_2 \tag{42}$$

*Proof.* Suppose $f$ has $n$ elements and $f_i$ has $k$ elements. Since $f_i \subseteq f$, every element in $f_i$ is also in $f$, and $k \leq n$. Therefore, $\sum f^2 \geq \sum f_i^2$ such that: $||f_i||_2 \leq ||f||_2$. $\square$

**Lemma 3.** *Let $R_A, R_B : \mathcal{S} \times \mathcal{A} \times \mathcal{S} \to \mathbb{R}$ be reward functions with corresponding optimal policies $\pi_A^*$ and $\pi_B^*$. Let $D_\pi(t, s_t, a_t, s_{t+1})$ denote the distribution over trajectories that policy $\pi$ induces at time step $t$. Let $\mathcal{D}(s,a,s')$ be the coverage distribution over transitions. Suppose that there exists some $K > 0$ such that $K\mathcal{D}(s_t, a_t, s_{t+1}) \geq D_\pi(t, s_t, a_t, s_{t+1})$ for all time steps $t \in \mathbb{N}$, triples $s_t, a_t, s_{t+1} \in \mathcal{S} \times \mathcal{A} \times \mathcal{S}$ and policies $\pi \in \{\pi_A^*, \pi_B^*\}$. Then the regret under $R_A$ from executing $\pi_B^*$ optimal for $R_B$ instead of $\pi_A^*$ is at most:*

$$G_{R_A}(\pi_A^*) - G_{R_A}(\pi_B^*) \leq \frac{2K}{1-\gamma} D_{L_1, \mathcal{D}}(R_A, R_B).$$

*where $D_{L_1, \mathcal{D}}$ is a pseudometric in $L_1$ space, and $G_R(\pi)$ resembles the return of $R$ under a policy $\pi$.*

*Proof.* See Gleave et al. (2021) Lemma A.11. $\square$

**Lemma 4.** *Let $R_A, R_B : \mathcal{S} \times \mathcal{A} \times \mathcal{S} \to \mathbb{R}$ be reward functions. Let $\pi_A^*$ and $\pi_B^*$ be policies optimal for reward functions $R_A$ and $R_B$. Suppose the regret under the standardized reward $R_A^S$ from executing $\pi_B^*$ instead of $\pi_A^*$ is upper bounded by some $U \in \mathbb{R}$:*

$$G_{R_A^S}(\pi_A^*) - G_{R_A^S}(\pi_B^*) \leq U. \tag{43}$$

*Assuming that $S_3$ is identically distributed to $S_4$ in $C_{SRRD}$, then the regret is bounded by:*

$$G_{R_A}(\pi_A^*) - G_{R_A}(\pi_B^*) \leq 8U\|R_A\|_2. \tag{44}$$

*Proof.* Following Gleave et al. (2021), we can express the standardized reward as:

$$R^S = \frac{C_{SRRD}(R)}{\|C_{SRRD}(R)\|_2}, \tag{45}$$

In $C_{SRRD}$ (Equation 11), the states $S_1$ and $S_5$ depend on $s'$, while $S_2$ and $S_6$ depend on $s$. Assuming that $S_3$ is identically distributed to $S_4$, we can see that $C_{SRRD}$ is a potential function, where $\phi(s') = \mathbb{E}[R(s', A, S_1)] + \gamma R(S_1, A, S_5) - \gamma R(S_4, A, S_5)]$, and $\phi(s) = \mathbb{E}[R(s, A, S_2)] + \gamma R(S_2, A, S_6) - \gamma R(S_3, A, S_6)]$. Therefore, $C_{SRRD}$ is simply $R$ shaped by some potential $\Phi$, such that:

$$G_{C_{SRRD}(R)}(\pi) = G_R(\pi) - \mathbb{E}_{s_0 \sim d_0}[\Phi(s_0)]$$

Therefore, we can write:

$$G_{R^S}(\pi) = \frac{1}{\|C_{SRRD}(R)\|_2} G_{C_{SRRD}(R)}(\pi) = \frac{1}{\|C_{SRRD}(R)\|_2}(G_R(\pi) - \mathbb{E}_{s_0 \sim d_0}[\Phi(s_0)]), \tag{46}$$

where, $s_0$ depends only on the initial state distribution $d_0$, but not $\pi$. Applying Equation 46 to $\pi_A^*$ and $\pi_B^*$:

$$G_{R^S}(\pi_A^*) - G_{R^S}(\pi_B^*) = \frac{1}{\|C_{SRRD}(R_A)\|_2}(G_{R_A}(\pi_A^*) - G_{R_A}(\pi_B^*)). \tag{47}$$

Combining Equation 47 and 43:

$$G_{R_A}(\pi_A^*) - G_{R_A}(\pi_B^*) \leq U\|C_{SRRD}(R_A)\|_2. \tag{48}$$

We now bound $\|C_{SRRD}(R_A)\|_2$ in terms of $\|R_A\|_2$. The SRRD canonical reward is expressed as:

$$C_{SRRD}(R)(s, a, s') = R(s, a, s') + \mathbb{E}[\gamma R(s', A, S_1) - R(s, A, S_2) - \gamma R(S_3, A, S_4)$$
$$+ \gamma^2 R(S_1, A, S_5) - \gamma R(S_2, A, S_6) + \gamma R(S_3, A, S_6) - \gamma^2 R(S_4, A, S_5)]$$

Now, using the triangular inequality rule on the $L_2$ distance, and linearity of expectation:

$$\|C_{SRRD}(R)(s, a, s')\|_2 \leq \|R(s, a, s')\|_2 + \mathbb{E}[\gamma\|R(s', A, S_1)\|_2 + \| - R(s, A, S_2)\|_2 + \gamma\| - R(S_3, A, S_4)\|_2$$
$$+ \gamma^2\|R(S_1, A, S_5)\|_2 + \gamma\| - R(S_2, A, S_6)\|_2 + \gamma\|R(S_3, A, S_6)\|_2 + \gamma^2\| - R(S_4, A, S_5)\|_2]$$

Using Lemma 2, the $L_2$ norm of each reward subspace is such that:

$$\|R(S_i, A_j, S_k)\|_2 \leq \|R(S, A, S')\|_2 = \|R\|_2. \tag{49}$$

Therefore,

$$\|C_{SRRD}(R)(s, a, s')\|_2 \leq 8\|R\|_2 \tag{50}$$

Combining Equation 50 and 48 we get:

$$G_{R_A}(\pi_A^*) - G_{R_A}(\pi_B^*) \leq 8U\|R_A\|_2.$$

$\square$

**Lemma 5.** *Given a reward function $R : S \times A \times S \to \mathbb{R}$, then: $\mathbb{E}[C_{SRRD}(R)(S, A, S')] = 0$, if $S$ and $S'$ are identically distributed.*

*Proof.* Applying the transitions $(S, A, S')$ into $C_{SRRD}$ (Equation 11), $S' = S_4 = S_1 = S_2 = S_5 = S_6$, and $S = S_3$, such that:

$$C_{SRRD}(R)(S, A, S') = R(S, A, S') + \mathbb{E}[\gamma R(S', A, S') - R(S, A, S') - \gamma R(S, A, S')]$$

Therefore:

$$\mathbb{E}[C_{SRRD}(R)(S, A, S')] = \mathbb{E}[R(S, A, S') + \mathbb{E}[\gamma R(S', A, S') - R(S, A, S') - \gamma R(S, A, S')]$$

if $S$ is identically distributed to $S'$, then $\mathbb{E}[R(S', A, S')] = \mathbb{E}[R(S, A, S')]$, hence:

$$\mathbb{E}[C_{SRRD}(R)(S, A, S')] = \mathbb{E}[R(S, A, S') + \mathbb{E}[\gamma R(S, A, S') - R(S, A, S') - \gamma R(S, A, S')] = 0$$

$\square$

**Theorem 2.** *Let $R_A, R_B : S \times A \times S \to \mathbb{R}$ be reward functions with respective optimal policies, $\pi_A^*, \pi_B^*$. Let $\gamma$ be a discount factor, $D_\pi(t, s_t, a_t, s_{t+1})$ be the distribution over the transitions induced by policy $\pi$ at time $t$, and $\mathcal{D}(s, a, s')$ be the coverage distribution. Suppose there exists $K > 0$ such that $K\mathcal{D}(s_t, a_t, s_{t+1}) \geq D_\pi(t, s_t, a_t, s_{t+1})$ for all times $t \in \mathbb{N}$, triples $(s_t, a_t, s_{t+1}) \in S \times A \times S$ and policies $\pi \in \{\pi_A^*, \pi_B^*\}$. Then the regret under $R_A$ from executing $\pi_B^*$ instead of $\pi_A^*$ is at most:*

$$G_{R_A}(\pi_A^*) - G_{R_A}(\pi_B^*) \leq 32K\|R_A\|_2(1-\gamma)^{-1}D_{SRRD}(R_A, R_B),$$

*where $G_R(\pi)$ is the return of policy $\pi$ under reward $R$.*

*Proof.* Since $C_{SRRD}$ is zero-mean centered for transition inputs $(S, A, S')$ (see Lemma 5), from Gleave et al. (2021) [A.4], it follows that:

$$D_{\text{SRRD}}(R_A, R_B) = \frac{1}{2}\left\|R_A^S(S, A, S') - R_B^S(S, A, S')\right\|_2^2. \tag{51}$$

From Lemma A.10 in Gleave et al. (2021), the $L^1$ norm of a function is upper bounded by its $L^2$ norm on a probability space, such that:

$$D_{L_1, \mathcal{D}}(R_A^S, R_B^S) = \left\|R_A^S(S, A, S') - R_B^S(S, A, S')\right\|_1 \leq 2D_{\text{SRRD}}(R_A, R_B). \tag{52}$$

Combining Lemma 3 and Equation 52:

$$G_{R_A^{SRRD}}(\pi_A^*) - G_{R_A^{SRRD}}(\pi_B^*) \leq \frac{2K}{1-\gamma}D_{L_1, \mathcal{D}}(R_A^{SRRD}, R_B^{SRRD}) \leq \frac{4K}{1-\gamma}D_{\text{SRRD}}(R_A, R_B). \tag{53}$$

Applying Lemma 4, we get:

$$G_{R_A}(\pi_A^*) - G_{R_A}(\pi_B^*) \leq \frac{32K\|R_A\|_2}{1-\gamma}D_{\text{SRRD}}(R_A, R_B). \tag{54}$$

$\square$

As shown, when $D_{\text{SRRD}} \to 0$, the regret: $G_{R_A}(\pi_A^*) - G_{R_A}(\pi_B^*) \to 0$

### A.10 Generalized SRRD Extensions

We derive a generalized form of SRRD by recursively eliminating shaping residuals via higher-order terms.

1. To create SRRD, the first step is to adopt the desirable characteristics from both DARD and EPIC (refer to Section 4), and derive $C_1$ as follows:

$$C_1(R)(s,a,s') = R(s,a,s') + \mathbb{E}[\gamma R(s',A,S_1) - R(s,A,S_2) - \gamma R(S_3,A,S_4)].$$

   $C_1$ yields the residual shaping term: $\phi_{res1} = \mathbb{E}[\gamma^2\phi(S_1) - \gamma^2\phi(S_4) + \gamma\phi(S_3) - \gamma\phi(S_2)]$.

2. To cancel $\mathbb{E}[\phi(S_i)]$, $\forall_i \in \{1,...,4\}$, we add rewards $R(S_i,A,k_i^1)$ to induce potentials $\gamma\phi(k_i^1) - \phi(S_i)$, which results in $C_2$:

$$\begin{aligned}
C_2(R)(s,a,s') = R(s,a,s') + \mathbb{E}[&\gamma R(s',A,S_1) - R(s,A,S_2) - \gamma R(S_3,A,S_4) \\
&+ \gamma^2 R(S_1,A,k_1^1) - \gamma^2 R(S_4,A,k_4^1) + \gamma R(S_3,A,k_3^1) - \gamma R(S_2,A,k_2^1)].
\end{aligned}$$

   $C_2$ yields the residual shaping: $\phi_{res2} = \mathbb{E}[\gamma^3\phi(k_1^1) - \gamma^3\phi(k_4^1) + \gamma^2\phi(k_3^1) - \gamma^2\phi(k_2^1)]$.

3. To cancel $\mathbb{E}[\phi(k_i^1)]$, we add rewards $R(k_i^1,A,k_i^2)$ to induce potentials $\gamma\phi(k_i^2) - \phi(k_i^1)$, yielding $C_3$:

$$\begin{aligned}
C_3(R)(s,a,s') = R(s,a,s') + \mathbb{E}[&\gamma R(s',A,S_1) - R(s,A,S_2) - \gamma R(S_3,A,S_4) \\
&+ \gamma^2 R(S_1,A,k_1^1) - \gamma^2 R(S_4,A,k_4^1) + \gamma R(S_3,A,k_3^1) - \gamma R(S_2,A,k_2^1) \\
&+ \gamma^3 R(k_1^1,A,k_1^2) - \gamma^3 R(k_4^1,A,k_4^2) + \gamma^2 R(k_3^1,A,k_3^2) - \gamma^2 R(k_2^1,A,k_2^2)]
\end{aligned}$$

   $C_3$ yields the residual shaping: $\phi_{res3} = \mathbb{E}[\gamma^4\phi(k_1^2) - \gamma^4\phi(k_4^2) + \gamma^3\phi(k_3^2) - \gamma^3\phi(k_2^2)]$.

4. As we can see, this process results in the generalized formula:

$$\begin{aligned}
C_n(R)(s,a,s') = R(s,a,s') + \mathbb{E}[&\gamma R(s',A,S_1) - R(s,A,S_2) - \gamma R(S_3,A,S_4) \\
&+ \gamma^2 R(S_1,A,k_1^1) - \gamma^2 R(S_4,A,k_4^1) + \gamma R(S_3,A,k_3^1) - \gamma R(S_2,A,k_2^1) \\
&+ \gamma^3 R(k_1^1,A,k_1^2) - \gamma^3 R(k_4^1,A,k_4^2) + \gamma^2 R(k_3^1,A,k_3^2) - \gamma^2 R(k_2^1,A,k_2^2) \\
&\qquad\qquad\qquad \cdots \\
&+ \gamma^n R(k_1^{n-2},A,k_1^{n-1}) - \gamma^n R(k_4^{n-2},A,k_4^{n-1}) + \gamma^{n-1} R(k_3^{n-2},A,k_3^{n-1}) \\
&- \gamma^{n-1} R(k_2^{n-2},A,k_2^{n-1})],
\end{aligned}$$

   where, $n \geq 3$. $C_n$ yields the residual shaping:

$$\phi_n = \mathbb{E}[\gamma^{n+1}\phi(k_1^{n-1}) - \gamma^{n+1}\phi(k_4^{n-1}) + \gamma^n\phi(k_3^{n-1}) - \gamma^n\phi(k_2^{n-1})].$$

- Looking at $\phi_n$, as $n$ increases, each residual shaping term is scaled by $\gamma^n$, so their contribution diminishes exponentially with $n$. Therefore, the upper bound magnitude of $\phi_n$ significantly decreases since $0 \leq \gamma < 1$, and each $|\phi(k_i)| \leq M$, where $M$ is the upper bound potential for all distributions $k_i \subseteq S^{\mathcal{D}}$ (see Appendix A.2.2). Therefore, as $n$ approaches infinity, $\phi_n$ approaches 0.

- The advantage of the generalized SRRD form is that $\phi_n$ approaches 0 as $n$ increases. However, computing many $k_i$ sets makes the process expensive and difficult to implement in practice. Therefore, a smaller $n$ is preferable. In SRRD, we choose $n = 2$, then use our intuition to select sets $k_i$, which further reduce the residual shaping.

### A.11 Computational Complexity

The pseudometrics discussed in this paper all utilize a double sampling method that uses the batches: $B_V$ of $N_V$ sampled transitions, and $B_M$ of $N_M$ state-action pairs. The sample-based approximations for the methods have the following computational complexities:

**EPIC Complexity:**

$$C_{EPIC}(R)(s,a,s') = R(s,a,s') + \mathbb{E}[\gamma R(s',A,S') - R(s,A,S') - \gamma R(S,A,S')].$$

- For all transitions in $B_V$, approximating $\mathbb{E}[R(S,A,S')]$ from $B_M$ takes approximately $O(N_M^2)$ complexity since we iterate $B_M$ in a double loop (see Equation 5). However, this expectation is the same for all transitions, hence it can be computed once.

- For each transition $(s,a,s') \in B_V$, the reward expectations, $\mathbb{E}[R(s',A,S')]$ and $\mathbb{E}[R(s,A,S')]$ can be approximated in one iteration through $B_M$, resulting in $O(N_M)$ time complexity per transition. Since the computation varies based on $(s,a,s')$, the overall complexity is $O(N_V N_M)$.

Therefore, the overall complexity for EPIC is $O(\max(N_V N_M, N_M^2))$. When $N_M$ is significantly larger compared to $N_V$, the complexity is approximately $O(N_M^2)$, and if $N_V$ is significantly larger relative to $N_M$, the complexity is approximately $O(N_V N_M)$.

**DARD Complexity:**

$$C_{DARD}(R)(s,a,s') = R(s,a,s') + \mathbb{E}[\gamma R(s',A,S'') - R(s,A,S') - \gamma R(S',A,S'')].$$

- For each transition $(s,a,s') \in B_V$, the reward expectations, $\mathbb{E}[R(s',A,S'')]$ and $\mathbb{E}[R(s,A,S')]$ can be approximated through a single iteration of $B_M$, resulting in $O(N_M)$ time complexity per transition. The reward expectation, $\mathbb{E}[R(S',A,S'')]$ can be computed via a double loop through $B_M$, and it varies based on $(s,a,s')$, resulting in $O(N_M^2)$ time complexity per transition.

Therefore, the overall complexity for DARD is $O(N_V N_M^2)$.

**SRRD Complexity:**

$$\begin{aligned}
C_{SRRD}(R)(s,a,s') = {} & R(s,a,s') + \mathbb{E}[\gamma R(s',A,S_1) - R(s,A,S_2) - \gamma R(S_3,A,S_4) \\
& + \gamma^2 R(S_1,A,S_5) - \gamma R(S_2,A,S_6) + \gamma R(S_3,A,S_6) - \gamma^2 R(S_4,A,S_5)].
\end{aligned}$$

- For all transitions in $B_V$, approximating $\mathbb{E}[R(S_3,A,S_4)]$ from $B_M$ takes approximately $O(N_M^2)$ complexity since we iterate $B_M$ in a double loop (see Equation 18). However, this expectation is the same for all transitions, hence it can be computed once.

- For each transition $(s,a,s') \in B_V$, the reward expectations, $\mathbb{E}[R(s',A,S_1)]$ and $\mathbb{E}[R(s,A,S_2)]$ can be approximated in one iteration through $B_M$, resulting in $O(N_M)$ time complexity per transition. The reward expectations, $\mathbb{E}[R(S_1,A,S_5)]$, $\mathbb{E}[R(S_2,A,S_6)]$, $\mathbb{E}[R(S_3,A,S_6)]$ and $\mathbb{E}[R(S_4,A,S_5)]$, all require double iterations through $B_M$, and they vary based on $(s,a,s')$, hence, they take $O(N_M^2)$ complexity per transition.

Therefore, the overall complexity for SRRD is $O(N_V N_M^2)$.

## B    Additional Considerations

### B.1    Deviations from Potential Shaping

Reward comparison pseudometrics are generally designed to eliminate potential shaping, and in this study, we examine how deviations from potential shaping can affect the performance of these pseudometrics. Within a $20 \times 20$ Gridworld domain, we generate a ground truth ($GT$) polynomial reward function and a corresponding shaped reward function ($SH$), both with full coverage (100% transitions). From $GT$ and $SH$, we sample rewards $R$ and $R'$ using uniform policy rollovers. For both samples, we add additional noise, $N$, with the

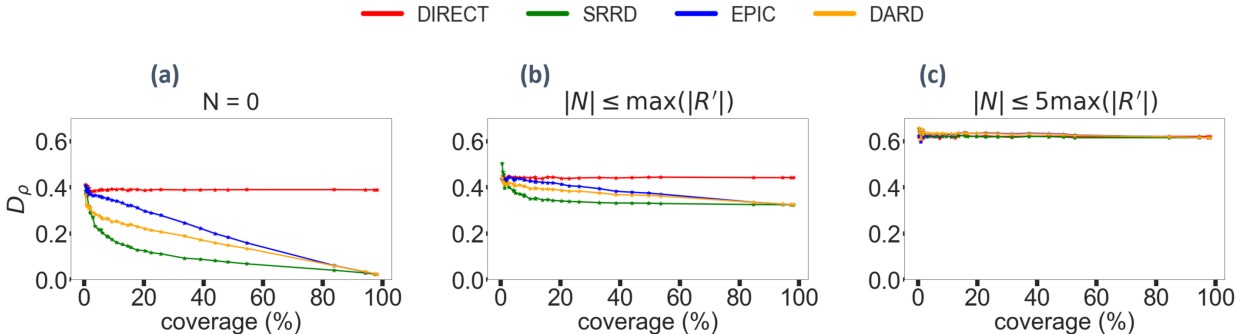

Figure 5: (Non-Potential Shaping Effects): As the severity of randomly generated noise increases from part (a) to (c), rewards deviate more from potential shaping, hence, all the pseudometrics degrade in performance. In the end (part (c)), the pseudometrics perform similarly to DIRECT, showing that canonicalization does not yield any additional advantages when the rewards significantly deviate from potential shaping.

following severity levels: **None:** $N = 0$, **Mild:** $|N| \leq \max(|R'|)$, and **High:** $|N| \leq 5\max(|R'|)$, where $N$ is randomly generated from a uniform distribution within bounds defined by the severity levels. Thus, the updated shaped reward is given by:

$$R'' = R' + N.$$

Figure 5 shows the performance variation of the reward comparison pseudometrics to different noise severity levels. When $N = 0$ (noise free), the difference between SRRD, EPIC, DARD, and DIRECT is the highest, with a performance order: SRRD > DARD > EPIC > DIRECT, which demonstrates SRRD's performance advantage over other pseudometrics under potential shaping. As the impact of $N$ increases, the shaped reward $R''$ becomes almost entirely comprised of noise, and the shaping component significantly deviates from potential shaping. As shown, SRRD's performance gap over other pseudometrics significantly diminishes. At high severity (Figure 5c), SRRD's performance is nearly identical to all other pseudometrics, including DIRECT, which does not involve any canonicalization. In conclusion, these results still demonstrate SRRD's superiority in canonicalizing rewards even with minor random deviations from potential shaping (Figure 5 b). However, as the rewards become non-potentially shaped, all pseudometrics generally become ineffective, performing similarly to non-canonicalized techniques such as DIRECT.

## B.2 Sample-based Approximations using Unbiased Estimates

In this paper, the sample-based SRRD approximation presented adopts a similar form to that of EPIC and DARD, primarily to maintain consistency with well-established methods. This approach utilizes two types of samples: $B_V$, a batch of transitions of size $N_V$, and $B_M$, a batch of state-action pairs of size $N_M$. Each transition in $B_V$ is canonicalized using state-action pairs from $B_M$. The advantage is that if $B_M$ is representative of the joint state-action distribution of the reward function (or sample), canonicalization can be highly effective even with smaller sizes of $B_M$. Since each transition in $B_V$ is canonicalized using state-action pairs from $B_M$, the computational load required during canonicalization can be reduced compared to using all possible transitions. For example, suppose we have a reward function with $10,000$ states and $100$ actions. In this case, the total number of transitions $= 10,000 \times 100 \times 10,000 = 10^{10}$. However, using the sample-based SRRD approximation method, we could generate a sample $B_V$ with $100$ transitions, and another sample $B_M$, with $100$ state-action pairs. Each transition in $B_V$ is then canonicalized using state-action pairs from $B_M$, and the total number of transitions needed is approximately $N_V * N_M{}^2 \approx 100 * 100^2 = 10^6$ transitions (see Appendix A.11); which is way less than the transitions needed in the full computation.

The double-batch sampling approach (using $B_V$ and $B_M$) generally works well in environments where transition sparsity is not a major problem. However, when transition sparsity is a concern, a large fraction of transitions generated from the combination of $B_V$ and $B_M$ might be unsampled, and can have undefined reward values, which can introduce errors in canonicalization. Our proposed pseudometric, SRRD, is much more robust under these conditions compared to both EPIC and DARD, even though it uses a similar sam-

pling form. An alternative approach to the double-batch sampling method is to use unbiased estimates. The unbiased estimate approximations are described as follows:

$$\hat{C}_{EPIC}(R)(s,a,s') \approx R(s,a,s') + \frac{\gamma}{N_1}\sum R(s',u,x') - \frac{1}{N_2}\sum R(s,u,x') - \frac{\gamma}{N_3}\sum R(x,u,x'), \qquad (55)$$

where: $\{(s',u,x')\}$ is the set of sampled transitions that start from the state $s'$, with the total number of transitions equal to $N_1$; $\{(s,u,x')\}$ is the set of sampled transitions that start from the state $s$, with the total number of transitions equal to $N_2$; and $\{(x,u,x')\}$ is the set of all the sampled transitions, with the total number of transitions equal to $N_3$.

$$\hat{C}_{DARD}(R)(s,a,s') \approx R(s,a,s') + \frac{\gamma}{N_1}\sum R(s',u,x'') - \frac{1}{N_2}\sum R(s,u,x') - \frac{\gamma}{N_3}\sum R(x',u,x''), \qquad (56)$$

where: $\{(s',u,x'')\}$, is the set of all sampled transitions that start from the state $s'$, with the total number of transitions equal to $N_1$; $\{(s,u,x')\}$ is the set of all sampled transitions that start from the state $s$, with the total number of transitions equal to $N_2$; and $\{(x',u,x'')\}$ is the set of transitions that start from the subsequent states of $s$ to the subsequent states of $s'$, and have a total number of transitions equal to $N_3$.

$$\begin{aligned}
C_{SRRD}(R)(s,a,s') \approx {}& R(s,a,s') + \frac{\gamma}{N_1}\sum R(s',u,x_1) - \frac{1}{N_2}\sum R(s,u,x_2) - \frac{\gamma}{N_3}\sum R(x_3,u,x_4) \\
& + \frac{\gamma^2}{N_4}\sum R(x_1,u,x_5) - \frac{\gamma}{N_5}\sum R(x_2,u,x_6) + \frac{\gamma}{N_6}\sum R(x_3,u,x_6) - \frac{\gamma^2}{N_7}\sum R(x_4,u,x_5).
\end{aligned} \qquad (57)$$

For SRRD, let $X_1$ represent the set of all subsequent states from $s'$; $X_2$ represent the set of all subsequent states to $s$; and $X_3$ be the set of all initial states for transitions, while $X_4$, $X_5$, and $X_6$ denote the sets of subsequent states from $X_3$, $X_1$, and $X_2$, respectively. For all transitions, the set: $\{(x_3,u,x_4)\}$ contains all sampled transitions where $x_3 \in X_3$ and $x_4 \in X_4$, with the total number of transitions equal to $N_3$. For each sampled transition, $(s,a,s')$, the set:$\{(s',u,x_1)\}$ contains observed transitions starting from $s'$ and ending in $x_1 \in X_1$, with a magnitude $N_1$. $\{(s,u,x_2)\}$ contains observed transitions starting from $s$ and ending in $x_2 \in X_2$, with a magnitude $N_2$. $\{(x_1,u,x_5)\}$ contains observed transitions starting from $x_1 \in X_1$ and ending in $x_5 \in X_5$, with a magnitude $N_4$. $\{(x_2,u,x_6)\}$ contains observed transitions starting from $x_2 \in X_2$ and ending in $x_6 \in X_6$, with a magnitude $N_5$. $\{(x_3,u,x_6)\}$ contains observed transitions starting from $x_3 \in X_3$ and ending in $x_6 \in X_6$, with a magnitude $N_6$. $\{(x_4,u,x_5)\}$ contains observed transitions starting from $x_4 \in X_4$ and ending in $x_5 \in X_5$, with a magnitude $N_7$.

Figure 6 shows results obtained for Experiment 1 (refer to Section 5.1), using sample-based approximations relying on unbiased estimates, instead of the double-batch sampling approach. The obtained results show a similar trend to the results obtained from the double sampling approach, with SRRD still yielding better performance compared to both EPIC and DARD. In general, unbiased estimates tend to be more accurate under transition sparsity, as reflected in Figure 7, showing results for both the double-sampling and the unbiased estimate approaches. Overall, the unbiased estimate approach tends to outperform the double-batch approach, however the double-sampling approach catches up as the level of transition sparsity decreases. The SRRD approach still outperforms both EPIC and SRRD especially when coverage is low ($\leq 20\%$), and it also performs well under the double-batch sampling mechanism, highlighting its robustness in eliminating potential shaping.

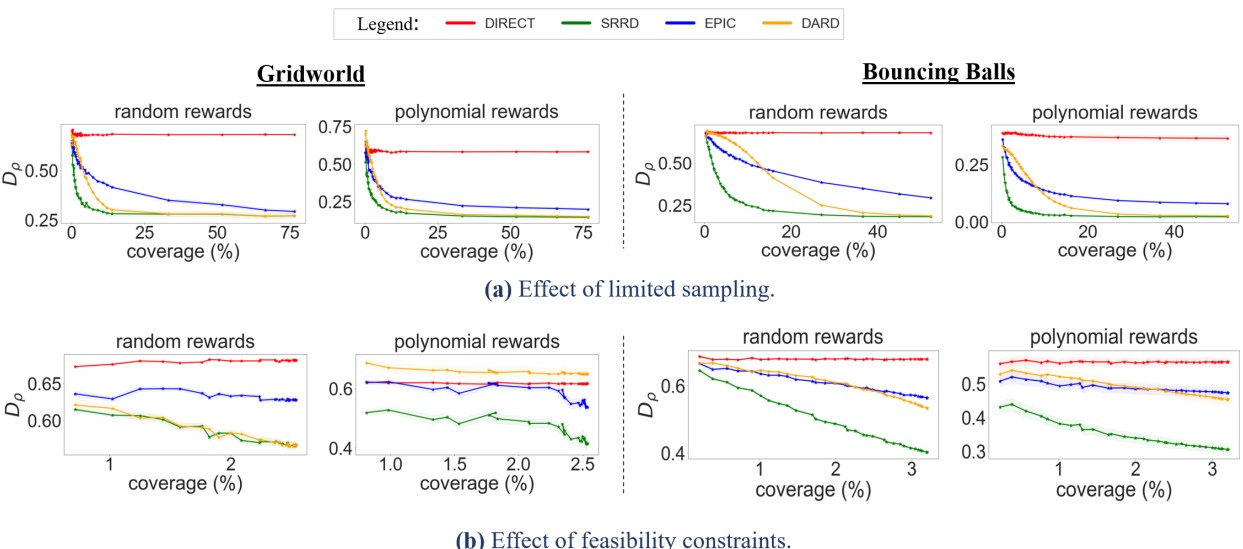

**(a)** Effect of limited sampling.

**(b)** Effect of feasibility constraints.

Figure 6: (Unbiased Estimate Approximations). The figure shows results for Experiment 1 (refer to Section 5.1) which is conducted using unbiased estimates as approximations for the SRRD, DARD and EPIC. As shown, results demonstrate a similar trend as the one obtained when using the double-batch sampling approach reliant on $B_V$ and $B_M$. The goal of the experiment is to compare the effectiveness of reward comparison pseudometrics at identifying the similarity between potentially shaped reward functions under two conditions: (a) limited sampling and (b) feasibility constraints. A more accurate pseudometric yields a Pearson distance $D_\rho$ close to 0, indicating a high degree of similarity between shaped reward functions, while a less accurate pseudometric results in $D_\rho$ close to 1. In **(a)**, EPIC and DARD lag behind SRRD at low coverage due to limited sampling, but their performance gradually improves as coverage increases. In **(b)**, movement restrictions significantly reduce transition coverage, negatively impacting both EPIC and DARD.

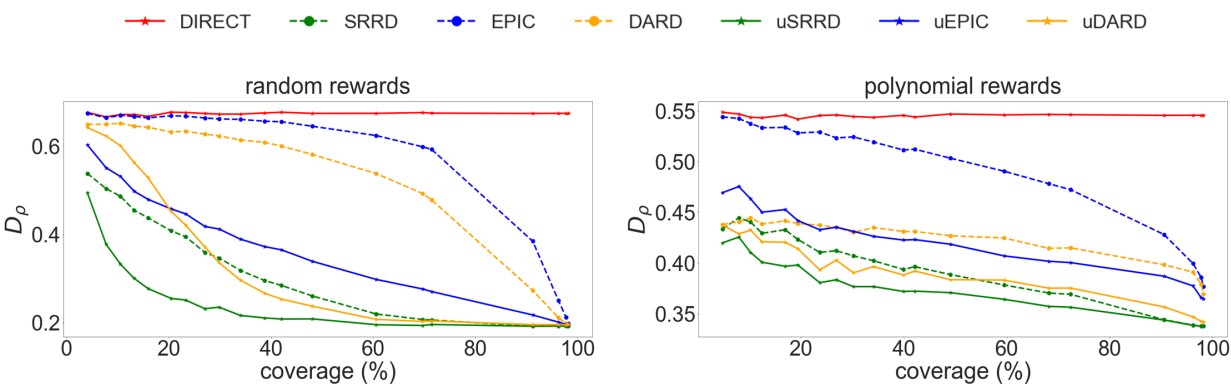

Figure 7: (Unbiased Estimates vs Double-batch Sampling). This figure presents the results for Experiment 1 under limited sampling for the Gridworld domain. It compares the performance of pseudometrics using the double-sampling approach (with batches $B_v$ and $B_N$) versus the unbiased estimate approach. For the unbiased estimates, the results are plotted in bold lines and for the double-sampling approach, they are plotted using broken lines. The initial 'u' (legend) denotes the unbiased estimate methods, for example uSRRD. The objective is to compare the difference in performance between unbiased estimates and the double-batch sampling approach. As shown, the unbiased estimate approaches tend to outperform the double-batch approaches. In both sampling approaches (double-sampling or unbiased estimates), SRRD also outperforms both EPIC and DARD especially when coverage is low ($\leq 20\%$). The SRRD approach also performs well under the double-batch sampling approach, highlighting its robustness at eliminating potential shaping.

### B.3 Inferring Rewards for Unsampled Transitions via Regression

All the canonicalization methods discussed in this paper are susceptible to unsampled transitions. However, $C_{SRRD}$ is more resilient since it mostly relies on forward transitions rather than non-forward transitions (refer to Section 4.1). In this study, instead of relying solely on sampled transitions during canonicalization, we explore the possibility of addressing transition sparsity by generalizing rewards from sampled transitions to unsampled transitions via regression. Using a $20 \times 20$ Bouncing Balls domain, we generate reward samples from a uniform policy and compute the SRRD, DARD, and EPIC distances. To vary coverage, we adjust the number of trajectories generated from policy rollouts, and compute coverage as the fraction of sampled transitions over the total number of transitions, $|S \times A \times S|$. In this experiment, reward samples are only defined for feasible transitions and unfeasible transitions have undefined reward values. However, during canonicalization, the reward values for the unsampled transitions (mostly unfeasible), are inferred via regression. Figure 8 shows the results obtained in this experiment:

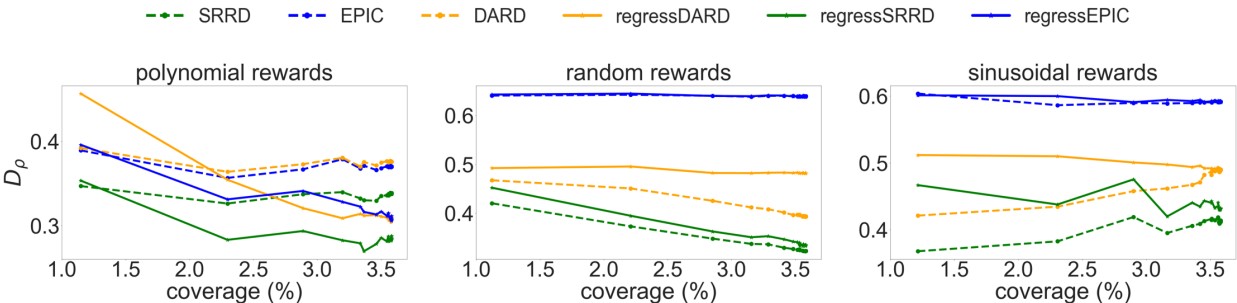

Figure 8: (Incorporating Regressed Rewards) When the relationship between rewards and state-action features is less complex, such as the case with polynomial rewards, learning the regression model is highly effective. Consequently, incorporating rewards that are inferred via regression into canonicalization can be beneficial, resulting in improved performance for *regressDARD*, *regressSRRD*, and *regressEPIC*. However, when the relationship between rewards and state-action features is complex, the learned regression model might struggle to generalize well such as the case for random and sinusoidal rewards.

As shown in Figure 8, the success of incorporating regressed rewards depends on the nature of the reward function. When the reward function is derived as a simple combination of state-action features, learning the regression model is highly effective due to the less complex relationship between state-action features and reward values. This is likely the case for polynomial rewards, where pseudometrics that incorporate regressed rewards, generally outperform the original non-regression-based pseudometrics, especially as the coverage increases. However, for sinusoidal and random reward functions, the relationship between state-action features and rewards is much more complex, making it challenging to effectively learn a highly effective model under transition sparsity. Consequently, the rewards learned via regression may not generalize well, and the original sample-based approximations tend to outperform the regression-based approximations. In general, the regressed SRRD approximation outperformed all other regressed-based approximations. This superiority is attributed to the inherent nature of SRRD which relies more on forward transitions that are likely in-distribution with the reward samples' transition dynamics, than non-forward transitions that are more prone to being out-of-distribution with the dynamics of the reward samples. Therefore, even when incorporating regression, SRRD depends less on the regressed rewards compared to EPIC and DARD, resulting in more accurate predictions. EPIC ideally requires full coverage, hence, under transition sparsity, it relies more heavily on regressed rewards which makes it more susceptible to errors when the regression model cannot generalize well. In this experiment, we compared linear regression, decision trees, and neural networks, and chose linear regression since it yielded the best results in terms of accuracy. This could result from the lack of diverse data, as rewards are only defined for feasible transitions, which are a small subset compared to the total number of transitions that would be needed if no feasibility constraints were imposed.

## B.4 Sensitivity of SRRD to Sampling Policy

A crucial challenge in reward comparison tasks is the fact that reward samples partially represent the true reward function and might not fully capture the structure of the actual reward function. This section examines the performance of SRRD to variations in the policy used to extract reward samples. The experiment is conducted in a $15 \times 15$ Gridworld environment, where an agent's objective is to navigate from an initial state $(0,0)$ to the target state $(14, 14)$. The agent selects actions from the set $A = \{\text{up}, \text{right}, \text{down}, \text{left}\}$. The reward function is derived from predefined expert behaviors using Adversarial Inverse Reinforcement Learning (AIRL). Figure 9 shows a reward function $R_{\text{diagonal}}$, computed for an agent with a diagonally oriented policy where for each state $s$: $\pi_{diagonal}(\text{up}|s) = 0.1$, $\pi_{diagonal}(\text{right}|s) = 0.4$, $\pi_{diagonal}(\text{down}|s) = 0.4$ and $\pi_{diagonal}(\text{left}|s) = 0.1$. The reward for each transition is represented by the triangular directional arrows, for example, the reward from state $(0,0)$ to state $(0,1)$ is 2.5. The reward function is defined exclusively for feasible transitions, and the intensity of rewards is depicted using three three colors: red for high rewards ($> 5$), blue for moderate rewards ($2 - 5$), and light brown for low rewards ($< 2$). In this study, we compare the similarity between two shaped reward samples derived from $R_{\text{diagonal}}$ using a specified sampling policy. We then analyze the variation in $D_{\text{SRRD}}$ based on different sampling policies. The experiment is repeated over 100 trials.

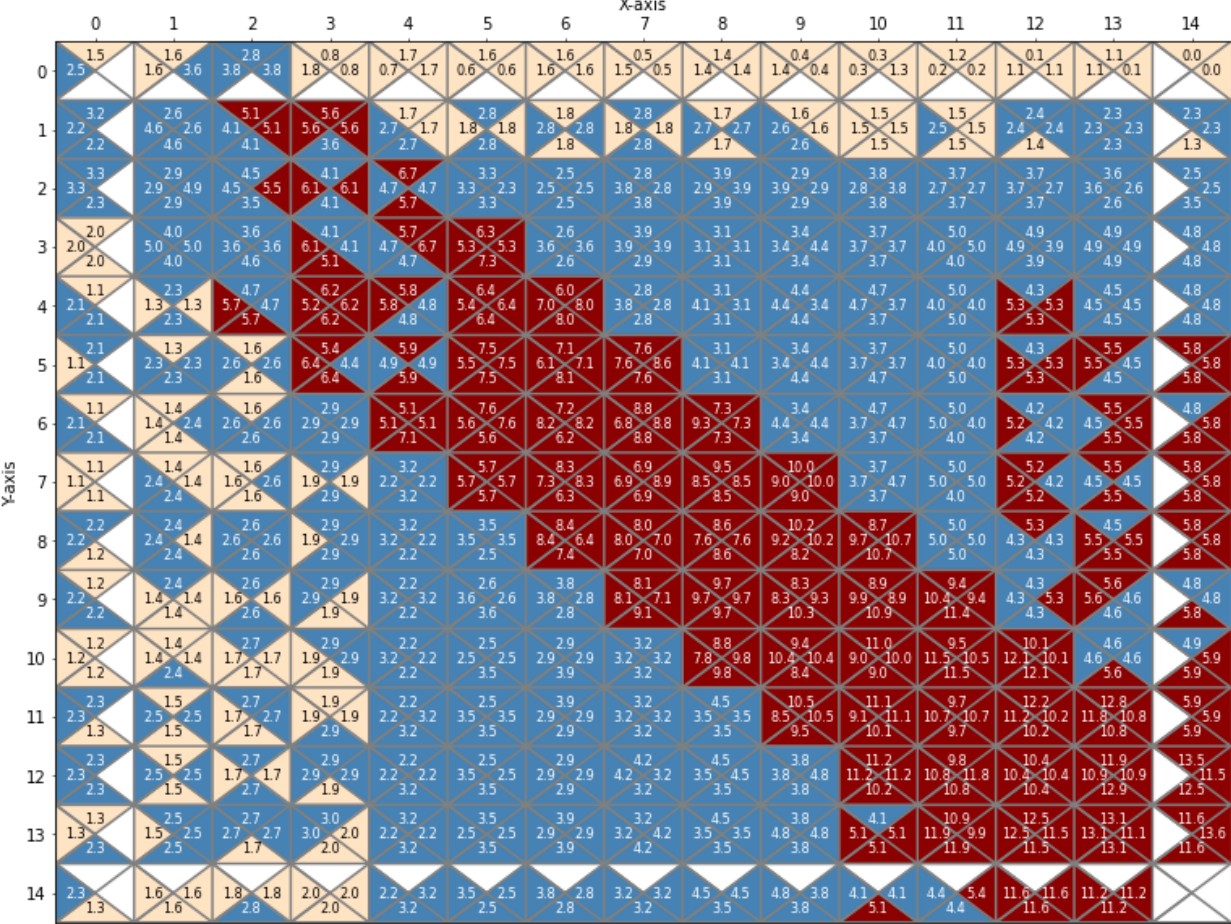

Figure 9: ($R_{\text{diagonal}}$). The reward function is generated via AIRL from an agent that executes the policy $\pi_{diagnoal}$, which favors movement along the grid's main diagonal. High rewards are highlighted in red, moderate rewards in blue, and low rewards in light brown.

Table 2: Sensitivity of SRRD to different reward sampling policies. The coverage is computed as the fraction of sampled transitions, over the total number of feasible transitions.

| | Reward Sampling Policy | | | | | | | |
|---|---|---|---|---|---|---|---|---|
| | $\pi_{\text{uniform}}$ | | $\pi_{\text{diagonal}}$ | | $\pi_{\text{top}}$ | | $\pi_{\text{left}}$ | |
| # of Trajectories | 50 | 100 | 50 | 100 | 50 | 100 | 50 | 100 |
| Coverage (%) | 33.2 | 43.4 | 56.9 | 74.6 | 11.91 | 14.3 | 10.86 | 13.03 |
| $D_{\text{SRRD}}$ for $R_{\text{diagonal}}$ | 0.62 | 0.60 | 0.57 | 0.56 | 0.64 | 0.63 | 0.62 | 0.62 |

Table 2 summarizes results for testing SRRD's sensitivity to different reward sampling policies: $\pi_{\text{diagonal}}$, $\pi_{\text{top}}$ -biased toward the upper segment of the grid; $\pi_{\text{uniform}}$ - a uniform policy across all four directions; and $\pi_{\text{left}}$ - biased towards the left grid segment. As shown, $D_{\text{SRRD}}$ varies based on the reward sampling policy. In $R_{\text{diagonal}}$ when all the feasible transitions are used in reward comparisons, $D_{\text{SRRD}} \approx 0.58$, which is the benchmark value to test the policy variations. This value closely matches the distance values obtained by $\pi_{\text{uniform}}$ and $\pi_{\text{diagonal}}$, which both sample transitions that closely match the actual distribution of the benchmark rewards, $R_{\text{diagonal}}$. However, policies such as $\pi_{\text{top}}$ and $\pi_{\text{left}}$, generally achieve distances that deviate from the benchmark score, since the reward distribution in the reward samples might be less representative of the structure of $R_{\text{diagonal}}$. Overall, $D_{\text{SRRD}}$ varies between: $[0.56 - 0.64]$, while the benchmark distance from $R_{\text{diagonal}}$ is 0.58. We also observe that increasing the number of sampled trajectories from 50 to 100 generally leads to lower SRRD distances. However, this effect is less pronounced in this experiment because the reward function is defined only for feasible transitions, limiting the variability and coverage inherently. In conclusion, SRRD is sensitive to variations in the policies that generate the reward samples, since canonicalization relies on the distribution of sampled transitions, which might not be representative of the actual transition distribution under full coverage. In sampling rewards, it is desirable to ensure that the sample has high coverage and broad support over the distribution of the true reward function. We also performed this study with EPIC and DARD, and they both yield higher distances within the range $[0.63 - 0.7]$.

## B.5 Environments with Infinite or Continuous State and Action Spaces

As previously mentioned in Section 3 and 4, computing the exact reward comparison distances is only feasible in small environments with finite discrete states and actions, where all the possible transitions between states and actions can be enumerated. For complex environments with infinite or continuous states and actions, computing the exact values for $D_{\text{EPIC}}$, $D_{\text{DARD}}$ and $D_{\text{SRRD}}$ becomes impractical, hence, sample-based approximation methods have been developed. These approximations take transition inputs composed of discrete states and actions, and when applied to continuous environments, it is essential to discretize the state and action observations from the continuous signals as demonstrated in prior works (Gleave et al., 2021; Wulfe et al., 2022). In our experiments, these approximations are necessary in environments such as Robomimic, MIMIC-IV, Drone Combat Scenario and StarCraft II. For StarCraft II for example, while not necessarily continuous in a strict sense, the environment operates in real-time with partial observability, multiagent decision-making, durative actions, and asynchronous, parallel action execution. The game updates at approximately 16 times per second, and it effectively has infinite states and actions. These fluid, real-time interactions align it more with continuous decision-making scenarios, and to manage the complexity, we perform feature preprocessing to discretize and cluster state and action observations before applying the sample-based approximations.

An intriguing area for future research is the integration of function approximation to generalize reward canonicalization to reward functions represented as neural networks. While function approximation might not be necessary for straightforward reward comparisons—where the goal is to retrieve a similarity distance—they become essential in applications requiring the canonicalization of the entire reward functions, such as standardizing rewards during IRL for example. Our initial proposed approach involves training a neural network to predict canonical rewards based on batches of transition observations. In the training process, the canonicalized rewards for the batch can be approximated via sample-based methods, and then the neural

network aims to predict the canonicalized rewards from the input transition batch. The network iteratively learns to predict the canonicalized rewards by minimizing the difference between its predictions and the sample-based approximations.

## C  Experimental Details

### C.1  Experiment 1: Transition Sparsity Pseudocode

---

**Algorithm 1** Analyzing the effect of limited sampling on reward distance

---

**Input**:
  $T$ - list of policy rollout counts,
  $E$ - number of experimental trials under same condition,
  $G$ - grid size,
  RD - list to store reward distances at different coverages,
  MC - maximum coverage $\approx S \times A \times S$.
**Output**: RD

1:  generate GT - ground truth reward, SH - shaped reward from all possible transitions.
2:  **for** rollout$_{\text{count}}$ in $T$ **do**
3:    trial$_{\text{distance}}$, trial$_{\text{coverage}}$ = $list()$, $list()$
4:    **for** trial in $E$ **do**
5:      $B_{gt}, B_{sh} = set(), set()$
6:      generate trajectories $\tau_{gt}$ and $\tau_{sh}$ using uniform policy rollouts.
7:      **for** $(s, a, s') \in \tau_{gt}$ **do**
8:        $B_{gt}$.add($(s, a, s')$)
9:      **end for**
10:     **for** $(s, a, s') \in \tau_{sh}$ **do**
11:       $B_{sh}$.add($(s, a, s')$)
12:     **end for**
13:     for $(s, a, s') \in B_{gt}$ and $(s, a, s') \in B_{sh}$, retrieve $R(s, a, s')$ using GT and $R'(s, a, s')$ using SH, respectively.
14:     coverage = $\frac{|B_{gt} \cup B_{sh}|}{\text{MC}}$
15:     compute $dist(R, R')$ using EPIC, SRRD, DARD, or DIRECT.
16:     trial$_{\text{distance}}$.append($dist(R, R')$)
17:     trial$_{\text{coverage}}$.append($coverage$)
18:    **end for**
19:    $RD$.append([mean(trial$_{\text{coverage}}$), mean(trial$_{\text{distance}}$)])
20:  **end for**

---

### C.2  Experiment 1: Reward Functions

Extrinsic reward functions are manually defined using a combination of state and action features. For the Drone Combat, Montezuma's Revenge, and StarCraft II domains, we use the default game engine scores as the reward function, and for Robomimic, rewards are based on task completion (see Appendix C.2). For the Gridworld and Bouncing Balls domains, in each reward function, the reward values are derived from the decomposition of state and action features, where, $(s_{f1}, ..., s_{fn})$ is from the starting state $s$; $(a_{f1}, ..., a_{fm})$ is from the action $a$; and $(s'_{f1}, ..., s'_{fn})$ is from the subsequent state $s'$. For each unique transition, using randomly generated constants: $\{u_1, ..., u_n\}$ for incoming state features; $\{w_1, ..., w_m\}$ for action features; $\{v_1, ...v_n\}$ for subsequent state features, we create reward models as follows:

- Linear:

$$R(s, a, s') = u_1 s_{f1} + ... + u_n s_{fn} + w_1 a_{f1} + ... + w_m a_{fm} + v_1 s'_{f1} + ... + v_n s'_{fn},$$

- Polynomial:

$$R(s, a, s') = u_1 s_{f1}^{\alpha} + ... + u_n s_{fn}^{\alpha} + w_1 a_{f1}^{\alpha} + ... + w_m a_{fm}^{\alpha} + v_1 s_{f1}'^{\alpha} + ... + v_n s_{fn}'^{\alpha},$$

  where, $\alpha$ is randomly generated from $1 - 10$, denoting the degree of the polynomial.

- Sinusoidal:

$$R(s, a, s') = u_1 sin(s_{f1}) + \cdots + u_n sin(s_{fn}) + w_1 sin(a_{f1}) + \cdots + w_m sin(a_{fm})$$
$$+ v_1 sin(s_{f1}') + \cdots + v_n sin(s_{fn}')$$

- Random

$$R(s, a, s') = \beta,$$

  where, $\beta$ is a randomly generated reward for each given transition.

The same relationships are used to model potential functions, where: $\phi(s) = f(s_{f1}, .., s_{fn})$, and $f$ is the relationship drawn from the set: {polynomial, sinusoidal, linear, random}. For StarCraft II, Drone Combat, and Montezuma's revenge, we used the default game score provided by the game engine as the reward function; and for Robomimic, sparse rewards based on task completion are used. For the StarCraft II domain, this score focuses on the composition of unit and resource features. Since the Drone Combat environment is originally designed for a predator-prey domain, we adapt the score `https://github.com/koulanurag/ma-gym` to essentially work for the Drone Combat scene (i.e instead of a predator being rewarded for eating some prey, the reward is now an ally attacking an enemy).

### C.3    Experiment 1: Parameters

A uniform policy in the Gridworld domain randomly selects one of the four actions, {north, east, south, west}, at each timestep. For the Bouncing Balls domain, it randomly selects an action from the set: {north, north-east, east, east-south, south, south-west, west, west-north, north}. The parameter $\epsilon$ dictates the ratio of times in which random transitions (instead of uniform policy) are executed. Table 3 and Table 4 shows the experimental parameters used in Experiment 1 (Algorithm 1).

Table 3: (Low Coverage): Parameters used to test the variation of coverage for the Gridworld and the Bouncing Balls domain.

| Parameter | Values |
|---|---|
| Rollout Counts, $T$ | $[1, 2, 3, 4, 5, 6, 7, 8, 9, 10, 15, 20, 30, 40, 50, 75, 100, 200, 300, 400, 500, 1000, 2000]$ |
| Epochs, $E$ | 200 |
| Policy, $\pi$ | uniform, $\epsilon = 0.1$ |
| Discount, $\gamma$ | 0.7 |
| Dimensions | $20 \times 20$ |

Table 4: (Feasibility Constraints): Parameters used to test the variation of coverage in the presence of movement restrictions, $\epsilon = 0$, for the Gridworld and Bouncing Balls domain.

| Parameter | Values |
|---|---|
| Rollout Counts, $T$ | $[1, 2, 3, 4, 5, 6, 7, 8, 9, 10, 15, 20, 30, 40, 50, 75, 100, 200, 300, 400, 500, 1000, 2000]$ |
| Epochs, $E$ | 200 |
| Policy, $\pi$ | uniform, $\epsilon = 0$ |
| Discount, $\gamma$ | 0.7 |
| Dimensions | $20 \times 20$ |

## C.4 Transition Sparsity: Additional Results

In both the Gridworld and the Bouncing Balls domains, we did not see much difference in the structure of results between the $10 \times 10$ domain and the $20 \times 20$ domains. Results were fairly consistent in that SRRD tends to outperform DARD and EPIC, and feasibility constraints tend to limit coverage significantly.

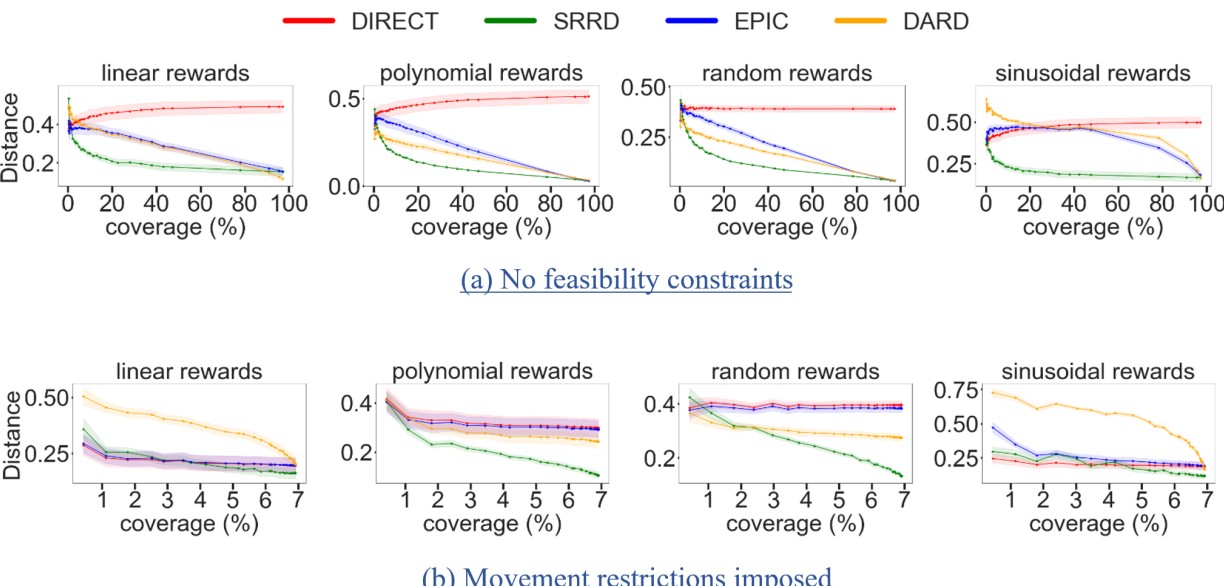

Figure 10: $10 \times 10$ Gridworld: Variation of reward relationships

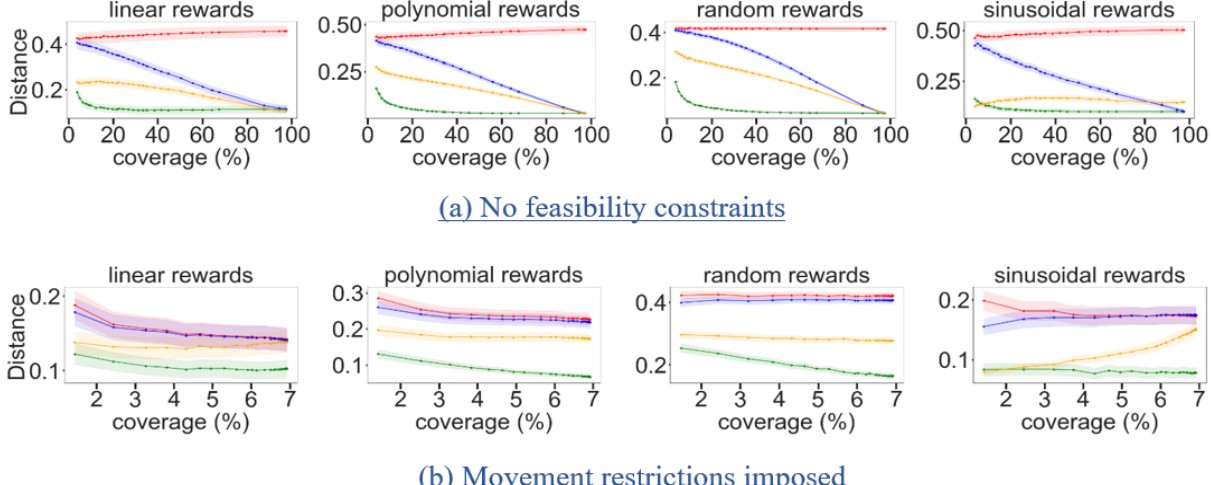

Figure 11: $20 \times 20$ Gridworld: Variation of reward relationships

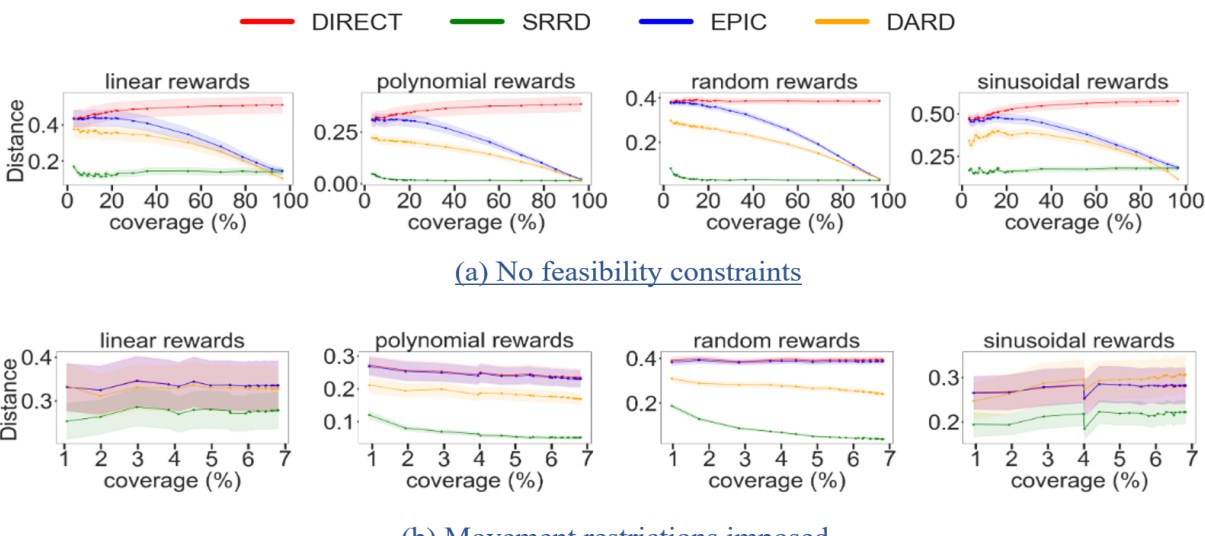

Figure 12: $10 \times 10$ Bouncing Balls: Variation of reward relationships

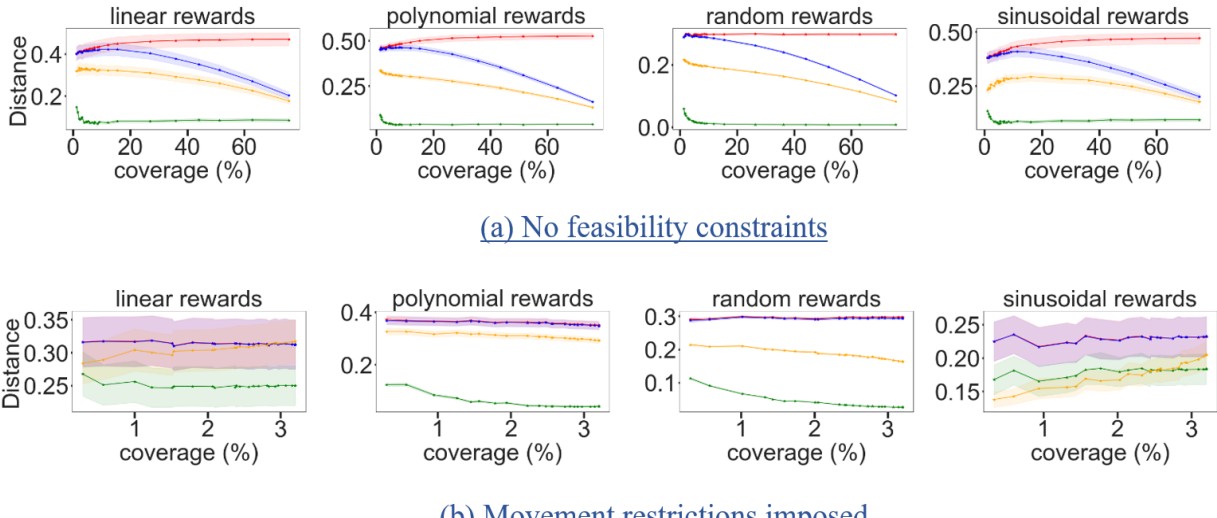

Figure 13: $20 \times 20$ Bouncing Balls: Variation of reward relationships

### C.5 Experiment 2: Reward Classification - Testbeds

**Gridworld:** The Gridworld domain simulates agent movement from a given initial state to a specified terminal state under a static policy. Each state is defined by an $(x, y)$ coordinate where $0 \leq x < N$, and $0 \leq y < M$ implying $|\mathcal{S}| = NM$. For Experiment 2, the action space only consists of four cardinal directions {north, east, south, west}, and to define classes, we use static policies based on the action-selection distribution (out of 100) per state. Table 6 shows the Gridworld parameters used for Experiment 2.

**Bouncing Balls:** The Bouncing Balls domain, adapted from Wulfe et al. (2022), simulates a ball's motion from a starting state to a target state while avoiding randomly mobile obstacles. These obstacles add complexity to the environment since the ball might need to change its strategy to avoid obstacles (at a distance, $d = 3$). Each state is defined by the tuple $(x, y, d)$, where $(x, y)$ indicates the ball's current location, and $d$ indicates the ball's Euclidean distance to the nearest obstacle, such that: $0 \leq x < N$, $0 \leq y < M$, and $d \leq \max(M, N)$. The action space includes eight directions (cardinals and ordinals), with the stochastic parameter $\epsilon$ for choosing random transitions. Table 5 describes the parameters for Experiment 2.

**Drone Combat:** The Drone Combat domain is derived from the multi-agent gym environment, which simulates a predator-prey interaction (Anurag, 2019). We adapt this testbed to simulate a battle between two drone swarms; a blue swarm denoting the ally team; and a red swarm denoting a default AI enemy. The goal is for the blue ally team to defeat the default AI team. This testbed offers discrete actions and states within a fully observable environment, while also offering flexibility for unit creation, and obstacle placement. However, the number of states and actions is still high such that we did not use the testbed in Experiment 1. Each unit (blue and red squares) possesses a distinct set of parameters and possible actions. Each team consists of drones and ships, and the team that wins either destroys the entire drones of the opponent or its ship. This ship adds complexity to the decision-making process of the teams which need to engage with the enemy, as well as safeguard their ships. Each drone is defined by the following attributes: visibility range (VR) - the range a unit can see from its current position (partial observability); health (H) - the number of firings a unit can sustain; movement range(MR) - the maximum distance that a unit can move to; and shot strength(SS) - the probability of a shot hitting its target. These attributes are drawn from the set:

$$U = \{(\text{VR}, \text{H}, \text{MR}, \text{SS}) \mid \text{VR} \in \{1, 3, 5\}, \text{H} \in \{5, 10, 15\}, \text{MR} \in \{1, 2, 3\}, \text{SS} \in \{0.05, 0.1\}\}$$

Table 7 summarizes the parameters used for Experiment 2.

**Robomimic** Robomimic is a framework designed for imitation learning and reinforcement learning in robotic manipulation tasks (Mandlekar et al., 2021). It provides standardized datasets, and environments to facilitate reproducibility in learning from demonstrations. In our experiments, we focus on collected offline datasets, which include human and simulated demonstrations on the following robotic tasks: *Lift* - the robot needs to pick up a cube and lift it off the table; *Can* - the task requires lifting and placing a soda can upright; *Square* - the robot aligns and places a square peg into a hole; *Tool Hang* - requires a robot to use a tool to hang it on a rack; and *Nut Assembly* - a multi-step task where the robot must pick up a nut and screw it onto a bolt. The goal in Experiment 2 is to distinguish between a single human expert, machine generated, and mixed human agent behaviors, using a k-NN classifier based on reward pseudometrics. For each agent type, we collect 200 trajectories, and utilize the extrinsic sparse rewards based on task completion, as well as IRL computed rewards. The details about the agent types, and data collection are available at: `https://robomimic.github.io/docs/datasets/robomimic_v0.1.html`.

**Montezuma's Revenge:** Montezuma's Revenge is a classic Atari 2600 game generally used as a benchmark in RL research due to its challenging environment. The game involves controlling an explorer agent, in navigating a series of interconnected rooms in a temple filled with traps, ladders, enemies and collectible items. The game is challenging for RL since its characterized by sparse rewards, complex exploration, long-term dependencies, and high penalty for mistakes. In our experiments, we used a collection of Atari games from human demonstrations (Kurin et al., 2017). We subdivide these games into three subgroups based on game scores: experts ($> 3000$), moderate ($1000 - 3000$) and novice($< 1000$); and sample 200 trajectories for

each agent type. The task objective is to determine the agents' expertise using the reward pseudometrics to classify the agents based on samples of their demonstrated behavior.

**StarCraft II (SC2):** The SC2 domain is a strategy game created by Blizzard Entertainment that features real-time actions on a complex battlefield environment. The game involves planning, strategy, and quick decision-making to control a multi-agent ally team, aiming to defeat a default AI team in a competitive, challenging, and time-sensitive environment. SC2 serves not only as entertainment but also as a platform for professional player competitions. Due to its complexity and the availability of commonly used interactive Python libraries, the SC2 game is widely employed in Reinforcement Learning, serving as a testbed for multi-agent research. The goal of the ally team is to gather resources, and build attacking units that are used to execute a strategic mission to defeat the AI enemy; within an uncharted map, that gets revealed after extensive exploration (introduces partial observability). The sheer size of the map and the multitude of possible actions for each type of unit, as well as the number of units, contribute to the enormity of the action and state spaces. During combat, each ally unit, moves in a decentralized manner and attacks an enemy unit using an assigned cooperative strategy from the set: $C = \{c_1, c_2, c_3, c_4\}$; where $c_1$ - move towards ally closest to enemy's start base; $c_2$ - move towards a random enemy unit; $c_3$ - move towards ally closest to an enemy unit; and $c_4$ - move towards the map's center. We focus on attack-oriented ally units to reduce the state space. Non-attacking units such as pylons are treated as part of the environment. The game state records the number of ally units ($num_{ally}$), and the total number of opponent units ($num_{enemy}$); as well as the central coordinates of the ally and the enemy. The action records the number of ally units attacking the enemy at an instance. Table 8 describes the StarCraft II parameters used for Experiment 2.

**MIMIC IV:** MIMIC-IV is a large, publicly available database of de-identified patients admitted to an emergency department (ED) or intensive care unit(ICU) at the Beth Israel Deaconess Medical Center in Boston, MA (Johnson et al., 2023). The dataset contains records for over $65,000$ patients admitted to an ICU, and over $200,000$ patients admitted to the ED. In our analysis, we model each patient's hospital's visit as a Markov Decision Process, where, each state is a list of assigned diagnoses (a patient may have multiple concurrent diagnoses), and the actions are the procedures and prescriptions provided. For manual rewards, we assign $R = 5 * \gamma^t$ when a patient survives, where $\gamma = 0.9$ is a discount factor (to penalize frequent visits) and $t$ is the trial number of the patient's visit. A manual reward of $-5$ is assigned when the patient dies. We also compute AIRL, Maxent and PTIRL rewards. From the data, because both the states (diagnoses) and actions (prescriptions and procedures) are expressed in natural language, we transform them into numerical vectors using frequency counts of medical keywords (without stop words) from the diagnoses or prescription reports. We then apply standard scaling (mean and variance), and perform PCA to extract numerical features. For the behavioral classification experiment, we manually group patients into 5 categories based on keywords from the first diagnosis record of the patients. These categories are: *diabetes* - has diabetes oriented diagnoses, *kidney* - has kidney oriented diagnoses, *limb* - has limb or skeletal oriented diagnoses, *respiratory* - has respiratory diagnoses, and *substance* - has substance abuse or mental health diagnoses. Our task is then to classify different patient trajectories (sequences of states and actions), based on their reward functions, using the reward pseudometrics. It is important to note that some diseases overlap, hence we used a majority vote to assign the overall category. Since this data is inherently sparse, our final dataset had 409 patients distributed such that we have: $[70, 75, 42, 35, 187]$ patients, corresponding to each disease class, respectively. Each trajectory has a length in the range 30 to 250. Most of the original MIMIC-IV data was not useful for our analysis since patients rarely visit an ICU facility more than once. In the experiments we did not consider age, race or gender in the analysis.

### C.6 Experiment 2: Parameters

The optimal values for $\gamma$ (discount factor) and $k$ (the neighborhood size) are not fixed for each independent trial. Therefore, for hyperparameter selection, we employ a grid search over the set defined by:

$$\{(\gamma, k) : \gamma \in \{0, 0.1, \ldots, 1\}, k \in \{10, 20, \ldots, 100\}\}$$

The agent classes shown describe the policy that an agent takes in each given state. For example, in the Gridworld domain, an agent with a policy [25, 25, 25, 25], randomly selects the cardinal direction to take

from a uniform distribution. For the Drone Combat and StarCraft II domains, the agent behaves based on the combination of the defined attributes.

Table 5: Bouncing Balls Parameters.

| Parameter | Values |
|---|---|
| Agent policies (10 classes) | $[[12, 12, 12, 12, 13, 13, 13, 13], [5, 5, 25, 25, 25, 5, 5, 5],$ $[25, 25, 25, 5, 5, 5, 5, 5], [5, 5, 5, 5, 5, 25, 25, 25], [5, 5, 65, 5, 5, 5, 5, 5],$ $[5, 5, 5, 65, 5, 5, 5, 5], [5, 5, 5, 5, 65, 5, 5, 5], [5, 25, 5, 25, 5, 25, 5, 5],$ $[20, 5, 20, 5, 20, 5, 20, 5], [5, 20, 5, 20, 5, 20, 5, 20]]$ |
| Trajectory sets per policy | 100 |
| Number of obstacles | 5 |
| Distance to obstacle (Manhattan) | 3 |
| Number of comparison trials | 200 |
| Actions | move: {north, north-east, east, east-south, south, south-west, west, west-north} |
| State Dimensions | $20 \times 20 \times 3$ |

Table 6: Gridworld Parameters.

| Parameter | Values |
|---|---|
| Agent policies (10 classes), | $[[25, 25, 25, 25], [5, 5, 5, 85], [85, 5, 5, 5], [5, 85, 5, 5], [5, 5, 85, 5], [5, 15, 30, 55], [55, 30, 15, 5], [15, 5, 55, 30], [5, 55, 30, 15], [15, 30, 5, 55]]$ |
| Trajectory sets per policy, | 100 |
| Number of comparison trials, | 200 |
| Actions, | move: {north, west, south, east} |
| State Dimensions | $20 \times 20$ |

Table 7: Drone Combat Parameters.

| Parameter | Values |
|---|---|
| Agent policies | 10 classes, each consisting of 5 agents. Each agent $x$ has attributes randomly drawn from the set: $U = \{(VR, H, MR, SS) \mid VR \in \{1, 3, 5\}, H \in \{5, 10, 15\}, MR \in \{1, 2, 3\}, SS \in \{0.05, 0.1\}\}$ |
| Trajectory sets per policy | 100 |
| Number of agents per team | 11 (1 ship, 10 drones) |
| Number of comparison trials | 200 |
| Actions, $\alpha$ is the movement range, $1 \leq \alpha \leq 3$ | $\{\{\text{left}^\alpha, \text{up}^\alpha, \text{right}^\alpha, \text{down}^\alpha\}, \text{attack}\}$ |
| Dimensions | $40 \times 25$, with obstacles occupying $\approx 30\%$ of the area |

Table 8: StarCraft II Parameters.

| Parameter | Values |
|---|---|
| Agent policies generated based on resources and strategy | 10 classes, agents attributes randomly chosen from: $U = \{(c, u) \mid c \in \{c_1, c_2, c_3, c_4\}, u \in \{\text{adept, voidray, phoenix, stalker}\}\}.$ |
| Trajectory sets per policy | 100 |
| Comparison trials | 200 |
| Actions | Number of attacking units per unit time |
| State representation | $(\text{num}_{ally}, \text{num}_{enemy}, (x_{ally}, y_{ally}), (x_{enemy}, y_{enemy}))$ |

### C.7 Experiment 2: Inverse Reinforcement Learning (IRL)

Table 9: Reward Learning Parameters Across Domains

| AIRL | MAXENT | PTIRL |
|---|---|---|
| Trajectories/run: 5 | Trajectories/run: 5 | Target Trajectories/run: 5 |
| RL Algorithm: PPO | RL Algorithm: PPO | Non-Target Trajectories/run: 10 |
| Discount ($\gamma$): 0.9 | Discount ($\gamma$): 0.9 | Max Reward Cap: +100 |
| Reward Network - MLP | Reward Network - MLP | |
| Hidden Size: [256, 128] | Hidden Size: [256, 128] | Min Reward Cap: -100 |
| Learning Rate: $10^{-4}$ | Learning Rate: $10^{-4}$ | LP Solver: Cplex |
| Time Steps: $10^5$ | | |
| Generator Batch Size: 2048 | | |
| Discriminator Batch Size: 256 | | |

In Experiment 2, we utilize Inverse Reinforcement Learning (IRL) to compute agent rewards based on demonstrated behavior. Specifically, we employ three IRL algorithms: Maximum Entropy IRL (Maxent-IRL) (Ziebart et al., 2008); Adversarial IRL (Fu et al., 2018); and the Preferential Trajectory IRL (PT-IRL) (Santos et al., 2021). In addition, we compute manual rewards that differ due to potential shaping.

**Maxent IRL** The objective of the Maxent IRL[4] algorithm is to compute a reward function that will generate a policy (learner) $\pi_L$ that matches the feature expectations of the trajectories generated by the expert's policy (demonstrations, assumed to be optimal) $\pi_E$. Formally, this objective can be expressed as:

$$\mathbb{E}_{\pi_L}[\phi(\tau)] = \mathbb{E}_{\pi_E}[\phi(\tau)], \tag{58}$$

where $\phi_\tau$ are trajectory features. $\mathbb{E}_{\pi_k} = \sum_{\tau \in \varphi_k} p_{\pi_k}(\tau) \cdot \phi(\tau)$, where $p_{\pi_k}(\tau)$ is the probability distribution of selecting trajectory $\tau$ from $\pi_k$. The original Maxent-IRL algorithm modeled the relationship between state features and agent rewards as linear, however, recent modifications now incorporate non-linear features via neural networks. To resolve the ambiguity of having multiple optimal policies which can explain an agent's behavior, this algorithm applies the principle of maximum entropy to select rewards yielding a policy with the highest entropy.

**AIRL:** The AIRL algorithm uses generative adversarial networks to train a policy that can mimic the expert's behavior. The IRL problem can be seen as training a generative model over trajectories, such that:

$$\max_w J(w) = \max_w \mathbb{E}_{\tau \sim \mathcal{D}}[\log p_w(\tau)], \tag{59}$$

where $p_{\mathbf{w}}(\tau) \propto p(s_0) \prod_{t=0}^{T-1} P(s_{t+1}|s_t, a_t) e^{\gamma^t . R_w(s_t, a_t)}$ and the parameterized reward is $R_w(s, a)$. Using the gradient of $J(w)$, the **entropy-regularized policy objective** can be shown to reduce to:

$$\max_\pi \mathbb{E}_\pi \left[ \sum_{t=0}^{T} (R_w(s_t, a_t) - log\pi(a_t|s_t)) \right] \tag{60}$$

The discriminator is designed to take the form: $D_w(s, a) = \exp(f_w(s, a))/(\exp(f_w(s, a)) + \pi(a|s))$, and the **training objective** aims to minimize the cross-entropy loss between expert demonstrations and generated samples: $L(w) = \sum_{t=0}^{T} (-\mathbb{E}_{\mathcal{D}}[\log D_w(s_t, a_t)] - \mathbb{E}_{\pi_t}[\log(1 - D_w(s_t, a_t))])$. The policy optimization objective then uses the reward: $R(s, a) = log(D_w(s, a)) - log(1 - D_w(s, a))$.

**PTIRL:** The PTIRL algorithm incorporates multiple agents, each with a set of demonstrated trajectories $\varphi_i$. To compute rewards for each agent, PTIRL considers target and non-target trajectories. Target trajectories are demonstrated trajectories from a target agent, and non-target trajectories are demonstrated

---

[4]Maxent and AIRL implementations adapted from: https://github.com/HumanCompatibleAI/imitation (Gleave et al., 2022)

trajectories from other agents. Denoting $P$ as the probability transition function for all the agents, the linear expected reward for each trajectory $\tau$ is defined as:

$$LER(\tau) = \sum_{k=1}^{m} P(s_k{}', a_k, s_k) \cdot r(s_k{}', a_k, s_k).$$

For each trajectory set, there is a lower bound value $lb(\varphi)$ and an upper bound value $ub(\varphi)$, defined as: $lb(\varphi) = \min_{\tau \in \varphi}(LER(\tau))$ and $ub(\varphi) = \max_{\tau \in \varphi}(LER(\tau))$, respectively. From $lb(\varphi)$ and $ub(\varphi)$, the spread $\delta$ is defined as: $\delta(\varphi_a, \varphi_b) = lb(\varphi_a) - ub(\varphi_b)$. PTIRL defines a preferential ordering between any two trajectories $\varphi_a$ and $\varphi_b$ as a poset, $\prec$, such that if $\varphi_b \prec \varphi_a$, then $\delta(\varphi_a, \varphi_b) > 0$. Given the above definitions, let $\varphi_i$ be the set of target trajectories and $\varphi_{ni}$ the set of non-target trajectories. The PTIRL objective is to compute rewards such that $\varphi_{nt} \prec \varphi_i$, $\delta(\varphi_i, \varphi_{nt}) \geq \alpha$, where $\alpha$ is the minimum threshold for the spread. PTIRL is generally fast because it directly computes rewards via linear optimization.

## C.8 Reward Classification: Significance Tests

Table 10: Experiment 2: Welch's t-tests

| Domain | Rewards | SRRD_vs_DIRECT | | SRRD_vs_EPIC | | SRRD_vs_DARD | |
|---|---|---|---|---|---|---|---|
| | | t-statistic | p-value | t-statistic | p-value | t-statistic | p-value |
| | Manual | 11.522 | 0 | 12.478 | 0 | 10.385 | 0 |
| | Maxent | 28.496 | 0 | 28.142 | 0 | 2.593 | 0.005 |
| Gridworld | AIRL | 13.610 | 0 | 5.117 | 0 | 4.266 | 0 |
| | PTIRL | 11.209 | 0 | 5.719 | 0 | 7.725 | 0 |
| | Manual | 18.801 | 0 | 17.375 | 0 | 6.955 | 0 |
| | Maxent | 32.341 | 0 | 12.104 | 0 | -2.586 | 0.995 |
| Bouncing Balls | AIRL | 45.020 | 0 | 28.226 | 0 | 19.488 | 0 |
| | PTIRL | 5.089 | 0 | 3.101 | 0.001 | 7.096 | 0 |
| | Manual | 16.152 | 0 | 15.851 | 0 | 16.786 | 0 |
| | Maxent | 17.829 | 0 | -2.543 | 0.994 | 9.123 | 0 |
| Drone Combat | AIRL | 9.772 | 0 | 8.023 | 0 | 3.935 | 0 |
| | PTIRL | 61.534 | 0 | 34.679 | 0 | 30.384 | 0 |
| | Manual | 24.419 | 0 | 20.633 | 0 | 15.760 | 0 |
| | Maxent | 6.171 | 0 | 1.717 | 0.04 | 2.233 | 0.013 |
| StarCraft II | AIRL | 4.992 | 0 | 4.300 | 0 | -2.913 | 0.998 |
| | PTIRL | 6.054 | 0 | 3.631 | 0 | 4.961 | 0 |
| | Manual | 4.409 | 0 | 2.23 | 0.01 | 3.04 | 0.001 |
| | Maxent | 8.02 | 0 | 3.19 | 0.043 | 10.89 | 0 |
| Robomimic | AIRL | 6.24 | 0 | 4.95 | 0 | 5.59 | 0 |
| | PTIRL | 4.27 | 0 | 0.65 | 0.16 | 1.18 | 0.04 |
| | Manual | 8.97 | 0 | 4.58 | 0 | 6.54 | 0 |
| | Maxent | 4.40 | 0 | 2.95 | 0.001 | 3.23 | 0 |
| Montezuma's Revenge | AIRL | 4.73 | 0 | 1.03 | 0.15 | 2.87 | 0 |
| | PTIRL | 2.30 | 0.01 | 0.763 | 0.223 | -0.51 | 0.69 |
| | Manual | 5.93 | 0 | 2.89 | 0 | 1.94 | 0.03 |
| | Maxent | 2.71 | 0 | 1.18 | 0.12 | 6.66 | 0 |
| MIMIC-IV | AIRL | 7.73 | 0 | 2.97 | 0 | 6.29 | 0. |
| | PTIRL | 2.14 | 0.02 | -0.55 | 0.71 | 0.65 | 0.26 |

In Table 10, we show the comprehensive results for the Welch's t-tests for unequal variances, which are conducted across all domain and reward type combinations, to test the null hypotheses: (1) $\mu_{\text{SRRD}} \leq \mu_{\text{DIRECT}}$, (2) $\mu_{\text{SRRD}} \leq \mu_{\text{EPIC}}$, and (3) $\mu_{\text{SRRD}} \leq \mu_{\text{DARD}}$; against the alternative: (1) $\mu_{\text{SRRD}} > \mu_{\text{DIRECT}}$, (2) $\mu_{\text{SRRD}} > \mu_{\text{EPIC}}$, and (3) $\mu_{\text{SRRD}} > \mu_{\text{DARD}}$, where $\mu$ represents the sample mean. Generally, the tests indicate that (1) $\mu_{\text{SRRD}} > \mu_{\text{DIRECT}}$ for all instances; (2) $\mu_{\text{SRRD}} > \mu_{\text{EPIC}}$ for 22 out of 28 instances, and (3) $\mu_{\text{SRRD}} > \mu_{\text{DARD}}$ for 24 out of 28 instances. These tests are performed at a significant level of $\alpha = 0.05$, assuming normality as per central limit theorem, since the number of trials is 200. In summary, we conclude that the SRRD pseudometric is more effective at classifying reward samples compared to its baselines. Detailed accuracy scores with variability are shown in Table 11.

Table 11: Experiment 2: Accuracy Scores

| Domain | Rewards | DIRECT | EPIC | DARD | SRRD |
|---|---|---|---|---|---|
| Gridworld | Manual | $69.8 \pm 4.6$ | $69.3 \pm 4.6$ | $70.0 \pm 5.0$ | $75.8 \pm 4.6$ |
| | Maxent | $57.4 \pm 4.5$ | $57.5 \pm 4.5$ | $68.9 \pm 4.5$ | $70.0 \pm 4.4$ |
| | AIRL | $82.3 \pm 3.0$ | $84.9 \pm 1.8$ | $85.0 \pm 2.6$ | $86.2 \pm 2.7$ |
| | PTIRL | $82.2 \pm 3.5$ | $84.2 \pm 3.3$ | $83.4 \pm 3.5$ | $86.0 \pm 3.3$ |
| Bouncing Balls | Manual | $46.5 \pm 4.8$ | $47.3 \pm 4.6$ | $52.0 \pm 4.8$ | $55.2 \pm 4.5$ |
| | Maxent | $39.7 \pm 3.1$ | $46.0 \pm 3.3$ | $50.8 \pm 3.2$ | $49.9 \pm 3.2$ |
| | AIRL | $41.2 \pm 3.4$ | $46.1 \pm 3.9$ | $49.8 \pm 3.3$ | $56.3 \pm 3.3$ |
| | PTIRL | $70.3 \pm 4.2$ | $71.1 \pm 4.3$ | $69.5 \pm 4.1$ | $72.4 \pm 4.0$ |
| Drone Combat | Manual | $67.1 \pm 4.1$ | $67.2 \pm 4.2$ | $66.2 \pm 4.9$ | $73.9 \pm 4.2$ |
| | Maxent | $70.3 \pm 3.7$ | $77.7 \pm 3.8$ | $73.2 \pm 4.2$ | $76.8 \pm 3.5$ |
| | AIRL | $90.1 \pm 3.9$ | $90.7 \pm 3.9$ | $92.3 \pm 3.7$ | $93.8 \pm 3.7$ |
| | PTIRL | $52.5 \pm 4.3$ | $63.7 \pm 4.3$ | $65.1 \pm 4.6$ | $78.3 \pm 4.1$ |
| StarCraft II | Manual | $65.5 \pm 4.6$ | $67.4 \pm 4.5$ | $69.5 \pm 4.5$ | $76.5 \pm 4.4$ |
| | Maxent | $72.3 \pm 4.1$ | $74.1 \pm 4.1$ | $73.9 \pm 4.2$ | $74.8 \pm 4.1$ |
| | AIRL | $75.1 \pm 4.0$ | $75.3 \pm 4.0$ | $78.1 \pm 3.8$ | $77.0 \pm 3.8$ |
| | PTIRL | $77.2 \pm 4.1$ | $78.1 \pm 4.2$ | $77.6 \pm 4.2$ | $79.6 \pm 4.0$ |
| Robomimic | Manual | $78.2 \pm 7.6$ | $80.3 \pm 9.3$ | $79.5 \pm 7.9$ | $82.4 \pm 8.6$ |
| | Maxent | $82.3 \pm 8.3$ | $86.8 \pm 7.9$ | $79.5 \pm 9.1$ | $89.8 \pm 8.8$ |
| | AIRL | $85.9 \pm 8.7$ | $87.1 \pm 9.1$ | $86.3 \pm 9.7$ | $91.8 \pm 9.5$ |
| | PTIRL | $80.3 \pm 7.7$ | $83.6 \pm 8.3$ | $83.1 \pm 8.5$ | $84.2 \pm 8.1$ |
| Montezuma's Revenge | Manual | $66.4 \pm 7.5$ | $70.1 \pm 8.3$ | $68.3 \pm 7.7$ | $73.5 \pm 7.5$ |
| | Maxent | $67.8 \pm 8.1$ | $69.1 \pm 8.3$ | $68.7 \pm 7.4$ | $71.2 \pm 7.9$ |
| | AIRL | $68.2 \pm 8.3$ | $71.4 \pm 8.7$ | $69.8 \pm 7.9$ | $72.3 \pm 8.2$ |
| | PTIRL | $68.2 \pm 7.9$ | $69.6 \pm 7.6$ | $70.6 \pm 7.9$ | $70.2 \pm 7.7$ |
| MIMIC-IV | Manual | $53.5 \pm 9.7$ | $56.5 \pm 9.2$ | $57.3 \pm 10.1$ | $59.2 \pm 9.5$ |
| | Maxent | $57.8 \pm 8.51$ | $59.1 \pm 9.5$ | $53.1 \pm 9.7$ | $60.2 \pm 9.2$ |
| | AIRL | $56.5 \pm 8.7$ | $60.7 \pm 8.6$ | $57.6 \pm 9.2$ | $63.3 \pm 8.9$ |
| | PTIRL | $58.9 \pm 9.4$ | $61.4 \pm 8.9$ | $60.3 \pm 9.1$ | $60.9 \pm 9.3$ |

