# OpenReview forum: "Reward Distance Comparisons Under Transition Sparsity"
_TMLR — Accepted by TMLR_

### Review · Reviewer_TrYj · 2025-01-21

**Summary Of Contributions:**

Reward functions govern the behavior of agents and therefore a sense of reward similarity is helpful in discerning how similar agent-learned behaviors are. This paper comprehensively reviewed the existing work EPIC and DARD on how they approached the problem, followed by the proposal to address their drawback of assumed high sample coverage. The authors provided many proofs to justify the proposed SRRD.

**Audience:**

Yes

**Claims And Evidence:**

Yes

**Requested Changes:**

Please refer to the weakness section above for my requested changes. I believe the paper can benefit from streamlining the main text into 12 pages so it becomes a short paper in TMLR.

**Strengths And Weaknesses:**

Disclaimer: I am not an expert in reward shaping or metrics and I cannot fully evaluate the correctness of this paper.

**Strengths:**
 - This paper addresses an important problem. Since reward functions dictate agent behaviors, direct reward comparison without computing the policies or value functions can provide important information to practitioners that design RL agents.
 - This paper meticulously reviewed existing work and their technicality. Based on the survey in Section 3, the authors pointed out a drawback of DARD with a didatic example that it requires high sample coverage which is intractable for large-scale problems.
 - The authors provided comprehensive theoretical justifications as well as sample-based version for the proposed method.

**Weaknesses:**
 - Because I am not an expert in this area, I find it hard to fully follow and appreciate the technical details the authors scattered in Section 3 and 4.  I was lost several times when trying to get through Section 3. The current draft has 19 pages which falls into the long paper category of TMLR, and in my opinion much of it can be moved to the appendix to keep the main story succinct. I believe it would help the reader to get to the point faster by having less than 12 pages in the main text so it becomes a short paper. This can be done by for example, merging Section 3.1-3.3 that provide preliminaries and Section 3.4 - 3.5 that discusses EPIC and DARD. Section 3.6 is okay as it reveals the claimed drawback of DARD. But having so much text in Section 4 before 4.1 seems a bit awkward from a paper structure viewpoint. The paper could benefit from clearly stating its proposal and key points to note in Section 4, followed by a list of proofs in a separate section. This way a reader like me can still appreciate the key idea without needing to wade through the technical part.

---

> ### Author Response · Authors · 2025-02-05
> **Response to Reviewer TrYj**
>
> **The current draft has 19 pages which falls into the long paper category of TMLR, and in my opinion much of it can be moved to the appendix to keep the main story succinct. I believe it would help the reader to get to the point faster by having less than 12 pages in the main text so it becomes a short paper. This can be done by for example, merging Section 3.1-3.3 that provide preliminaries and Section 3.4 - 3.5 that discusses EPIC and DARD. Section 3.6 is okay as it reveals the claimed drawback of DARD. But having so much text in Section 4 before 4.1 seems a bit awkward from a paper structure viewpoint. The paper could benefit from clearly stating its proposal and key points to note in Section 4, followed by a list of proofs in a separate section. This way a reader like me can still appreciate the key idea without needing to wade through the technical part.**
>
> > Thank you very much for the feedback. We agree that the previous version of the paper might have been more dense (19 pages), and part of the challenge was that in our previous TMLR submission, a couple of reviewers mentioned that we should add more theoretical components that we had placed in the Appendix, to the main paper (propositions, definitions and approximations) for theoretical rigor and comprehensiveness.
>
> > However, we do share the same sentiment for succinctness. Therefore, we have decided to move some of the propositions, definitions etc. back to the Appendix to streamline the main section. We have also moved most of the approximations to the Appendix and we believe the paper still maintains its rigor and communication. We have compressed section 3.1-3.6 to Section 3.1-3.3, and we have also reduced the size of Section 4. As of now, the main segment of the paper now has 16 pages, and we are contemplating if we should move section 4.1 to the Appendix to reach towards 12 pages. However, we believe this section is also important as a theoretical reference to evaluate the algorithms. We welcome any further advice on where additional cuts or reorganizations could enhance clarity without compromising completeness.

---

### Review · Reviewer_DLvw · 2025-01-21

**Summary Of Contributions:**

This paper introduces the Sparsity Resilient Reward Distance (SRRD), a pseudometric designed to robustly compare reward functions in reinforcement learning under transition sparsity, where data coverage is limited due to sampling challenges or feasibility constraints. SRRD eliminates the need for high transition coverage by leveraging observed transition distributions and maintaining shaping invariance, ensuring accurate comparisons even with sparse or biased data. Theoretical guarantees for SRRD’s robustness are provided, and extensive experiments across multiple domains (e.g., Gridworld, StarCraft II) demonstrate its superiority over existing methods like EPIC and DARD, particularly in scenarios with sparse transitions.

**Audience:**

Yes

**Broader Impact Concerns:**

No concern.

**Claims And Evidence:**

No

**Requested Changes:**

- More experiments are needed both in simulated environment and real world applications. The author should justify why the experimental platform are chosen and why that testbeds results would imply the applicability of the proposed methods in real world scenarios. For example, include experiments on real-world datasets with inherent sparsity (e.g., healthcare or autonomous driving). Additionally, consider Atari game series for benchmarking, e.g. Montezuma's revenge.
- Simplify the mathematical derivations or provide intuitive summaries alongside the equations.

**Strengths And Weaknesses:**

**Strength**
- The paper tackles a significant issue in reward function comparisons under transition sparsity, a common challenge in real-world reinforcement learning (RL) applications.
- Sparsity Resilient Reward Distance (SRRD) pseudometric is theoretically innovative and addresses a gap in prior methods like EPIC and DARD.
- I appreciate that the authors provide detailed theoretical justifications for SRRD, including its robustness to transition sparsity.
- I also like the comment after each theoretical results, which help reader for intuitively understanding

**Weakness**
- All the figures are poorly plotted. The authors should use vector graphics during plotting. The most of the figures blur when enlarged.
- The paper’s structure is dense and could benefit from a clearer separation between theoretical developments, empirical studies, and practical implications
- I found it is confusing why the authors directly choose Gridworld, Bouncing Balls, Drone Combat and etc. as the testing platform. There is  no justification for the experimental setup, nor why these environments would sufficiently demonstrate the applicability of SRRD in real world problems. In fact, without exploration with real-world related experiments, it is difficult to see how SRRD would have some solid impact, e.g. military scenarios mentioned during introduction.

---

> ### Author Response · Authors · 2025-02-05
> **Response to Reviewer DLvw**
>
> **All the figures are poorly plotted. The authors should use vector graphics during plotting. The most of the figures blur when enlarged. The paper’s structure is dense and could benefit from a clearer separation between theoretical developments, empirical studies, and practical implications.**
>
> > Thank you very much, we have attempted to simplify the content of the paper especially the preliminaries and the approach section. We have also moved some propositions as well as exact details on approximations, to the Appendix, to ensure that the paper is much more readable. We are also working to update the current plots using vector graphics.
>
> **More experiments are needed both in simulated environment and real world applications. The author should justify why the experimental platform are chosen and why that testbeds results would imply the applicability of the proposed methods in real world scenarios. For example, include experiments on real-world datasets with inherent sparsity (e.g., healthcare or autonomous driving). Additionally, consider Atari game series for benchmarking, e.g. Montezuma's revenge.**
>
> > Thank you very much for the feedback, we appreciate your emphasis on broader experimentation. As we mentioned in the paper, we incorporated the Gridworld and the Bouncing Balls domains, since, these domains are relatively simple, and they provide the flexibility to vary parameters, and hence, control the level of transition sparsity, which can help with assessing how the reward comparison metrics vary due to limited sampling or feasibility constraints (Experiment 1). These datasets have also been extensively studied in prior works, for example, Gleave et al. 2020, utilizes the Gridworld environment to analyze different scenarios while studying EPIC, and the Bouncing Balls environment is also used in Wulfe et al 2021 in DARD.
>
> > However, these datasets are generally too simplistic and might not be representative of real-world scenarios. Therefore, we incorporate the Drone Combat and StarCraft 2 environment into Experiment 2. The Drone Combat environment is adapted from a gym predator-prey environment, and we add complexity to simulate a combat scenario between two swarm teams under different strategies. The StarCraft 2 environment simulates a battle between a multiagent ally team and the default AI team. We included StarCraft 2 since its one of the most complex and well-studied simulated environment, with a large number of states and features (almost infinite). We however acknowledge the need for additional and diverse datasets. Therefore, in addition to these four domains, have we incorporated the suggested Atari game, Montezuma’s Revenge. We found an online repository with a collection of real human generated data (https://github.com/yobibyte/atarigrandchallenge The Atari Grand Challenge Dataset), and we labelled this dataset into 3 categories: novice, expert, and moderate based on the game scores. We then performed Experiment 2, where, the goal was to classify behaviors, using reward pseudometrics, to determine the level of expertise of the agents based on reward functions (game score, and IRL rewards). This dataset is complex, and trajectories are a sequence of image data, hence, the number of states and actions is almost unbounded.
>
> > Based on the comment from Reviewer ZWFj, we also decided to add an additional offline Robomimic dataset (robomimic). This dataset is open source and it includes a collection of human demonstrations, and trained behavioral models (via imitation learning), at performing a variety of robotic manipulation tasks. The dataset also contains three categories: human, simulated, mixed agents etc; and we performed the second experiment (k-NN classification), to determine if reward distances could help classify the label of an agent’s trajectories. This dataset offers some realism, as it contains real-human demonstrations, and trajectories are a combination of image data, with near infinite states and actions.
>
> > In our latest revision, we have also added results from the MIMIC-IV dataset (https://physionet.org/content/mimiciv/3.1/), which contains electronic medical health data for patients admitted into an emergency or ICU center, at a hospital in Boston. We treat each patient’s diagnoses per visit as the state, and the prescribed medication as the action. Based on their initial diagnosis upon first hospital visit, we group patients into five illness classes—diabetes, kidney, limb, respiratory, and substance abuse—even though the future diagnoses generally evolve over time. We then conduct Experiment 2 to classify patient diagnosis histories using reward pseudometrics (see Section 5.2 and Appendix C.5 for more details).

---

### Review · Reviewer_ZWFj · 2025-01-26

**Summary Of Contributions:**

This paper introduces SRRD pseudometric, a robust method for comparing reward functions under transition sparsity. SRRD eliminates the need for high transition coverage by leveraging sampled transitions, making it more effective than existing methods like EPIC and DARD. It is theoretically invariant to shaping and empirically outperforms prior approaches across diverse domains, demonstrating practical utility for reward comparison and agent behaviour classification.

**Audience:**

Yes

**Claims And Evidence:**

Yes

**Requested Changes:**

1. Could the author provide more details about the experiments? For instance, in the $20 \times 20$ Gridworld domain, what is the specific goal of the agent, and how is the ground truth reward function defined? Including such details would clarify the experimental setup.

2. I think that goal-conditioned offline reinforcement learning could serve as a valuable benchmark for reward comparison. In this scenario, the agent receives a positive reward of $1$ only when it successfully achieves the goal (e.g., controlling a robotic arm to manipulate a cube to a designated position). The sparse reward would represent the optimal reward function, and the offline dataset could consist of policy rollouts, creating an ideal setting for evaluating reward comparison methods. Do you think whether it makes sense?

3. While SRRD is robust to transition sparsity, do you think its performance might still depend on the quality and representativeness of the sampled transitions? In highly sparse environments, where critical transitions may remain unsampled, could this affect the accuracy and reliability of SRRD's reward comparisons?

**Strengths And Weaknesses:**

**Strengths**

1. SRRD is designed to handle sparse sampling conditions, outperforming existing methods like EPIC and DARD.

2. The method is theoretically proven to be invariant to potential-based shaping.

3. SRRD avoids the computational cost of policy learning, making it scalable and effective for real-world applications.

4. Extensive experiments validate SRRD’s superior performance across multiple environments, even under conditions of high transition sparsity.


**Weaknesses**

The paper could be organised more effectively:

1. The author might consider switching the second and third paragraphs in Section 1. Presenting the objective of reward comparison first, followed by its applications, could provide a clearer structure for readers. Additionally, only a brief summary of the applications could be appended to the end of the third paragraph for better flow.

2. The author should briefly introduce SRRD in Section 1 and explain the rationale behind its ability to address transition sparsity.

3. The Preliminaries and Approach sections include numerous definitions, propositions, and existing methods without sufficient connections, making them challenging to follow.

While this paper appears robust, I suggest the author refine its presentation for improved clarity and readability.

---

> ### Author Response · Authors · 2025-02-05
> **Response to Reviewer ZWFj**
>
> **The author might consider switching the second and third paragraphs in Section 1. Presenting the objective of reward comparison first, followed by its applications, could provide a clearer structure for readers. Additionally, only a brief summary of the applications could be appended to the end of the third paragraph for better flow. The author should briefly introduce SRRD in Section 1 and explain the rationale behind its ability to address transition sparsity.**
>
> >Thank you for this valuable feedback. We have revised the introduction so that the objective of reward comparison appears earlier. We also streamlined the discussion of applications, moving some details to the end of the third paragraph to create a more logical flow and progression. We now provide a concise overview of SRRD in Section 1, including a brief explanation of why it can better handle transition sparsity.
>
> **The Preliminaries and Approach sections include numerous definitions, propositions, and existing methods without sufficient connections, making them challenging to follow.**
>
> > We have streamlined the preliminaries and approach section, by limiting the number of propositions and definitions, and moving some of these to the appendix. We have also attempted to rephrase statements to make the connections flow better.
>
> **Could the author provide more details about the experiments? For instance, in the 20×20 Gridworld domain, what is the specific goal of the agent, and how is the ground truth reward function defined? Including such details would clarify the experimental setup.**
>
> > Thank you for the request for more clarity. In the Gridworld task, especially in Experiment 1, the general task is for an agent to traverse from the top left corner of the Grid, $ (0, 0)$, to the bottom right corner $(19, 19)$, within a limit of 200 steps. Since our goal is to examine the impact of different types of reward functions on the reward comparison pseudometrics, we varied the nature of the ground truth reward functions from sinusoidal, polynomial, and random, as described in Section 5. Note that these rewards are not necessarily based on reaching the terminal state, but are assigned based on state features, which are the $(x, y)$ coordinates for the Gridworld. This setup ensures that we can evaluate a broad range of reward function variations.
>
> **I think that goal-conditioned offline reinforcement learning could serve as a valuable benchmark for reward comparison. In this scenario, the agent receives a positive reward of 1 only when it successfully achieves the goal (e.g., controlling a robotic arm to manipulate a cube to a designated position). The sparse reward would represent the optimal reward function, and the offline dataset could consist of policy rollouts, creating an ideal setting for evaluating reward comparison methods. Do you think whether it makes sense?**
>
> > Yes indeed, we agree that the goal-conditioned offline RL domains could serve as valuable benchmarks, to evaluate behaviors under sparse rewards. In fact, since reviewer **DLvw** requested that we add more experimental domains, we have added the Robomimic domain (https://robomimic.github.io/) to our benchmarks, which contains samples of human and simulated robotic manipulation tasks, with sparse rewards that are based primarily on task completion. We then evaluate the different reward pseudometrics at their accuracy in determining the type of agent (human, trained agent, mixed agents) that generated some given behavior based on reward similarity (Experiment 2). In addition, we also explore the effectiveness of IRL generated rewards at this task.
>
> **While SRRD is robust to transition sparsity, do you think its performance might still depend on the quality and representativeness of the sampled transitions? In highly sparse environments, where critical transitions may remain unsampled, could this affect the accuracy and reliability of SRRD's reward comparisons?**
>
> > Yes, SRRD is robust against transition sparsity compared to the other pseudometrics, however, its performance is also impacted by the quality and representativeness of the sampled transitions. Ideally, SRRD as well as the other pseudometrics perform relatively better with samples that have a broader coverage of the state-action space. In Appendix B.4 we examine how SRRD is sensitive to the quality and distribution of sampled transitions, and we see that indeed, SRRD’s performance can vary due to biased samples. However, compared to the EPIC and DARD pseudometrics, we generally find SRRD being much more robust under the given sampled transitions, which explains its superior performance in Experiment 2, where we test its performance in a wider range of domains, where we have no control over the quality of the data and sparsity.

---

### Author Response · Authors · 2025-02-22
**Summary of Changes in Response to Reviewer Feedback**

**We sincerely thank the reviewers for their insightful feedback and valuable suggestions, which have greatly improved this paper. In summary, we have implemented the following changes:**

> Updated all plots to vector graphics for enhanced clarity and resolution.

> Reorganized the introduction and streamlined the paper’s structure: merged Sections 3.1–3.6 into Sections 3.1–3.3, reduced the main body from 19 to 16 pages, and relocated approximations and certain propositions (e.g., regret bounds) to the Appendix to improve readability and accessibility.

> Incorporated additional datasets, Montezuma's Revenge, Robomimic and MIMIC-IV to diversify our testbeds.

> We believe that these seven selected datasets effectively showcase realistic complexity and rich diversity, while aligning with prior works and established RL literature. We also welcome references to additional open-source datasets for further exploration.

---

### Decision · Action_Editor_qS1E · 2025-03-18

**Recommendation:** Accept as is

**Comment:**

Though all reviewers agreed that the paper had sufficient evidence and a substantial audience, some of the reviewers expressed a concern about the paper's clarity. The paper is indeed technically dense. The authors made substantial revisions to the paper in response to these concerns, streamlining some sections, adding more intuition in the discussion, and moving some proofs and derivations to the appendix. Based on my own reading of the revised manuscript (aligning with one of the reviewer's assessment), I believe that, though the paper does require active engagement from the reader to follow the theoretical discussion, it is sufficiently clear and readable (and provides sufficient high-level discussion) to reach its audience.

**Audience:**

The reviewers agree that the paper addresses interesting and important issues and that there is an audience for the insights presented.

**Claims And Evidence:**

The reviewers agree that the theoretical support for the paper's claims is thorough and sound. The authors have considerably expanded the empirical results in response to reviewer comments, and the paper now more thoroughly supports claims of the generality of the approach.

---

> ### Author Response · Authors · 2025-04-09
> **Camera-Ready Submission for TMLR Paper3831**
>
> Dear Action Editor,
>
> We sincerely thank you and the reviewers for your time, effort, and valuable feedback throughout the review process. We are grateful for the thoughtful comments and are pleased that our paper has been accepted.
>
> We have now uploaded the camera-ready version of the paper.
>
> Thank you once again for your support and for overseeing our submission.
>
> Sincerely,
> The Authors of TMLR Paper3831